# Estimating Generalization Performance Along the Trajectory of Proximal SGD in Robust Regression

**Kai Tan**
Department of Statistics
Rutgers University
Piscataway, NJ 08854
kai.tan@rutgers.edu

**Pierre C. Bellec**
Department of Statistics
Rutgers University
Piscataway, NJ 08854
pierre.bellec@rutgers.edu

## Abstract

This paper studies the generalization performance of iterates obtained by Gradient Descent (GD), Stochastic Gradient Descent (SGD) and their proximal variants in high-dimensional robust regression problems. The number of features is comparable to the sample size and errors may be heavy-tailed. We introduce estimators that precisely track the generalization error of the iterates along the trajectory of the iterative algorithm. These estimators are provably consistent under suitable conditions. The results are illustrated through several examples, including Huber regression, pseudo-Huber regression, and their penalized variants with non-smooth regularizer. We provide explicit generalization error estimates for iterates generated from GD and SGD, or from proximal SGD in the presence of a non-smooth regularizer. The proposed risk estimates serve as effective proxies for the actual generalization error, allowing us to determine the optimal stopping iteration that minimizes the generalization error. Extensive simulations confirm the effectiveness of the proposed generalization error estimates.

## 1 Introduction

Consider the linear model:
$$\boldsymbol{y} = \boldsymbol{X}\boldsymbol{b}^* + \boldsymbol{\varepsilon}, \tag{1}$$
where $\boldsymbol{y} \in \mathbb{R}^n$ is the response vector, $\boldsymbol{X} \in \mathbb{R}^{n \times p}$ is the design matrix, $\boldsymbol{b}^* \in \mathbb{R}^p$ is the unknown regression vector, and $\boldsymbol{\varepsilon} \in \mathbb{R}^n$ is the noise vector that we assume independent of $\boldsymbol{X}$. The entries of $\boldsymbol{\varepsilon}$ may be heavy-tailed, for instance our working assumptions allow for infinite second moment.

For the estimation of $\boldsymbol{b}^*$, we consider the following regularized optimization problem
$$\widehat{\boldsymbol{b}} \in \underset{\boldsymbol{b} \in \mathbb{R}^p}{\arg\min} \frac{1}{n} \sum_{i=1}^{n} \rho(y_i - \boldsymbol{x}_i^\top \boldsymbol{b}) + g(\boldsymbol{b}), \tag{2}$$
where $\rho : \mathbb{R} \to \mathbb{R}$ is a data-fitting loss and $g : \mathbb{R}^p \to \mathbb{R}$ is a regularization function. In the present robust regression setting, typical examples of $\rho$ include the Huber [14] loss $\rho(r; \delta) = \delta^2 \int_0^{|r/\delta|} \min(1, x)\, \mathrm{d}x$, the Pseudo-Huber loss $\rho(r; \delta) = \delta^2(\sqrt{1 + (r/\delta)^2} - 1)$ or other Lipschitz loss functions to combat the possible heavy-tails of the additive noise. Typical examples of penalty functions include the L1/Lasso [27] penalty $g(\boldsymbol{b}) = \lambda\|\boldsymbol{b}\|_1$, group-Lasso penalty [29] for grouped variables, or their non-convex variants including for instance SCAD [12] or MCP [30].

In order to solve the optimization problem (2), practitioners resort to iterative algorithms, for instance gradient descent, accelerated gradient descent, stochastic gradient descent, and the corresponding proximal methods [20] in the presence of a non-smooth regularizer. Let the algorithm starts with

some initializer $\widehat{\boldsymbol{b}}^1 \in \mathbb{R}^p$ (typically $\widehat{\boldsymbol{b}}^1 = \boldsymbol{0}$) followed by consecutive iterates $\widehat{\boldsymbol{b}}^2, \widehat{\boldsymbol{b}}^3, \ldots$, where $\widehat{\boldsymbol{b}}^t$ is typically obtained, for gradient descent and its variants as will be detailed below, from $\widehat{\boldsymbol{b}}^{t-1}$ and by applying an additive correction involving the gradient of the objective function. Our goal of this paper is to quantify the predictive performance of each iterate $\widehat{\boldsymbol{b}}^t$.

We assume throughout that the covariance $\mathbb{E}[\boldsymbol{x}_i \boldsymbol{x}_i^\top] = \boldsymbol{\Sigma}$ of the feature vectors is finite. We measure the predictive performance of $\widehat{\boldsymbol{b}}^t$ using the out-of-sample error

$$\mathbb{E}\Big[ \Big( \boldsymbol{x}_{new}^\top \widehat{\boldsymbol{b}}^t - \boldsymbol{x}_{new}^\top \boldsymbol{b}^* \Big)^2 \mid (\boldsymbol{x}_i, y_i)_{i \in [n]} \Big] = \|\boldsymbol{\Sigma}^{1/2}(\widehat{\boldsymbol{b}}^t - \boldsymbol{b}^*)\|^2$$

where $\boldsymbol{x}_{new}$ is a new feature vector, independent of the data $(\boldsymbol{x}_i, y_i)_{i \in [n]}$ and has the same distribution as $\boldsymbol{x}_i$. The above squared metric is used because the noise $\varepsilon_i$ (and thus $y_i$) is allowed to have infinite variance, and in this case the squared prediction error $\mathbb{E}[(\boldsymbol{x}_{new}^\top \widehat{\boldsymbol{b}}^t - y_{new})^2 \mid (\boldsymbol{x}_i, y_i)_{i \in [n]}] = +\infty$ irrespective of the value of $\widehat{\boldsymbol{b}}^t$.

The paper proposes to estimate the out-of-sample error $\|\boldsymbol{\Sigma}^{1/2}(\widehat{\boldsymbol{b}}^t - \boldsymbol{b}^*)\|^2$ of the $t$-th iterate using the right-hand side of the approximation

$$\|\boldsymbol{\Sigma}^{1/2}(\widehat{\boldsymbol{b}}^t - \boldsymbol{b}^*)\|^2 + \|\boldsymbol{\varepsilon}\|^2/n \approx \left\| (\boldsymbol{y} - \boldsymbol{X}\widehat{\boldsymbol{b}}^t) + \sum_{s=1}^{t-1} w_{t,s} \boldsymbol{S}_s \boldsymbol{\psi}(\boldsymbol{y} - \boldsymbol{X}\widehat{\boldsymbol{b}}^s) \right\|^2 / n, \qquad (3)$$

where $\boldsymbol{\psi} : \mathbb{R}^n \to \mathbb{R}$ is the derivative of $\rho$ acting component-wise on each coordinate in $\mathbb{R}^n$ and $\boldsymbol{S}_s$ is a diagonal matrix of the form $\boldsymbol{S}_s = \sum_{i \in I_s} \boldsymbol{e}_i \boldsymbol{e}_i^\top$ where $I_s \subset [n]$ is the batch for the $s$-th stochastic gradient update and $\boldsymbol{e}_i \in \mathbb{R}^n$ is the $i$-th canonical basis vector. Here the $w_{t,s}$ are quantities, introduced in Section 3.3 below, that can be computed from data and do not require the knowledge of $\boldsymbol{\Sigma}$. The approximation (3) is made rigorous in Theorem 3.6, where the right-hand side is proved to be consistent (i.e., the difference between the two sides of the inequality converges to 0 in probability) for a first set of weights $(w_{s,t})_{s<t}$, and in Theorem 3.7 where a second set of weights are proposed.

Because the right-hand side of (3) is observable from the data and the iterates $(\widehat{\boldsymbol{b}}^s)_{s \le t}$ are computed from the iterative algorithm, the approximation (3) lets us compare the out-of-sample error of iterates $\widehat{\boldsymbol{b}}^t$ at different time $t$ up to the additive term $\|\boldsymbol{\varepsilon}\|^2/n$ (which does not depend on $t$ nor on the choice of the iterative scheme or the choice of loss and penalty). It also lets us compare different tuning parameters, for instance learning rate, multiplicative parameter of the penalty function, batch size in Stochastic Gradient Descent (SGD). The right-hand side of (3) can serve as the criteria to choose the iteration number or tuning parameters that achieves the smallest out-of-sample error.

## 1.1 Related literature

Estimation of prediction risk of regression estimates has received significant attention in the last few decades. One natural avenue to estimate the generalization performance is to use $V$-fold cross-validation or leave-one-out schemes. In the proportional regime of interest here, where dimension $p$ and sample size $n$ are of the same order, $V$-fold cross-validation with finite $V$, e.g., $V = 5, 10$ is known to fail at consistently estimate the risk of the estimator trained on the full dataset [23, Figure 1]; this is simply explained because training with the biased sample size $n(V-1)/V$ may behave differently than training with the full dataset. Leave-one-out schemes, or drastically increasing $V$, requires numerous refitting and is thus computationally expensive.

This motivates computationally efficient estimates of the risk of an estimator trained on the full dataset without sample-splitting, including Approximate Leave-One-out (ALO) schemes [23] that do not rely on sample-splitting and refitting; see [1] and the references therein for recent developments. For ridge regression and other estimators constructed from the square loss, the Generalized Cross-Validation (GCV) [28] has been shown be to be effective, and it avoids data-splitting and refitting; it only needs to fit the full data once and then adjust the training error by a multiplicative factor larger than 1. Beyond ridge regression, the extension of GCV using degrees-of-freedom has been studied for Lasso regression [2, 3, 18, 9], and alternatives were developed for robust M-estimators [3, 4]. While ALO or GCV and its extensions are good estimators of the predictive risk of a solution $\widehat{\boldsymbol{b}}$ to the optimization problem (2), they are not readily applicable to quantify the prediction risk of iterates $\widehat{\boldsymbol{b}}^t$ obtained by widely-used iterative algorithms such as gradient descent (GD), stochastic gradient

descent (SGD) or their proximal variants: ALO and GCV focus on estimating the final $(t \to +\infty)$ iterate of the algorithm, when a solution $\widehat{\boldsymbol{b}}$ in (2) is found. Our goal in the present paper is to develop risk estimation methodologies along the trajectory of the algorithm.

Luo et al. [17] developed methods to estimate the cross-validation error of iterates that solves an empirical risk minimization problem. Their approach requires the Hessian of the objective function to be well-conditioned (i.e., the smallest and largest eigenvalues are bounded) along all iterates. This condition is not satisfied for the regression problems we consider in this paper, such as high-dimensional robust regression with a Lasso penalty. In the context of least squares problems with both $p$ and $n$ being large, [7] studied the fundamental limits on the performance of first order methods, showing that these are dominated by a specific Approximate Message Passing algorithm. Paquette et al. [19] demonstrated that the dynamics of Stochastic Gradient Descent (SGD) become deterministic in the large sample and dimensional limit, providing explicit expressions for these dynamics when the design matrix is isotropic. Our work differs from [19] in two key ways: First, we address a more general regression problem incorporating a non-smooth regularizer, thereby considering both SGD and proximal SGD; second, we offer explicit risk estimates for each iteration, rather than focusing solely on the theoretical dynamics of the iterates. Celentano et al. [8] and [13] characterize the dynamical mean-field dynamics of iterative schemes, and identify that the limiting process involves a "memory" kernel, describing how the dynamics of early iterates affect later ones.

Most recently, and most closely related to the present paper, [5] proposed risk estimate for iterates $\widehat{\boldsymbol{b}}^t$ obtained by running gradient descent and proximal gradient descent methods for solving penalized least squares optimizations. However, [5] focuses exclusively on the square loss for $\rho$ in (3), which is not readily applicable to robust regression with heavy tailed noise for which the Huber or other robust losses must be used. Bellec and Tan [5] is further restricted to gradient updates using the full dataset, which does not cover stochastic gradient descent. While several proof techniques used in the present paper are inspired by [5], we will explain in Remark 3.8 that directly generalizing [5] to SGD in robust regression leads to a poor risk estimate for small batch sizes. The proposal of the present paper leverages out of batch samples to overcome this issue.

For gradient descent for the square loss and without penalty, Patil et al. [21] demonstrates both the failure of GCV along the trajectory and the success of computationally expensive leave-one out schemes, and develops a proposal to reduce the computational cost. Finally, let us mention the works [10, 16] that characterize the dynamics of the iterates in phase retrieval and matrix sensing problems, assuming that a fresh batch of observations (independent of all previous updates) is used at each iteration. This is different from the usual SGD setting studied in the present paper where the observations used during a stochastic gradient update may be reused in future stochastic gradient updates, creating intricate probabilistic dependence between gradient updates at different iterations.

Robust regression is highly valuable in real data analysis due to its ability to handle heavy-tailed noise effectively, and we will see below that the use of stochastic gradient updates and data-fitting loss functions different from the square loss require estimates that have a drastically different structure that in the square loss case. The present paper develops generalization error estimates in situations where no consistent estimate have been proposed: (1) we develop generalization error estimates along the trajectory of iterative algorithms aimed at solving (2) for robust loss functions including the pseudo-Huber loss; (2) the estimates are applicable not only to gradient updates involving the full dataset (gradient descent and its variants), but also to SGD and proximal SGD where a random batch is used for each update.

## 2 Problem setup

The paper studies iterative algorithms aimed at solving the optimization problem (2). We consider the algorithm that generates iterates $\widehat{\boldsymbol{b}}^t$ for $t = 1, 2, ..., T$ according to the following iteration:

$$\widehat{\boldsymbol{b}}^{t+1} = \phi_t\Big(\widehat{\boldsymbol{b}}^t + \frac{\eta_t}{|I_t|}\boldsymbol{X}^\top \boldsymbol{S}_t \psi(\boldsymbol{y} - \boldsymbol{X}\widehat{\boldsymbol{b}}^t)\Big), \tag{4}$$

where $\boldsymbol{S}_t \in \mathbb{R}^{n \times n}$ is the diagonal matrix $\boldsymbol{S}_t = \sum_{i \in I_t} \boldsymbol{e}_i \boldsymbol{e}_i^\top$ for $I_t \subset [n]$ the $t$-th batch (independent of $(\boldsymbol{X}, \boldsymbol{y})$), where $\phi_t : \mathbb{R}^p \mapsto \mathbb{R}^p$ and $\psi : \mathbb{R}^n \mapsto \mathbb{R}^n$ are two functions and $\eta_t$ is the step size. Typically, $\psi : \mathbb{R}^n \to \mathbb{R}^n$ is the componentwise application of $\rho'$ (where $\rho$ is the data-fitting loss in (2)), and the matrix $\boldsymbol{S}_t \in \mathbb{R}^{n \times n}$ is diagonal with elements in $\{0, 1\}$ encoding the observations $i \in [n]$

used in the $t$-th stochastic gradient update. The presence of $\boldsymbol{S}_t$ and possibly nonlinear function $\boldsymbol{\psi}$ is such that the above iteration scheme is not covered by previous related works including [5, 21], which only tackle $\boldsymbol{S}_t = \boldsymbol{I}_n$ (full batch gradient updates) and $\boldsymbol{\psi} : \mathbb{R}^n \to \mathbb{R}^n$ the identity map ($\rho$ in (2) restricted to be the square loss). The iterative scheme (4), on the other hand, covers SGD with robust loss functions.

In the next section, we first provide a few examples of algorithms encompassed in the general iteration (4). This includes Gradient Descent (GD), Stochastic Gradient Descent (SGD), and their corresponding proximal methods [20], Proximal GD and Proximal SGD. GD and SGD are widely used in practice, while the proximal methods are particularly useful for solving the optimization problem (2) with non-smooth regularizers.

## 2.1 Robust regression without penalty

If there are no penalties in (2), i.e., $g(\boldsymbol{b}) = 0$, then the minimization problem becomes

$$\widehat{\boldsymbol{b}} \in \underset{\boldsymbol{b} \in \mathbb{R}^p}{\arg\min} \frac{1}{n} \sum_{i=1}^{n} \rho(y_i - \boldsymbol{x}_i^\top \boldsymbol{b}).$$

To solve this problem, provided $\rho$ is differentiable, one may use gradient descent (SGD) and stochastic gradient descent (SGD).

**Example 2.1** (GD)**.** The GD method consists of the following iteration:

$$\widehat{\boldsymbol{b}}^{t+1} = \widehat{\boldsymbol{b}}^t + \tfrac{\eta_t}{n} \boldsymbol{X}^\top \boldsymbol{\psi}(\boldsymbol{y} - \boldsymbol{X}\widehat{\boldsymbol{b}}^t), \tag{5}$$

where $\boldsymbol{\psi}$ is the derivative of $\rho$ acting component-wise on its argument, and $\eta_t$ is the step size (also known as learning rate). For the least squares loss $\rho(x) = x^2/2$, we have $\boldsymbol{\psi}(\boldsymbol{u}) = \boldsymbol{u}$.

**Example 2.2** (SGD)**.** Suppose at $t$-th iteration, we use the batch $I_t \subset [n]$ to compute the gradient,

$$\widehat{\boldsymbol{b}}^{t+1} = \widehat{\boldsymbol{b}}^t + \frac{\eta_t}{|I_t|} \sum_{i \in I_t} \boldsymbol{x}_i \psi(y_i - \boldsymbol{x}_i^\top \widehat{\boldsymbol{b}}^t) = \widehat{\boldsymbol{b}}^t + \frac{\eta_t}{|I_t|} \boldsymbol{X}^\top \boldsymbol{S}_t \boldsymbol{\psi}(\boldsymbol{y} - \boldsymbol{X}\widehat{\boldsymbol{b}}^t), \tag{6}$$

where $\boldsymbol{S}_t = \sum_{i \in I_t} \boldsymbol{e}_i \boldsymbol{e}_i^\top$ and $\boldsymbol{e}_i$ is the $i$-th canonical vector in $\mathbb{R}^n$. If $I_t = [n]$ for each $t$, then $|I_t| = n$ and $\boldsymbol{S}_t = \boldsymbol{I}_n$, hence this SGD method reduces to the GD method in (5).

## 2.2 Robust regression with Lasso penalty

Regularized regression is useful for high-dimensional regression problems where $p$ is larger than $n$. We consider the Lasso penalty $g(\boldsymbol{b}) = \lambda\|\boldsymbol{b}\|_1$ to fight for the curse of dimensionality and obtain sparse estimates (our working assumptions, on the other hand, do not assume that the ground truth $\boldsymbol{b}^*$ is sparse). While GD and SGD are not directly applicable to solve the optimization problem (2) with Lasso penalty due to $\|\cdot\|_1$ lacking differentiability at 0, Proximal Gradient Descent (Proximal GD) [20] and Stochastic Proximal Gradient Descent (Proximal SGD) can be used to solve this optimization with Lasso penalty.

**Example 2.3** (Proximal GD)**.** For $g(\boldsymbol{b}) = \lambda\|\boldsymbol{b}\|_1$ in (2), the Proximal GD gives the following iterations:

$$\widehat{\boldsymbol{b}}^{t+1} = \mathrm{soft}_{\lambda\eta_t}\big(\widehat{\boldsymbol{b}}^t + \tfrac{\eta_t}{n} \boldsymbol{X}^\top \boldsymbol{\psi}(\boldsymbol{y} - \boldsymbol{X}\widehat{\boldsymbol{b}}^t)\big),$$

where $\mathrm{soft}_\theta(\cdot)$ applies the soft-thresholding $u \mapsto \mathrm{sign}(u)(|u| - \theta)_+$ component-wise.

**Example 2.4** (Proximal SGD)**.** Similar to the Proximal GD, the Proximal SGD consists of the following iterations:

$$\widehat{\boldsymbol{b}}^{t+1} = \mathrm{soft}_{\lambda\eta_t}\big(\widehat{\boldsymbol{b}}^t + \tfrac{\eta_t}{|I_t|} \boldsymbol{X}^\top \boldsymbol{S}_t \boldsymbol{\psi}(\boldsymbol{y} - \boldsymbol{X}\widehat{\boldsymbol{b}}^t)\big).$$

Let $\boldsymbol{\rho}' : \mathbb{R}^n \to \mathbb{R}^n$ be the function applies the derivative of $\rho : \mathbb{R} \to \mathbb{R}$ to each of its component, i.e., $\boldsymbol{\rho}'(\boldsymbol{u}) = (\rho'(u_1), ..., \rho'(u_n))^\top$. Then the above examples can be summarized in the following table with different definition of $\boldsymbol{\psi}$, $\boldsymbol{\phi}_t$, and $\boldsymbol{S}_t$.

Table 1: Specification of $\boldsymbol{\psi}, \boldsymbol{\phi}_t, \boldsymbol{S}_t$ for each algorithm

|  | GD | SGD | Proximal GD | Proximal SGD |
|---|---|---|---|---|
| $\boldsymbol{\psi}(\boldsymbol{u})$ | $\boldsymbol{\rho}'(\boldsymbol{u})$ | $\boldsymbol{\rho}'(\boldsymbol{u})$ | $\boldsymbol{\rho}'(\boldsymbol{u})$ | $\boldsymbol{\rho}'(\boldsymbol{u})$ |
| $\boldsymbol{\phi}_t(\boldsymbol{v})$ | $\boldsymbol{v}$ | $\boldsymbol{v}$ | $\mathrm{soft}_{\lambda\eta_t}(\boldsymbol{v})$ | $\mathrm{soft}_{\lambda\eta_t}(\boldsymbol{v})$ |
| $\boldsymbol{S}_t$ | $\boldsymbol{I}_n$ | $\boldsymbol{S}_t$ | $\boldsymbol{I}_n$ | $\boldsymbol{S}_t$ |

To define the proposed estimators of the generalization error, we further define the following Jacobian matrices:

$$\boldsymbol{D}_t = \frac{\partial \boldsymbol{\psi}(\boldsymbol{r})}{\partial \boldsymbol{r}}\Big|_{\boldsymbol{r}=\boldsymbol{y}-\boldsymbol{X}\widehat{\boldsymbol{b}}^t} \in \mathbb{R}^{n\times n}, \quad \widetilde{\boldsymbol{D}}_t = \frac{\partial \boldsymbol{\phi}_t(\boldsymbol{v})}{\partial \boldsymbol{v}}\Big|_{\boldsymbol{v}=\widehat{\boldsymbol{b}}^t + \frac{\eta_t}{|I_t|}\boldsymbol{X}^\top \boldsymbol{S}_t \boldsymbol{\psi}(\boldsymbol{y}-\boldsymbol{X}\widehat{\boldsymbol{b}}^t)} \in \mathbb{R}^{p\times p}.$$

Then, we have $\widetilde{\boldsymbol{D}}_t = \boldsymbol{I}_p$ for GD and SGD, and $\widetilde{\boldsymbol{D}}_t = \sum_{j\in\hat{S}_t} \boldsymbol{e}_j \boldsymbol{e}_j^\top$ for Proximal GD and Proximal SGD based on soft-thresholding, where $\hat{S}_t = \{j \in [p] : \boldsymbol{e}_j^\top \widehat{\boldsymbol{b}}^{t+1} \neq 0\}$.

## 3 Main results

**Assumption 3.1.** The design matrix $\boldsymbol{X}$ has i.i.d. rows from $\mathsf{N}_p(\boldsymbol{0}, \boldsymbol{\Sigma})$ for some positive definite matrix $\boldsymbol{\Sigma}$ satisfying $0 < \lambda_{\min}(\boldsymbol{\Sigma}) \leq 1 \leq \lambda_{\max}(\boldsymbol{\Sigma})$ and $\|\boldsymbol{\Sigma}\|_{\mathrm{op}}\|\boldsymbol{\Sigma}^{-1}\|_{\mathrm{op}} \leq \kappa$. We assume $\mathrm{Var}[\boldsymbol{x}_i^\top \boldsymbol{b}^*] \leq \delta^2$, that is, the signal of the model (1) is bounded from above.

**Assumption 3.2.** The noise $\boldsymbol{\varepsilon}$ is independent of $\boldsymbol{X}$ and has i.i.d. entries from a fixed distribution independent of $n, p$, with $\mathbb{E}[|\varepsilon_i|] \leq \delta$, that is, bounded first moment.

**Assumption 3.3.** The data fitting loss $\rho : \mathbb{R} \to \mathbb{R}$ is convex, continuously differentiable and its derivative $\psi$ is 1-Lipschitz and $|\psi(x)| \leq \delta$ for all $x \in \mathbb{R}$. The function $\phi_t$ is 1-Lipschitz and satisfies $\boldsymbol{\phi}_t(\boldsymbol{0}) = \boldsymbol{0}$. The matrices $\boldsymbol{S}_t = \sum_{i\in I_t} \boldsymbol{e}_i \boldsymbol{e}_i^\top$, and $|I_t| \geq c_0 n$ for some positive constant $c_0 \in (0, 1]$. Let $\eta_{\max} = \max_{t\in[T]} \eta_t$.

Huber loss and Psuedo-Huber loss all satisfy Assumption 3.3.

**Assumption 3.4.** The data fitting loss $\rho$ is twice continuously differentiable with positive second derivative.

**Assumption 3.5.** The sample size $n$ and feature dimension $p$ satisfy $p/n \leq \gamma$ for a constant $\gamma \in (0, \infty)$.

### 3.1 Intuition regarding the estimates of the generalization error

This subsection provides the intuition behind the definition of the estimates define below. For the sake of clarify, and in this subsection only, assume that

$$\boldsymbol{\Sigma} = \boldsymbol{I}_p, \qquad \boldsymbol{\varepsilon} = \boldsymbol{0}, \qquad \eta_t/|I_t| = 1/n. \tag{7}$$

With the above working assumptions, the validity of the estimates defined below relies on the probabilistic approximation

$$\|\widehat{\boldsymbol{b}}^t - \boldsymbol{b}^*\|^2 \approx \frac{1}{n}\sum_{i=1}^n \Big(-\boldsymbol{x}_i^\top(\widehat{\boldsymbol{b}}^t - \boldsymbol{b}^*) + \sum_{j=1}^p \boldsymbol{e}_j^\top \frac{\partial \widehat{\boldsymbol{b}}^t}{\partial x_{ij}}\Big)^2,$$

which was developed in [3] for risk estimation purposes, but outside the context of iterative algorithms. Above, $\boldsymbol{e}_j \in \mathbb{R}^p$ is the $j$-th canonical basis vector. In the present noiseless case with $\boldsymbol{\varepsilon} = \boldsymbol{0}$, the first term inside the squared norm in the right-hand side is equal to the residual $y_i - \boldsymbol{x}_i^\top \widehat{\boldsymbol{b}}^t$, so that the above display resembles (3). Taking this probabilistic approximation for granted, to study the second term in the right-hand side, we must understand the derivatives of $\widehat{\boldsymbol{b}}^t$ with respect to the entries $(x_{ij})_{i\in[n], j\in[p]}$ of $\boldsymbol{X}$. In (4), each iterate is a relatively simple function of the previous ones, with the simplifications (7) this is $\widehat{\boldsymbol{b}}^{t+1} = \boldsymbol{\phi}_t(\widehat{\boldsymbol{b}}^t + \boldsymbol{X}^\top \boldsymbol{S}_t \boldsymbol{\psi}(\boldsymbol{y} - \boldsymbol{X}\widehat{\boldsymbol{b}}^t)/n)$. For $t = 1$, given that $\widehat{\boldsymbol{b}}^1$ is a constant initialization, $\frac{\partial}{\partial x_{ij}}\widehat{\boldsymbol{b}}^2 = \widetilde{\boldsymbol{D}}_1 \boldsymbol{e}_j \boldsymbol{e}_i^\top \boldsymbol{S}_1 \boldsymbol{\psi}(\boldsymbol{y} - \boldsymbol{X}\widehat{\boldsymbol{b}}^1)/n - \widetilde{\boldsymbol{D}}_1 \boldsymbol{X}^\top \boldsymbol{S}_1 \boldsymbol{D}_1 \boldsymbol{e}_i(\widehat{\boldsymbol{b}}^1 - \boldsymbol{b}^*)_j/n$. We find in the proof, that when summing these quantities over $j \in [p]$, the second term involving

$(\widehat{\boldsymbol{b}}^1 - \boldsymbol{b}^*)_j$ is negligible, and the same negligibility holds at later iterations with terms involving $(\widehat{\boldsymbol{b}}^t - \boldsymbol{b}^*)_j$ (or any $(\widehat{\boldsymbol{b}}^s - \boldsymbol{b}^*)_j$, $s \leq t$). By performing a similar simple calculation at the next iteration, and ignoring these terms, we find with $f_i^1 = \boldsymbol{e}_i^\top \boldsymbol{S}_1 \psi(\boldsymbol{y} - \boldsymbol{X}\widehat{\boldsymbol{b}}^1)$ and $f_i^2 = \boldsymbol{e}_i^\top \boldsymbol{S}_2 \psi(\boldsymbol{y} - \boldsymbol{X}\widehat{\boldsymbol{b}}^2)$ by the chain rule

$$\sum_{j=1}^p \boldsymbol{e}_j^\top \frac{\partial \widehat{\boldsymbol{b}}^2}{\partial x_{ij}} \approx \underbrace{\mathrm{Tr}\Big[\frac{\widetilde{\boldsymbol{D}}_1}{n}\Big]}_{w_{2,1}} f_i^1, \quad \sum_{j=1}^p \boldsymbol{e}_j^\top \frac{\partial \widehat{\boldsymbol{b}}^3}{\partial x_{ij}} \approx \underbrace{\mathrm{Tr}\Big[\frac{\widetilde{\boldsymbol{D}}_2}{n}\Big]}_{w_{3,2}} f_i^2 + \underbrace{\mathrm{Tr}\Big[\widetilde{\boldsymbol{D}}_2(\boldsymbol{I}_p - \boldsymbol{X}^\top \boldsymbol{S}_2 \boldsymbol{D}_2 \boldsymbol{X}/n)\widetilde{\boldsymbol{D}}_1/n\Big]}_{w_{3,1}} f_i^1.$$

This reveals the weights $(w_{s,t})_{s<t}$ in (3) at iteration $t = 2$ and $t = 3$. We could continue this further by successive applications of the chain rule, although for later iterations this unrolling of the derivatives, capturing the interplay between the Jacobians $\boldsymbol{D}_t, \widetilde{\boldsymbol{D}}_t$ and the stochastic gradient matrix $\boldsymbol{S}_t$, becomes increasingly complex. This recursive unrolling of the derivatives can be performed numerically at the same time as the computation of the iterates. On the other hand, for the mathematical proof, for the formal definition of the weights in (3) and for the proposed estimates of the generalization error, the matrix notation defined in the next subsection exactly captures this unrolling of the derivatives.

### 3.2 Formal matrix notation to capture recursive derivatives

We now set up the matrix notation that captures this recursive unrolling of the derivatives by the chain rule. Throughout, $T$ is the final number of iterations. Define three block diagonal matrices $\mathcal{D} \in \mathbb{R}^{nT \times nT}$, $\widetilde{\mathcal{D}} \in \mathbb{R}^{pT \times pT}$, and $\mathcal{S} \in \mathbb{R}^{nT \times nT}$ by $\mathcal{D} = \sum_{t=1}^T \big((\boldsymbol{e}_t \boldsymbol{e}_t^\top) \otimes \boldsymbol{D}_t\big)$, $\widetilde{\mathcal{D}} = \sum_{t=1}^T \big((\boldsymbol{e}_t \boldsymbol{e}_t^\top) \otimes \widetilde{\boldsymbol{D}}_t\big)$, and $\mathcal{S} = \sum_{t=1}^T \big((\boldsymbol{e}_t \boldsymbol{e}_t^\top) \otimes \boldsymbol{S}_t\big)$. Now we are ready to introduce the following matrices of size $T \times T$:

$$\boldsymbol{W} = \sum_{j=1}^p (\boldsymbol{I}_T \otimes \boldsymbol{e}_j^\top)(\boldsymbol{I}_T \otimes \boldsymbol{\Sigma}^{1/2})\boldsymbol{\Gamma}(\boldsymbol{I}_T \otimes \boldsymbol{\Sigma}^{1/2})(\boldsymbol{I}_T \otimes \boldsymbol{e}_j), \tag{8}$$

$$\widehat{\boldsymbol{A}} = \sum_{i=1}^n (\boldsymbol{I}_T \otimes \boldsymbol{e}_i^\top)\mathcal{D}(\boldsymbol{I}_T \otimes \boldsymbol{X})\boldsymbol{\Gamma}(\boldsymbol{I}_T \otimes \boldsymbol{X}^\top)(\boldsymbol{I}_T \otimes \boldsymbol{e}_i), \tag{9}$$

$$\widehat{\boldsymbol{K}} = \sum_{t=1}^T \mathrm{Tr}(\boldsymbol{D}_t)\boldsymbol{e}_t \boldsymbol{e}_t^\top - \sum_{i=1}^n (\boldsymbol{I}_T \otimes \boldsymbol{e}_i^\top)\mathcal{D}(\boldsymbol{I}_T \otimes \boldsymbol{X})\boldsymbol{\Gamma}(\boldsymbol{I}_T \otimes \boldsymbol{X}^\top)\mathcal{S}\mathcal{D}(\boldsymbol{I}_T \otimes \boldsymbol{e}_i), \tag{10}$$

where $\boldsymbol{\Gamma} = \mathcal{M}^{-1}\boldsymbol{L}(\boldsymbol{\Lambda} \otimes \boldsymbol{I}_p)\widetilde{\mathcal{D}} \in \mathbb{R}^{pT \times pT}$, $\boldsymbol{L} = \sum_{t=2}^T \big((\boldsymbol{e}_t \boldsymbol{e}_{t-1}^\top) \otimes \boldsymbol{I}_p\big)$, $\boldsymbol{\Lambda} = \sum_{t=1}^T \frac{\eta_t}{|I_t|}\boldsymbol{e}_t \boldsymbol{e}_t^\top$,

$$\mathcal{M} = \begin{bmatrix} \boldsymbol{I}_p & & & \\ -\boldsymbol{P}_1 & \boldsymbol{I}_p & & \\ & \ddots & \ddots & \\ & & -\boldsymbol{P}_{T-1} & \boldsymbol{I}_p \end{bmatrix} \qquad \text{and} \qquad \boldsymbol{P}_t = \widetilde{\boldsymbol{D}}_t\Big(\boldsymbol{I}_p - \frac{\eta_t}{|I_t|}\boldsymbol{X}^\top \boldsymbol{S}_t \boldsymbol{D}_t \boldsymbol{X}\Big).$$

Although notationally involved, the purpose of these matrices is just to formalize the recursive computation of the derivatives by the chain rule mentioned in Section 3.1.

### 3.3 Main results: estimating the generalization error consistently

For each iterate $\widehat{\boldsymbol{b}}^t$, define the target $r_t$ (generalization error) and its estimate $\hat{r}_t$ by

$$r_t \stackrel{\text{def}}{=} \|\boldsymbol{\Sigma}^{1/2}(\widehat{\boldsymbol{b}}^t - \boldsymbol{b}^*)\|^2 + \frac{\|\boldsymbol{\varepsilon}\|^2}{n}, \qquad \hat{r}_t = \frac{1}{n}\Big\|(\boldsymbol{y} - \boldsymbol{X}\widehat{\boldsymbol{b}}^t) + \sum_{s=1}^{t-1} w_{t,s}\boldsymbol{S}_s \psi(\boldsymbol{y} - \boldsymbol{X}\widehat{\boldsymbol{b}}^s)\Big\|^2, \tag{11}$$

where $w_{t,s} := \boldsymbol{e}_t^\top \boldsymbol{W} \boldsymbol{e}_s$ and $\boldsymbol{W} \in \mathbb{R}^{T \times T}$ is the matrix defined in (8). The following shows that $|\hat{r}_t - r_t| \to^P 0$ (convergence to 0 in probability) under suitable assumptions.

**Theorem 3.6** (Proved in Appendix C.1). *Let Assumptions 3.1, 3.3 and 3.5 be fulfilled. Then $\forall \epsilon > 0$,*

$$\mathbb{P}\Big(|\hat{r}_t - r_t| > \epsilon\Big) \leq \max\Big\{1, \frac{C(T, \gamma, \eta_{\max}, c_0, \delta, \kappa)}{\epsilon}\Big\}\Big(\frac{1}{\sqrt{n}} + \mathbb{E}\Big[\min\Big\{1, \frac{\|\boldsymbol{\varepsilon}\|}{n}\Big\}\Big]\Big). \tag{12}$$

*If additionally Assumption 3.2 holds then $\mathbb{E}[\min\{1, \frac{\|\boldsymbol{\varepsilon}\|}{n}\}] \to 0$, so that, as $n, p \to +\infty$ while $(T, \gamma, \eta_{\max}, c_0, \delta, \kappa, \epsilon)$ are held fixed, the right-hand side converges to 0 and $\hat{r}_t - r_t$ converges to 0 in probability.*

This establishes that $\hat{r}_t$ is consistent at estimating $r_t$. The statement $\mathbb{E}[\min\{1, \|\boldsymbol{\varepsilon}\|/n\}] \to 0$ is equivalent to $\|\boldsymbol{\varepsilon}\|^2/n^2 \to^P 0$ (convergence in probability), and is proved in [22] under the assumption that $\mathbb{E}|\varepsilon_i| < +\infty$ with $\varepsilon_i$ i.i.d. from a fixed distribution; this allows $\mathrm{Var}[\varepsilon_i] = +\infty$ as long as the first moment is finite. The expression of $\boldsymbol{W}$ involves $\boldsymbol{\Sigma}$, which is typically unknown in practice. Our next result provides a consistent estimate of $\boldsymbol{W}$ using quantities that do not require the knowledge of $\boldsymbol{\Sigma}$.

We propose to estimate $\boldsymbol{W}$ by $\widetilde{\boldsymbol{W}} \stackrel{\text{def}}{=} \widehat{\boldsymbol{K}}^{-1}\widehat{\boldsymbol{A}}$ where $\widehat{\boldsymbol{K}}$ and $\widehat{\boldsymbol{A}}$ are the $T \times T$ matrices defined in (9)-(10). We define another estimate $\tilde{r}_t$ by replacing $\boldsymbol{W}$ in (11) with $\widetilde{\boldsymbol{W}} = \widehat{\boldsymbol{K}}^{-1}\widehat{\boldsymbol{A}}$:

$$\tilde{r}_t = \frac{1}{n}\left\|(\boldsymbol{y} - \boldsymbol{X}\widehat{\boldsymbol{b}}^t) + \sum_{s=1}^{t-1} \tilde{w}_{t,s} \boldsymbol{S}_s \boldsymbol{\psi}(\boldsymbol{y} - \boldsymbol{X}\widehat{\boldsymbol{b}}^s)\right\|^2,$$

where $\tilde{w}_{t,s} = \boldsymbol{e}_t^\top \widetilde{\boldsymbol{W}} \boldsymbol{e}_s$.

**Theorem 3.7** (Proved in Appendix C.3). *Under Assumptions 3.1 and 3.3 to 3.5, for any $\epsilon > 0$,*

$$\mathbb{P}\left(|\tilde{r}_t - r_t| > \epsilon\right) \leq 2e^{-n/18} + \max\left\{1, \frac{C(T, \gamma, \eta_{\max}, c_0, \delta, \kappa)}{\epsilon}\right\}\left[\frac{1}{\sqrt{n}} + \mathbb{E}[\min(1, \frac{\|\boldsymbol{\varepsilon}\|}{n})]\right].$$

*If additionally Assumption 3.2 holds then $\mathbb{E}[\min\{1, \frac{\|\boldsymbol{\varepsilon}\|}{n}\}] \to 0$, so that, as $n, p \to +\infty$ while $(T, \gamma, \eta_{\max}, c_0, \delta, \kappa, \epsilon)$ are held fixed, the right-hand side converges to 0 and $\tilde{r}_t - r_t$ converges to 0 in probability.*

This establishes the consistency of $\tilde{r}_t$. The simulations presented next confirm that the two estimates $\tilde{r}_t$ and $\hat{r}_t$ both are accurate estimates of $r_t$. The estimate $\tilde{r}_t$ has the advantage of not relying on the knowledge of $\boldsymbol{\Sigma}$ and are recommended in practice.

*Remark* 3.8. We highlight that directly generalizing the approach in [5] would lead to the approximation $\widetilde{\boldsymbol{A}} \approx \widetilde{\boldsymbol{K}}\boldsymbol{W}$, where $\widetilde{\boldsymbol{A}}$ and $\widetilde{\boldsymbol{K}}$ are given in (23) and (26), respectively. From $\widetilde{\boldsymbol{A}} \approx \widetilde{\boldsymbol{K}}\boldsymbol{W}$, obtaining an estimate of $\boldsymbol{W}$ requires inverting $\widetilde{\boldsymbol{K}}$. However, this inversion fails for SGD for small (but still very realistic) batch sizes of order $0.1n$ in simulations (see Figure 4). The matrix $\widetilde{\boldsymbol{K}}$ is lower triangular, and the reason for the lack of invertibility of $\widetilde{\boldsymbol{K}}$ can be seen in the diagonal terms equal to $\mathrm{Tr}[\boldsymbol{S}_t \boldsymbol{D}_t]$ in (26), where $\boldsymbol{S}_t \in \{0,1\}^{n \times n}$ is the diagonal matrix with 1 in position $(i, i)$ if and only if the $i$-th observation is used in the $t$-th batch. This diagonal element of $\widetilde{\boldsymbol{K}}$ can easily be small (or even 0) for small batches, if the batch only contains observations such that $(\boldsymbol{D}_t)_{ii}$ is 0 or small. Let $\tilde{r}_t^{\mathrm{sub}}$ denote the estimate of the same form as $\tilde{r}_t$ but using the weight matrix $\widetilde{\boldsymbol{K}}^{-1}\widetilde{\boldsymbol{A}}$ instead. Simulation results in Figure 4 confirm that $\tilde{r}_t^{\mathrm{sub}}$ is suboptimal compared to our proposed $\tilde{r}_t$. For SGD and proximal SGD, we solved this issue regarding the invertibility of $\widetilde{\boldsymbol{K}}$ by using out-of-batch samples in the construction of $\widehat{\boldsymbol{K}}$ and $\widehat{\boldsymbol{A}}$, in order to avoid $\boldsymbol{S}_t$ in the diagonal elements of $\widehat{\boldsymbol{K}}$ in equation (31). This is the key to making these estimators work for SGD and proximal SGD, and this use of out-of-batch samples is new compared to [5] (which only tackles the square loss with full-batch gradients).

*Remark* 3.9. The constant $C(T, \gamma, \eta_{\max}, c_0, \delta, \kappa)$ in the above results is not explicit. Inspection of the proof reveals that the dependence of this constant in $T$ is currently $T^T$, allowing $T$ of order $\log(n)/\log\log n$ before the bound becomes vacuous. Improving this dependence in $T$ appears challenging and possibly out of reach of current tools, even for the well-studied Approximate Message Passing (AMP) algorithms. The papers [25, 24] feature for instance the same $\log(n)/\log\log n$ dependence for approximating the risk of AMP. The preprint [15] offers the latest advances on the dependence on $T$ in the bounds satisfied by AMP. It allows $T \asymp \mathrm{poly}(n)$ while still controlling certain AMP related quantities, although for the risk [15, equations (16)-(17)] the condition required on $T$ is still logarithmic in $n$. This suggests that advances on this front are possible, at least for isotropic design and specific loss and regularizer such as those studied in [15]: Lasso or Robust M-estimation with no regularizer. Since these latest advances in [15] are obtained for specific estimates (Lasso or Robust M-estimation with no regularizer), it may be possible to follow a similar strategy and improve our bounds for specific examples of iterative algorithms closer to AMP, or algorithms featuring only separable losses and penalty. We leave such improvements for specific examples for future work, as the goal of the current paper is to cover a general framework allowing iterations of the form (4) with little restriction on the nonlinear functions except being Lipschitz.

# 4 Simulation

In this section, we present numerical experiments to assess the performance of the proposed risk estimates. All necessary code for reproducing these experiments is provided in the supplementary material and is publicly available in the GitHub repository `https://github.com/kaitan365/SGD-generlization-errors`. Our goal is to compare the performance of the proposed risk estimates with the true risk $r_t$ for different regression methods and iterative algorithms.

We generate the dataset $(\boldsymbol{X}, \boldsymbol{y})$ from the linear model (1), that is, $\boldsymbol{y} = \boldsymbol{X}\boldsymbol{b}^* + \boldsymbol{\varepsilon}$. Here, the rows of $\boldsymbol{X} \in \mathbb{R}^{n \times p}$ are sampled from a centered multivariate normal distribution with covariance matrix $\boldsymbol{\Sigma} = \boldsymbol{I}_p$. The noise vector $\boldsymbol{\varepsilon}$ consists of i.i.d. entries drawn from a $t$ distribution with two degrees of freedom so that the noise variance is infinite. The true regression vector $\boldsymbol{b}^* \in \mathbb{R}^p$ is chosen with $p/20$ nonzero entries, set to a constant value such that the signal strength $\|\boldsymbol{b}^*\|^2$ equals 10.

We explore two scenarios of the $(n, p)$ pairs and corresponding iterative algorithms:

(i) $(n, p) = (10000, 5000)$: In this configuration, with $n$ much larger than $p$, we examine Huber regression and Pseudo-Huber regression (without penalty or soft-thresholding). Both the gradient descent (GD) and stochastic gradient descent (SGD) algorithms are implemented for each type of regression.

(ii) $(n, p) = (10000, 12000)$: Here, we investigate Huber regression and Pseudo-Huber regression with an L1 penalty, $\lambda\|\boldsymbol{b}\|_1$ ($\lambda = 0.002$) and corresponding soft-thresholding step. For each penalized regression, we employ the Proximal Gradient Descent (Proximal GD) and Stochastic Proximal Gradient Descent (Proximal SGD) algorithms.

In all algorithms, we start with the initial vector $\widehat{\boldsymbol{b}}^1 = \boldsymbol{0}_p$ and proceed with a fixed step size $\eta = (1 + \sqrt{p/n_*})^{-2}$ where $n_* = n$ for GD and proximal GD, and $n_* = n/5$ for SGD and proximal SGD. We run each algorithm for $T = 100$ steps. For SGD and Proximal SGD, batches $I_t \subset \{1, 2, \ldots, n\}$ are randomly sampled without replacement and independently of $(\boldsymbol{X}, \boldsymbol{y}, (I_s)_{s \neq t})$, each with cardinality $|I_t| = \frac{n}{5}$.

A crucial component of the proposed risk estimates $\hat{r}_t$ and $\tilde{r}_t$ involve the weight matrices $\boldsymbol{W}$ and $\widetilde{\boldsymbol{W}}$. The matrix $\boldsymbol{W}$ is defined in Theorem 3.6, and $\widetilde{\boldsymbol{W}} = \widehat{\boldsymbol{K}}^{-1}\widehat{\boldsymbol{A}}$ is defined in Theorem 3.7. We employ Hutchinson's trace approximation to compute $\boldsymbol{W}$, $\widehat{\boldsymbol{A}}$, and $\widehat{\boldsymbol{K}}$. This implementation is computationally efficient. We refer readers to [5, Section 4] for more details.

Recall that we have proposed two estimates for $r_t = \|\boldsymbol{\Sigma}^{1/2}(\widehat{\boldsymbol{b}}^t - \boldsymbol{b}^*)\|^2 + \|\boldsymbol{\varepsilon}\|^2/n$, one is $\hat{r}_t$ in Theorem 3.6 which requires knowing $\boldsymbol{\Sigma} = \mathbb{E}[\boldsymbol{x}_i\boldsymbol{x}_i^\top]$, and the other is $\tilde{r}_t$ in Theorem 3.7 which does not need $\boldsymbol{\Sigma}$. Since the quantity $\|\boldsymbol{\varepsilon}\|^2/n$ remains constant along the algorithm trajectory, we only focus on the estimation of $\|\boldsymbol{\Sigma}^{1/2}(\widehat{\boldsymbol{b}}^t - \boldsymbol{b}^*)\|^2$. We repeat each numerical experiment 100 times and present the aggregated results in Figures 1, 2 and 3.

In Figure 1, we focus on the scenario with $(n, p) = (10000, 5000)$, and plot the actual risk $\|\boldsymbol{\Sigma}^{1/2}(\widehat{\boldsymbol{b}}^t - \boldsymbol{b}^*)\|^2$, and its two estimates $\hat{r}_t - \|\boldsymbol{\varepsilon}\|^2/n$ and $\tilde{r}_t - \|\boldsymbol{\varepsilon}\|^2/n$ along with the 2 standard error bar for GD and SGD algorithms applied to both Huber and Pseudo Huber regression. In Figure 2, we focus on the scenario with $(n, p) = (10000, 12000)$, and present the risk curves for the Proximal GD and Proximal SGD algorithms applied to both L1-penalized Huber regression and Pseudo-Huber regression.

Figure 1 and Figure 2 confirm the three curves are in close agreement, indicating that the proposed estimates $\hat{r}_t - \|\boldsymbol{\varepsilon}\|^2/n$ and $\tilde{r}_t - \|\boldsymbol{\varepsilon}\|^2/n$ are consistent estimates of the actual risk $\|\boldsymbol{\Sigma}^{1/2}(\widehat{\boldsymbol{b}}^t - \boldsymbol{b}^*)\|^2$. The two estimates closely capture the risk $\|\boldsymbol{\Sigma}^{1/2}(\widehat{\boldsymbol{b}}^t - \boldsymbol{b}^*)\|^2$ over the entire trajectory of the algorithms. For GD and Proximal GD, the risk curves exhibit a U-shape, first decreasing and then increasing, and the estimates $\hat{r}_t$ and $\tilde{r}_t$ closely capture this pattern. This suggests that the proposed estimates are reliable and can be used to monitor the risk of the iterates and find the optimal iteration (the iteration minimizing the generalization error) along the trajectory of the algorithm.

**Additional experiments: varying step sizes for different iterations.** We also conduct simulations to investigate the accuracy of the proposed risk estimates in a setting with varying step size. We consider two types of step sizes: 1). $\eta_t = 1$ if $t$ is odd, and $\eta_t = 0$ if $t$ is even; 2). $\eta_t = 1$ if $t$ is odd,

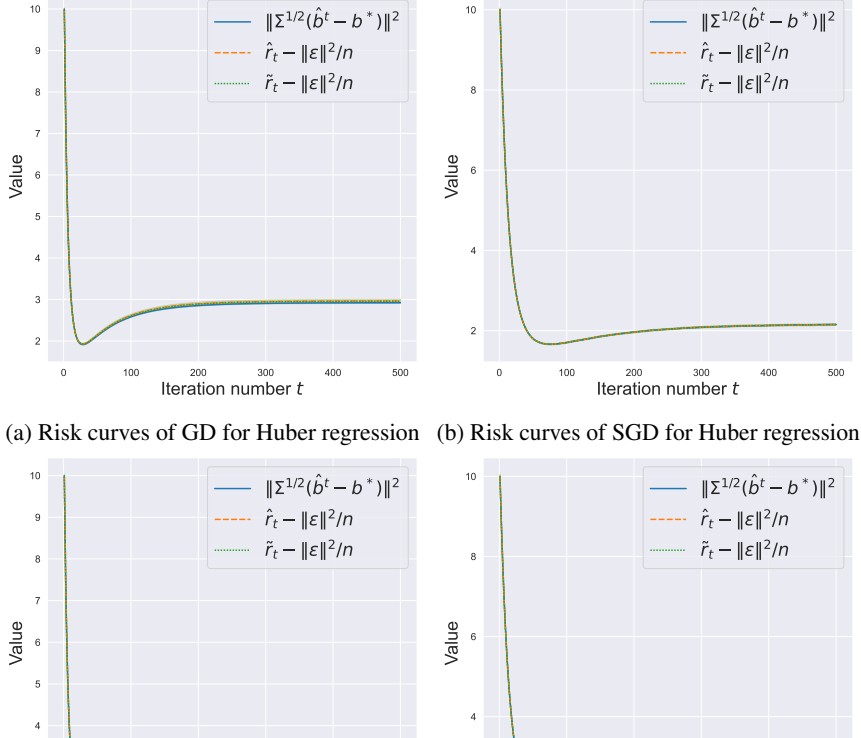

(a) Risk curves of GD for Huber regression

(b) Risk curves of SGD for Huber regression

(c) Risk curves of GD for Pseudo-Huber regression

(d) Risk curves of SGD for Pseudo-Huber regression

Figure 1: Risk curves for Huber and Pseudo-Huber regression with GD and SGD algorithms for the scenario $(n, p) = (10000, 5000)$. **Upper row:** Huber regression, **Lower row:** Pseudo-Huber regression. **Left column:** GD, **Right column:** SGD.

and $\eta_t = 0.5$ if $t$ is even. While the above choices of step size are not preferred in practice, here the goal is show that the proposed risk estimates is able to accurately capture the dynamics of the risk even when the step size changes along the trajectory of the algorithm. For instance, the first choice of step size should produce a risk curve that is flat when $t$ is even. The results are presented in Figure 3, illustrating that the risk estimates accurately capture the flat segments of the true risk curve.

**Additional experiments: the estimate $\tilde{r}_t^{\mathrm{sub}}$ is suboptimal.** We compare the performance of $\tilde{r}_t^{\mathrm{sub}}$ with our proposed estimates in Huber regression with $(n, p, T) = (4000, 1000, 20)$ and batch size $|I_t| = n/10$ and $\eta_t = 0.2$ for all $t \in [T]$. It is clear from Figure 4 that $\tilde{r}_t$ is more accurate than the suboptimal estimator $\tilde{r}_t^{\mathrm{sub}}$, especially when $t$ increases.

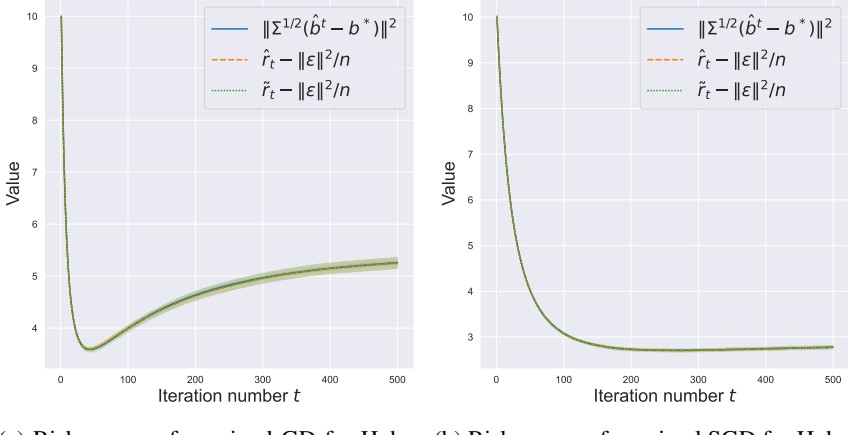

(a) Risk curves of proximal GD for Huber regression

(b) Risk curves of proximal SGD for Huber regression

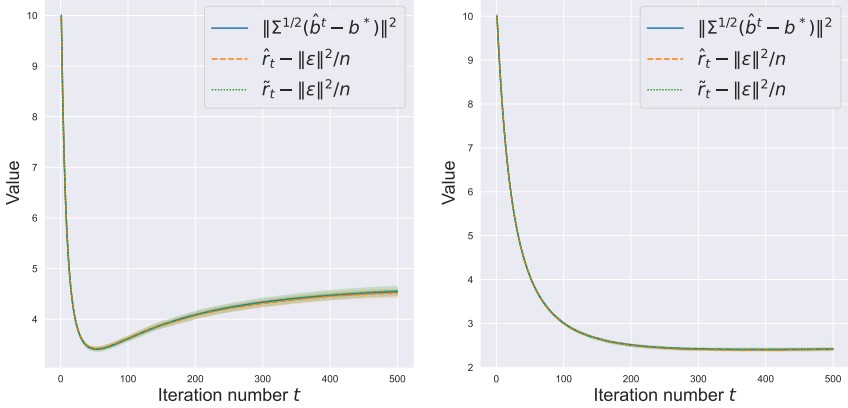

(c) Proximal GD for Pseudo-Huber regression

(d) Proximal SGD for Pseudo-Huber regression

Figure 2: Risk curves for L1-penalized Huber and Pseudo-Huber regression with Proximal GD and Proximal SGD algorithms for the scenario $(n, p) = (10000, 12000)$. **Upper row:** L1-penalized Huber regression, **Lower row:** L1-penalized Pseudo-Huber regression. **Left column:** Proximal GD, **Right column:** Proximal SGD.

## 5    Discussion

This paper proposes a novel risk estimate for the generalization error of iterates generated by the proximal GD and proximal SGD algorithms in robust regression. The proposed risk estimates accurately capture the predictive risk of the iterates along the trajectory of the algorithms, and are provably consistent (Theorems 3.6 and 3.7). Three matrices in $\mathbb{R}^{T \times T}$ in (8)-(10) reveal the interplay between the squared risk, the residuals and the gradients, so that the approximation (3) holds. This structure is different from the square loss case studied in [5] where only two matrices (inverse of each other) are sufficient.

Let us mention some open questions along with potential future research directions. The first question regards the probabilistic model: we currently assume Gaussian features $\boldsymbol{x}_i$, and it would be of interest to study the extension in which our consistency results are universal, allowing non-Gaussian feature distributions. Second, it is of interest to extend the current estimates to more general optimization problems of the form (2) with non-smooth data-fitting loss, for instance the Least Absolute Deviation loss $\|\boldsymbol{y} - \boldsymbol{X}\boldsymbol{b}\|_1$. In this case the gradient does not exist at the origin, which calls for different algorithms than the GD and SGD variants presented here, for instance the Alternating Direction Method of Multipliers (ADMM) [6]. It is of independent interest to derive the risk estimates for iterates obtained by such primal-dual methods.

## Acknowledgments

P. C. Bellec acknowledges partial support from the NSF Grants DMS-1945428 and DMS-2413679. The authors acknowledge the Office of Advanced Research Computing (OARC) at Rutgers, The State University of New Jersey for providing access to the Amarel cluster and associated research computing resources that have contributed to the results reported here. The authors thank the anonymous reviewers for their valuable comments and suggestions that helped improve the presentation of the paper.

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

# Supplementary Material of "Estimating Generalization Performance Along the Trajectory of Proximal SGD in Robust Regression"

## A  Additional simulation results

The following figures illustrate the proposed risk estimates accurately estimate the trajectory of the risk even when the step size changes at every step.

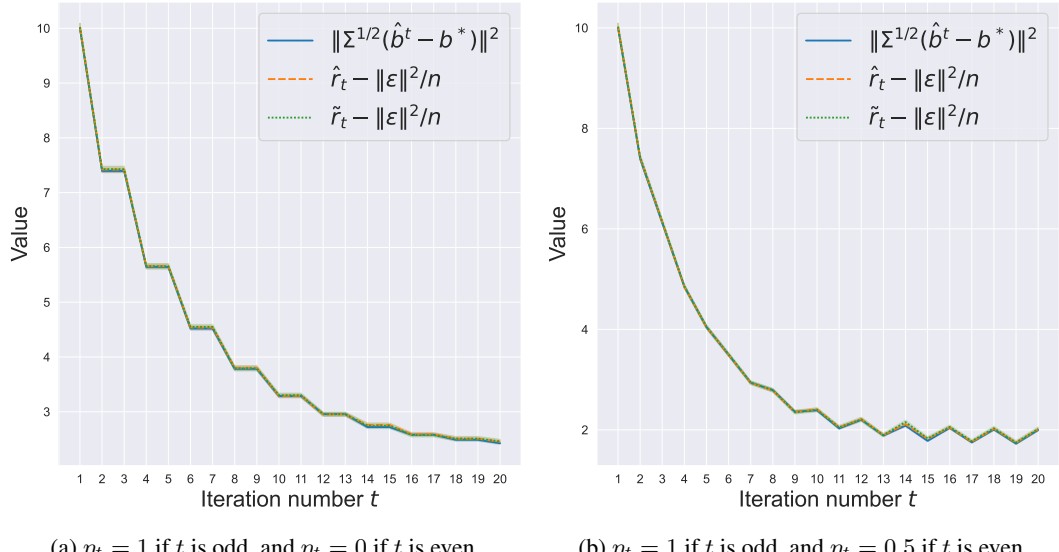

(a) $\eta_t = 1$ if $t$ is odd, and $\eta_t = 0$ if $t$ is even.  (b) $\eta_t = 1$ if $t$ is odd, and $\eta_t = 0.5$ if $t$ is even.

Figure 3: Risk curves for SGD applied to Huber regression with $(n, p) = (3000, 1000)$ using different choices of step sizes. **Left panel:** $\eta_t = 1$ if $t$ is odd, and $\eta_t = 0$ if $t$ is even. **Right panel:** $\eta_t = 1$ if $t$ is odd, and $\eta_t = 0.5$ if $t$ is even.

Figure 4 compares the performance of the proposed estimators $\hat{r}_t$, $\tilde{r}_t$ and the estimator $\tilde{r}_t^{\mathrm{sub}}$ generalized directly from [5]. It confirms that our proposed estimators outperforms $\tilde{r}_t^{\mathrm{sub}}$.

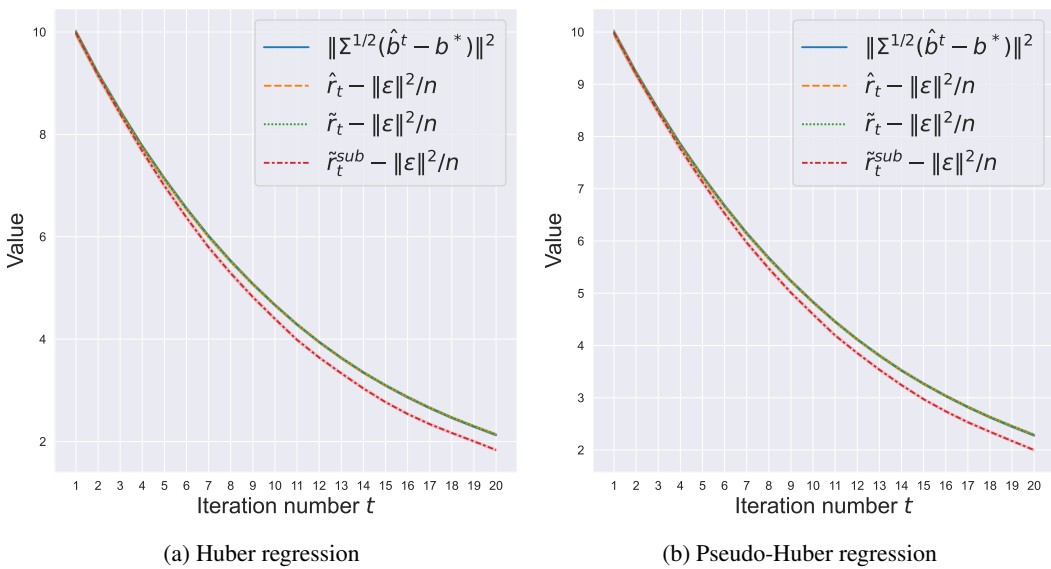

(a) Huber regression  (b) Pseudo-Huber regression

Figure 4: Risk curves for SGD applied to Huber and pseudo-Huber regression with $(n, p, T) = (4000, 1000, 20)$, $|I_t| = n/10$ and $\eta_t = 0.2$ for all $t$.

## B  Auxiliary Results

Throughout, we define

$$\boldsymbol{E} = [\boldsymbol{\varepsilon}, ..., \boldsymbol{\varepsilon}] \in \mathbb{R}^{n \times T}, \qquad\qquad \boldsymbol{F} = [\boldsymbol{S}_1 \psi(\boldsymbol{y} - \boldsymbol{X}\widehat{\boldsymbol{b}}^1), ..., \boldsymbol{S}_T \psi(\boldsymbol{y} - \boldsymbol{X}\widehat{\boldsymbol{b}}^T)] \in \mathbb{R}^{n \times T}, \quad (13)$$

$$\boldsymbol{H} = \boldsymbol{\Sigma}^{1/2}[\widehat{\boldsymbol{b}}^1 - \boldsymbol{b}^*, ..., \widehat{\boldsymbol{b}}^T - \boldsymbol{b}^*] \in \mathbb{R}^{p \times T}, \qquad \boldsymbol{R} = [\boldsymbol{y} - \boldsymbol{X}\widehat{\boldsymbol{b}}^1, ..., \boldsymbol{y} - \boldsymbol{X}\widehat{\boldsymbol{b}}^T] \in \mathbb{R}^{n \times T}. \quad (14)$$

Note that in the above $\boldsymbol{E}, \boldsymbol{H}$ are not observable since $\boldsymbol{\varepsilon}$ and $\boldsymbol{b}^*$ are unknown. However, $\boldsymbol{F}$ and $\boldsymbol{R}$ are observable and can be easily computed once the iterates $(\widehat{\boldsymbol{b}}^t)_{t \in [T]}$ are calculated.

### B.1  Change of variables

In this section, we conduct the change of variable to simplify the proof. Specifically, we view the linear model $\boldsymbol{y} = \boldsymbol{X}\boldsymbol{b}^* + \boldsymbol{\varepsilon}$ as a model with design matrix $\boldsymbol{G}$ and the regression vector $\boldsymbol{\theta}^*$, i.e.

$$\boldsymbol{y} = \boldsymbol{X}\boldsymbol{b}^* + \boldsymbol{\varepsilon} = \underbrace{\boldsymbol{X}\boldsymbol{\Sigma}^{-1/2}}_{\boldsymbol{G}} \underbrace{\boldsymbol{\Sigma}^{1/2}\boldsymbol{b}^*}_{\boldsymbol{\theta}^*} + \boldsymbol{\varepsilon}.$$

This way, the design matrix $\boldsymbol{G}$ has i.i.d. entries from standard normal distribution. Using the same argument in [5, Appendix D], we can show that the matrices $\boldsymbol{H}, \boldsymbol{F}, \widehat{\boldsymbol{A}}, \widehat{\boldsymbol{K}}$ remains the same under the change of variable. Therefore, we can prove the main results using the model with design matrix $\boldsymbol{G}$ and the regression vector $\boldsymbol{\theta}^*$. In other words, we assume without of loss of generality that the design matrix $\boldsymbol{X}$ has i.i.d. $N(0, 1)$, or equivalently that the independent rows of $\boldsymbol{X}$ are normally distributed with covariance $\boldsymbol{\Sigma} = \boldsymbol{I}_p$. We prove the main results using $\boldsymbol{\Sigma} = \boldsymbol{I}_p$, and the results for general $\boldsymbol{\Sigma}$ follow by this change of variable with the constant $C(T, \gamma, \eta_{\max}, c_0, \delta)$ appearing in the bounds depending additionally on $\kappa$ (the upper bound of the condition number of $\boldsymbol{\Sigma}$ from Assumption 3.1).

### B.2  Derivative formulae

In this section, we present derivative formulae that will be useful in later proofs. The following formulas differ from [5] due to the use of robust loss functions and the application of SGD with random batches at each iteration. The formulae are also significantly more complex than in the case of regularized M-estimators [3, 4].

**Lemma B.1** (Proved in Appendix D.1). *Let* $(\widehat{\boldsymbol{b}}^t)_{t \in [T]}$ *be the iterates generated from the recursion* (4) *and the initial value* $\widehat{\boldsymbol{b}}^1$ *is independent of* $\boldsymbol{X}$. *Then the derivative of* $\widehat{\boldsymbol{b}}^t$ *with respect to* $\boldsymbol{X}$ *is given by*

$$\frac{\partial \widehat{\boldsymbol{b}}^t}{\partial x_{ij}} = (\boldsymbol{e}_t^\top \otimes \boldsymbol{I}_p)\boldsymbol{\Gamma}\Big[((\boldsymbol{F}^\top \boldsymbol{e}_i) \otimes \boldsymbol{e}_j) - (\boldsymbol{I}_T \otimes \boldsymbol{X}^\top)\mathcal{SD}((\boldsymbol{H}^\top \boldsymbol{e}_j) \otimes \boldsymbol{e}_i)\Big], \qquad (15)$$

*where* $\boldsymbol{\Gamma} = \mathcal{M}^{-1}\boldsymbol{L}(\boldsymbol{\Lambda} \otimes \boldsymbol{I}_p)\widetilde{\mathcal{D}}$, $\boldsymbol{L} = \sum_{t=2}^T \big((\boldsymbol{e}_t \boldsymbol{e}_{t-1}^\top) \otimes \boldsymbol{I}_p\big)$, $\boldsymbol{\Lambda} = \sum_{t=1}^T \frac{\eta_t}{|I_t|}\boldsymbol{e}_t \boldsymbol{e}_t^\top$, *and*

$$\mathcal{M} = \begin{bmatrix} \boldsymbol{I}_p & & & \\ -\boldsymbol{P}_1 & \boldsymbol{I}_p & & \\ & \ddots & \ddots & \\ & & -\boldsymbol{P}_{T-1} & \boldsymbol{I}_p \end{bmatrix} \qquad \text{where} \quad \boldsymbol{P}_t = \widetilde{\boldsymbol{D}}_t(\boldsymbol{I}_p - \tfrac{\eta_t}{|I_t|}\boldsymbol{X}^\top \boldsymbol{S}_t \boldsymbol{D}_t \boldsymbol{X}).$$

Recall $\boldsymbol{F} = [\boldsymbol{S}_1 \psi(\boldsymbol{y} - \boldsymbol{X}\widehat{\boldsymbol{b}}^1), ..., \boldsymbol{S}_T \psi(\boldsymbol{y} - \boldsymbol{X}\widehat{\boldsymbol{b}}^T)]$, we have $F_{lt} = \boldsymbol{e}_l^\top \boldsymbol{S}_t \psi(\boldsymbol{y} - \boldsymbol{X}\widehat{\boldsymbol{b}}^t)$. The following two corollaries are a direct consequence of Lemma B.1.

**Lemma B.2** (Proved in Appendix D.2 ). *Under the same conditions of Lemma B.1. Let* $F_{lt} = \boldsymbol{e}_l^\top \boldsymbol{F}\boldsymbol{e}_t = \boldsymbol{e}_l^\top \boldsymbol{S}_t \psi(\boldsymbol{y} - \boldsymbol{X}\widehat{\boldsymbol{b}}^t)$, *we have*

$$\frac{\partial F_{lt}}{\partial x_{ij}} = D_{ij}^{lt} + \Delta_{ij}^{lt}, \qquad (16)$$

*where*

$$D_{ij}^{lt} = -\boldsymbol{e}_l^\top \boldsymbol{S}_t \boldsymbol{D}_t \boldsymbol{e}_i \boldsymbol{e}_j^\top \boldsymbol{H}\boldsymbol{e}_t + ((\boldsymbol{e}_j^\top \boldsymbol{H}) \otimes \boldsymbol{e}_i^\top)\mathcal{DS}(\boldsymbol{I}_T \otimes \boldsymbol{X})\boldsymbol{\Gamma}^\top(\boldsymbol{I}_T \otimes \boldsymbol{X}^\top)\mathcal{SD}(\boldsymbol{e}_t \otimes \boldsymbol{e}_l),$$

$$\Delta_{ij}^{lt} = -((\boldsymbol{e}_i^\top \boldsymbol{F}) \otimes \boldsymbol{e}_j^\top)\boldsymbol{\Gamma}^\top(\boldsymbol{I}_T \otimes \boldsymbol{X}^\top)\mathcal{SD}(\boldsymbol{e}_t \otimes \boldsymbol{e}_l).$$

**Lemma B.3** (Proved in Appendix D.3). *Let $\widetilde{F} = [\psi(y - X\widehat{b}^1), ..., \psi(y - X\widehat{b}^T)]$ and $\widetilde{F}_{l,t} = e_l^\top \widetilde{F} e_t$. Under the same conditions of Lemma B.1. We have*

$$\frac{\partial \widetilde{F}_{l,t}}{\partial x_{ij}} = \tilde{D}_{ij}^{lt} + \tilde{\Delta}_{ij}^{lt}, \tag{17}$$

*where*

$$\tilde{D}_{ij}^{lt} = -e_l^\top D_t e_i e_j^\top H e_t + ((e_j^\top H) \otimes e_i^\top) \mathcal{D}\mathcal{S}(I_T \otimes X)\Gamma^\top(I_T \otimes X^\top)\mathcal{D}(e_t \otimes e_l),$$
$$\tilde{\Delta}_{ij}^{lt} = -((e_i^\top F) \otimes e_j^\top)\Gamma^\top(I_T \otimes X^\top)\mathcal{D}(e_t \otimes e_l).$$

**Definition B.4.** Define the matrices $\Upsilon_1 \in \mathbb{R}^{p \times T}$, $\Upsilon_2 \in \mathbb{R}^{n \times T}$, $\Upsilon_3 \in \mathbb{R}^{T \times T}$, $\Upsilon_4 \in \mathbb{R}^{T \times T}$, $\Upsilon_5 \in \mathbb{R}^{T \times T}$ by the identities

$$\forall j \in [p], \quad \sum_{i=1}^n \frac{\partial e_i^\top F}{\partial x_{ij}} = -e_j^\top H \widetilde{K}^\top - e_j^\top \Upsilon_1, \tag{18}$$

$$\forall i \in [n], \quad \sum_{j=1}^p \frac{\partial e_j^\top H}{\partial x_{ij}} = e_i^\top F W^\top - e_i^\top \Upsilon_2, \tag{19}$$

$$\sum_{i=1}^n \sum_{j=1}^p \frac{\partial F^\top e_i e_j^\top H}{\partial x_{ij}} = -\widetilde{K} H^\top H + F^\top F W^\top - \Upsilon_3, \tag{20}$$

$$\sum_{i=1}^n \sum_{j=1}^p \frac{\partial H^\top X^\top e_i e_j^\top H}{\partial x_{ij}} = (nI_T - \widetilde{A})H^\top H + H^\top X^\top F W^\top - \Upsilon_4, \tag{21}$$

$$\sum_{i=1}^n \sum_{j=1}^p \frac{\partial F^\top e_i e_j^\top X^\top \widetilde{F}}{\partial x_{ij}} = -\widetilde{K} H^\top X^\top \widetilde{F} + p F^\top \widetilde{F} - F^\top F \widehat{A}^\top - \Upsilon_5, \tag{22}$$

*where the matrices* $\widetilde{K}, \widehat{A}, \widetilde{A}, W$ *are defined as follows*

$$\widetilde{A} = \sum_{i=1}^n (I_T \otimes e_i^\top)(I_T \otimes X)\Gamma(I_T \otimes X^\top)\mathcal{S}\mathcal{D}(I_T \otimes e_i), \tag{23}$$

$$\widehat{A} = \sum_{i=1}^n (I_T \otimes e_i^\top)\mathcal{D}(I_T \otimes X)\Gamma(I_T \otimes X^\top)(I_T \otimes e_i), \tag{24}$$

$$W = \sum_{j=1}^p (I_T \otimes e_j^\top)\Gamma(I_T \otimes e_j), \tag{25}$$

$$\widetilde{K} = \sum_{t=1}^T \text{Tr}(S_t D_t)e_t e_t^\top - \sum_{i=1}^n (I_T \otimes e_i^\top)\mathcal{S}\mathcal{D}(I_T \otimes X)\Gamma(I_T \otimes X^\top)\mathcal{S}\mathcal{D}(I_T \otimes e_i). \tag{26}$$

The matrices $\Upsilon_1, \Upsilon_2, \dots$ are negligible in the sense that their Frobenius norms are of smaller orders compared to their preceding terms in (18)–(22). We provide the bounds in next lemma, which is obtained by deriving alternative expressions for $\Upsilon_1, ..., \Upsilon_5$ in Appendix D.4.

**Lemma B.5** (Proved in Appendix D.6). *Under the same conditions of Theorem 3.6 with $\Sigma = I_p$, we have*

$$\max_{k \in \{1,3,4\}} \mathbb{E}[\|\Upsilon_k\|_{\text{op}}^2 \mid \varepsilon] \le C(T, \gamma, \eta_{\max}, c_0)(\delta^2 + \|b^*\|^2),$$

$$\mathbb{E}[\|\Upsilon_2\|_{\text{op}}^2 \mid \varepsilon] \le n^{-1}C(T, \gamma, \eta_{\max}, c_0)(\delta^2 + \|b^*\|^2),$$

$$\mathbb{E}[\|\Upsilon_5\|_{\text{op}}^2 \mid \varepsilon] \le n^2 C(T, \gamma, \eta_{\max}, c_0)(\delta^2 + \|b^*\|^2).$$

We further define a few matrices of size $T \times T$:

$$\Theta_1 = F^\top X H + \widetilde{K} H^\top H - F^\top F W^\top,$$

$$\Theta_2 = n^{-1}[F^\top X X^\top \widetilde{F} + \widetilde{K} H^\top X^\top \widetilde{F} - p F^\top \widetilde{F} + F^\top F \widehat{A}^\top],$$

$$\Theta_3 = H^\top X^\top X H - (n I_T - \widetilde{A}) H^\top H - H^\top X^\top F W^\top,$$

$$\Theta_4 = \frac{p}{n} F^\top \widetilde{F} - \frac{1}{n}(\widetilde{K} H^\top + F^\top X)(\widehat{K} H^\top + \widetilde{F}^\top X)^\top,$$

$$\Theta_5 = n H^\top H - (W F^\top - H^\top X^\top)(W F^\top - H^\top X^\top)^\top,$$

$$\Theta_6 = \|E\|_{\mathrm{F}}^{-1}(E^\top X H - E^\top F W^\top). \tag{27}$$

The next lemma provides the moment bounds for the Frobenius norm of these matrices.

**Lemma B.6** (Proved in Appendix D.7). *Under the same conditions of Theorem 3.6, we have*

$$\max_{k \in \{1,2,3\}} \mathbb{E}[\|\Theta_k\|_{\mathrm{F}}^2 \mid \varepsilon] \leq n C(T, \gamma, \eta_{\max}, c_0)(\delta^2 + \|b^*\|^2), \tag{28}$$

$$\max_{k \in \{4,5\}} \mathbb{E}[\|\Theta_k\|_{\mathrm{F}} \mid \varepsilon] \leq n^{1/2} C(T, \gamma, \eta_{\max}, c_0)(\delta^2 + \|b^*\|^2), \tag{29}$$

$$\mathbb{E}[\|\Theta_6\|_{\mathrm{F}}^2 \mid \varepsilon] \leq C(T, \gamma, \eta_{\max}, c_0)(\delta^2 + \|b^*\|^2), \tag{30}$$

*almost surely, where $\mathbb{E}[\cdot \mid \varepsilon]$ is the conditional expectation given $\varepsilon$.*

We are able to prove the main theorems using Lemma B.6.

# C    Proof of main results

## C.1    Proof of Theorem 3.6

It suffices to prove this theorem for the case $\Sigma = I_p$. When $\Sigma \neq I_p$, the result can be derived using a change of variables argument, as outlined in Appendix B.1. By basic algebra, we have

$$\begin{aligned}
&\Theta_5 + \|E\|_{\mathrm{F}}(\Theta_6 + \Theta_6^\top) \\
&= n H^\top H - (X H - F W^\top)^\top (X H - F W^\top) + E^\top (X H - F W^\top) + (X H - F W^\top)^\top E \\
&= n H^\top H + E^\top E - (E - X H + F W^\top)^\top (E - X H + F W^\top) \\
&= n H^\top H + E^\top E - (R + F W^\top)^\top (R + F W^\top).
\end{aligned}$$

Notice that $r_t = \|\widehat{b}^t - b^*\|^2 + \|\varepsilon\|^2/n$ is the $t$-th diagonal entry of $(H^\top H + E^\top E/n)$, and $\hat{r}_t$ is the $t$-th diagonal entry of $(R + F W^\top)^\top (R + F W^\top)/n$. Since $\|E\|_{\mathrm{F}} = \sqrt{T}\|\varepsilon\|$, using the previous display that conditionally on $\varepsilon$, we have

$$\mathbb{E}\Big[|\hat{r}_t - r_t| \mid \varepsilon\Big] \leq n^{-1}\mathbb{E}\Big[\|\Theta_5\|_{\mathrm{F}} + 2\|E\|_{\mathrm{F}}\|\Theta_6\|_{\mathrm{F}} \mid \varepsilon\Big] = n^{-1}\mathbb{E}\Big[\|\Theta_5\|_{\mathrm{F}} + 2\sqrt{T}\|\varepsilon\|\|\Theta_6\|_{\mathrm{F}} \mid \varepsilon\Big].$$

Using the moment bounds of $\Theta_5$ and $\Theta_6$ in Lemma B.6, we have

$$\mathbb{E}\Big[|\hat{r}_t - r_t| \mid \varepsilon\Big] \leq \frac{C(T, \gamma, \eta_{\max}, c_0, \delta)}{\sqrt{n}}\Big(1 + \frac{\|\varepsilon\|}{\sqrt{n}}\Big).$$

Furthermore, if $\mathbb{E}[|\varepsilon_i|]$ is finite, we have by [22] that $\|\varepsilon\|/n \to^P 0$ (convergence in probability) if $\varepsilon$ has i.i.d. entries with a fixed distribution independent of $n$. Under this assumption, the right-hand side of the previous display converges to 0 in probability. By enlarging the constant if necessary, assume $C(T, \gamma, \eta_{\max}, c_0, \delta) \geq 1$. To obtain a quantitative bound, by the conditional version of Markov's inequality, for any $\epsilon > 0$, and almost surely with respect to $\varepsilon$ that

$$\mathbb{P}\Big(|\hat{r}_t - r_t| > \epsilon \mid \varepsilon\Big) \leq \min\Big\{1, \frac{C(T, \gamma, \eta_{\max}, c_0, \delta)}{\epsilon}\Big(\frac{1}{\sqrt{n}} + \frac{\|\varepsilon\|}{n}\Big)\Big\}$$

$$\leq \max\Big\{1, \frac{C(T, \gamma, \eta_{\max}, c_0, \delta)}{\epsilon}\Big\} \min\Big\{1, \frac{1}{\sqrt{n}} + \frac{\|\varepsilon\|}{n}\Big\}.$$

Taking expectation with respect to $\varepsilon$, we obtain

$$\mathbb{P}\Big(|\hat{r}_t - r_t| > \epsilon\Big) \leq \max\Big\{1, \frac{C(T, \gamma, \eta_{\max}, c_0, \delta)}{\epsilon}\Big\} \mathbb{E}\Big[\min\Big\{1, \frac{1}{\sqrt{n}} + \frac{\|\varepsilon\|}{n}\Big\}\Big]$$

$$\leq \max\Big\{1, \frac{C(T, \gamma, \eta_{\max}, c_0, \delta)}{\epsilon}\Big\} \Big(\frac{1}{\sqrt{n}} + \mathbb{E}\Big[\min\Big\{1, \frac{\|\varepsilon\|}{n}\Big\}\Big]\Big)$$

with $\mathbb{E}[\min\{1, \frac{\|\varepsilon\|}{n}\}] \to 0$ (equivalently, $\|\varepsilon\|/n \to^P 0$) if the entries of $\varepsilon$ are i.i.d. with a fixed distribution independent of $n$ with $\mathbb{E}[|\varepsilon_i|] < +\infty$ by [22]. This finishes the proof of Theorem 3.6.

## C.2 Operator norm bound on $\widehat{K}$

We first recall the definition of $\widehat{K}$ from (10) in the main text:

$$\widehat{K} = \sum_{t=1}^{T} \mathrm{Tr}(D_t) e_t e_t^\top - \sum_{i=1}^{n} (I_T \otimes e_i^\top)\mathcal{D}(I_T \otimes X)\Gamma(I_T \otimes X^\top)\mathcal{S}\mathcal{D}(I_T \otimes e_i). \qquad (31)$$

Define two events: $\Omega_1 = \{X \in \mathbb{R}^{n \times p} : \|X\|_{\mathrm{op}}/\sqrt{n} \leq 2 + \sqrt{p/n}\}$ and $\Omega_2 = \{|\{i \in [n] : |\varepsilon_i| \leq M\}| \geq \frac{2n}{3}\}$, where $M$ is a large enough constant such that $\mathbb{P}(|\varepsilon_i| > M) \leq 1/6$.

**Lemma C.1.** *Under the same conditions of Theorem 3.6 with $\Sigma = I_p$, we have in the event $\Omega_* = \Omega_1 \cap \Omega_2$ that*

$$n\|\widehat{K}^{-1}\|_{\mathrm{op}} \leq C(T, \gamma, \eta_{\max}, c_0, \delta, \|b^*\|).$$

*Furthermore, $\Omega_*$ has probability at least $1 - e^{-n/18} - e^{-n/2}$.*

*Proof of Lemma C.1.* Under Assumptions 3.1 and 3.5, we know that $\mathbb{P}(\Omega_1) \geq 1 - e^{-n/2}$ from [11]. In the event $\Omega_1$, we have by Lemma D.2 that

$$\|XH\|_{\mathrm{F}}^2/n \leq C(T, \gamma, \eta_{\max}, c_0)(\delta^2 + \|b^*\|^2) := C_*.$$

Markov's inequality further implies

$$|\{i \in [n] : \|x_i^\top H\|^2 > 3C_*\}| \leq n/3.$$

In other words, $|\{i \in [n] : \|x_i^\top H\|^2 \leq 3C_*\}| \geq \frac{2n}{3}$. Recall that $M$ is such that $\mathbb{P}(|\varepsilon_i| > M) \leq 1/6$. By Hoeffding's inequality, we have

$$\mathbb{P}\Big(\frac{1}{n}\sum_{i=1}^{n} \mathbf{1}\{|\varepsilon_i| > M\} \geq \mathbb{P}(|\varepsilon_i| > M) + a\Big) \leq e^{-2na^2}.$$

Taking $a = 1/6$, we have $\mathbb{P}\big(\frac{1}{n}\sum_{i=1}^{n} \mathbf{1}\{|\varepsilon_i| > M\} \geq \mathbb{P}(|\varepsilon_i| > M) + 1/6\big) \leq e^{-n/18}$. Furthermore, using $|\{i \in [n] : |\varepsilon_i| > M\}| = \sum_{i=1}^{n} \mathbf{1}\{|\varepsilon_i| > M\}$, we have

$$\Big\{|\{i \in [n] : |\varepsilon_i| > M\}| \geq n/3|\Big\} = \Big\{\frac{1}{n}\sum_{i=1}^{n} \mathbf{1}\{|\varepsilon_i| > M\} \geq 1/6 + 1/6\Big\}$$

$$\subseteq \Big\{\frac{1}{n}\sum_{i=1}^{n} \mathbf{1}\{|\varepsilon_i| > M\} \geq \mathbb{P}(|\varepsilon_i| > M) + 1/6\Big\}.$$

Therefore,

$$\mathbb{P}\Big(|\{i \in [n] : |\varepsilon_i| > M\}| \geq n/3\Big) \leq e^{-n/18}.$$

Equivalently, we have $\mathbb{P}(|\{i \in [n] : |\varepsilon_i| \leq M\}| \geq \frac{2n}{3}) \geq 1 - e^{-n/18}$. That is, at least $\frac{2n}{3}$ of the entries of $\varepsilon$ are bounded by $M$ with probability at least $1 - e^{-n/18}$.

Recall that $\Omega_2 = \{|\{i \in [n] : |\varepsilon_i| \leq M\}| \geq \frac{2n}{3}\}$, then $\mathbb{P}(\Omega_2) \geq 1 - e^{-n/18}$. Hence, $\mathbb{P}(\Omega_1 \cap \Omega_2) \geq 1 - e^{-n/18} - e^{-n/2}$. In the event $\Omega_1 \cap \Omega_2$, the set

$$\hat{I} = \{i \in [n] : |\varepsilon_i| \leq M, \|x_i^\top H\|^2 \leq 3C_*\}$$

has size at least $\frac{n}{3}$. For any $i \in \hat{I}$ and $t \in [T]$, we have

$$|y_i - \boldsymbol{x}_i^\top \widehat{\boldsymbol{b}}^t| = |\varepsilon_i - \boldsymbol{x}_i^\top \boldsymbol{H} \boldsymbol{e}_t| \leq |\varepsilon_i| + |\boldsymbol{x}_i^\top \boldsymbol{H} \boldsymbol{e}_t| \leq M + \sqrt{3C_*}. \qquad (32)$$

By the definition of $\boldsymbol{D}_t$, under Assumption 3.4, we have

$$
\begin{aligned}
\mathrm{Tr}(\boldsymbol{D}_t) = \sum_{i=1}^n \rho''(y_i - \boldsymbol{x}_i^\top \widehat{\boldsymbol{b}}^t) & \\
> \sum_{i \in \hat{I}} \rho''(y_i - \boldsymbol{x}_i^\top \widehat{\boldsymbol{b}}^t) & \qquad \text{since } \rho'' \geq 0 \text{ by convexity} \\
\geq |\hat{I}| \min_{u:|u| \leq M + \sqrt{3C_*}} \rho''(u) & \qquad \text{due to (32)} \\
\geq n/3 \min_{u:|u| \leq M + \sqrt{3C_*}} \rho''(u) := c_* n & \qquad \text{since } \hat{I} \text{ has size at least } n/3.
\end{aligned}
$$

Here $c_*$ is a constant depending on $\rho, M, C_*$ only.

By the definition of $\widehat{\boldsymbol{K}}$ in (31), $\widehat{\boldsymbol{K}}/n$ is a lower triangular matrix with diagonal entries equal to $\mathrm{Tr}(\boldsymbol{D}_t)/n$. It is invertible if and only if all its diagonal entries are non-zero. Therefore, in the event $\Omega_1 \cap \Omega_2$, we have $\widehat{\boldsymbol{K}}/n$ is invertible.

Let $\widehat{\boldsymbol{\Lambda}} = \sum_{t=1}^T \mathrm{Tr}(\boldsymbol{D}_t) \boldsymbol{e}_t \boldsymbol{e}_t^\top$. Then it is diagonal, $\|\widehat{\boldsymbol{\Lambda}}^{-1}\|_{\mathrm{op}} = \max_{t \in [T]} \mathrm{Tr}[\boldsymbol{D}_t]^{-1} \leq (c_* n)^{-1}$ and

$$\widehat{\boldsymbol{K}} = \widehat{\boldsymbol{\Lambda}} - \widehat{\boldsymbol{L}} \qquad (33)$$

where $\widehat{\boldsymbol{L}} = \sum_{i=1}^n (\boldsymbol{I}_T \otimes \boldsymbol{e}_i^\top) \mathcal{D}(\boldsymbol{I}_T \otimes \boldsymbol{X}) \boldsymbol{\Gamma}(\boldsymbol{I}_T \otimes \boldsymbol{X}^\top) \mathcal{S} \mathcal{D}(\boldsymbol{I}_T \otimes \boldsymbol{e}_i)$. Here $\widehat{\boldsymbol{L}}$ is a strictly lower triangular matrix. Using the upper bound of $\|\boldsymbol{\Gamma}\|_{\mathrm{op}}$ in Lemma D.4, we have $\|\widehat{\boldsymbol{L}}\|_{\mathrm{op}} \leq n C(T, \gamma, \eta_{\max}, c_0)$ in the event $\Omega_1$. Now we rewrite $\widehat{\boldsymbol{K}}^{-1}$ as

$$\widehat{\boldsymbol{K}}^{-1} = (\widehat{\boldsymbol{K}} \widehat{\boldsymbol{\Lambda}}^{-1} \widehat{\boldsymbol{\Lambda}})^{-1} = \widehat{\boldsymbol{\Lambda}}^{-1} (\widehat{\boldsymbol{K}} \widehat{\boldsymbol{\Lambda}}^{-1})^{-1} = \widehat{\boldsymbol{\Lambda}}^{-1} (\boldsymbol{I}_T - \widehat{\boldsymbol{L}} \widehat{\boldsymbol{\Lambda}}^{-1})^{-1}.$$

Notice that $\widehat{\boldsymbol{L}} \widehat{\boldsymbol{\Lambda}}^{-1} \in \mathbb{R}^{T \times T}$ is a strictly lower triangular matrix, thus

$$(\boldsymbol{I}_T - \widehat{\boldsymbol{L}} \widehat{\boldsymbol{\Lambda}}^{-1})^{-1} = \sum_{k=0}^\infty (\widehat{\boldsymbol{L}} \widehat{\boldsymbol{\Lambda}}^{-1})^k = \sum_{k=0}^{T-1} (\widehat{\boldsymbol{L}} \widehat{\boldsymbol{\Lambda}}^{-1})^k.$$

By the triangle inequality,

$$\|(\boldsymbol{I}_T - \widehat{\boldsymbol{L}} \widehat{\boldsymbol{\Lambda}}^{-1})^{-1}\|_{\mathrm{op}} \leq \sum_{k=0}^{T-1} \|\widehat{\boldsymbol{L}} \widehat{\boldsymbol{\Lambda}}^{-1}\|_{\mathrm{op}}^k \leq C(T, \gamma, \eta_{\max}, c_0, \delta, \|\boldsymbol{b}^*\|).$$

Therefore, in the event $\Omega_1 \cap \Omega_2$ which has probability $\mathbb{P}(\Omega_1 \cap \Omega_2) \geq 1 - e^{-n/18} - e^{-n/2}$,

$$\|\widehat{\boldsymbol{K}}^{-1}\|_{\mathrm{op}} \leq \|\widehat{\boldsymbol{\Lambda}}^{-1}\|_{\mathrm{op}} \|(\boldsymbol{I}_T - \widehat{\boldsymbol{L}} \widehat{\boldsymbol{\Lambda}}^{-1})^{-1}\|_{\mathrm{op}} \leq n^{-1} C(T, \gamma, \eta_{\max}, c_0, \delta, \|\boldsymbol{b}^*\|).$$

$\square$

**Lemma C.2.** *Under the same conditions of Theorem 3.7 with $\boldsymbol{\Sigma} = \boldsymbol{I}_p$, we have*

$$\|\widehat{\boldsymbol{K}}\|_{\mathrm{op}} \leq n(1 + \|\boldsymbol{X}\|_{\mathrm{op}}^2 \|\boldsymbol{\Gamma}\|_{\mathrm{op}}), \qquad \|\widehat{\mathbf{A}}\|_{\mathrm{op}} \leq n \|\boldsymbol{X}\|_{\mathrm{op}}^2 \|\boldsymbol{\Gamma}\|_{\mathrm{op}}, \qquad \|\boldsymbol{W}\|_{\mathrm{op}} \leq n \|\boldsymbol{\Gamma}\|_{\mathrm{op}}.$$

*Proof of Lemma C.2.* By the definition of $\widehat{\boldsymbol{K}}$ in (31), using $\|\mathcal{D}\|_{\mathrm{op}} \leq 1$ and $\|\mathcal{S}\|_{\mathrm{op}} \leq 1$, we have

$$\|\widehat{\boldsymbol{K}}\|_{\mathrm{op}} \leq \|\widehat{\boldsymbol{\Lambda}}\|_{\mathrm{op}} + \|\widehat{\boldsymbol{L}}\|_{\mathrm{op}} \leq n(1 + \|\boldsymbol{X}\|_{\mathrm{op}}^2 \|\boldsymbol{\Gamma}\|_{\mathrm{op}}).$$

Similarly, by the definition of $\widehat{\mathbf{A}}$ in (24), we have

$$\|\widehat{\mathbf{A}}\|_{\mathrm{op}} \leq n \|\boldsymbol{X}\|_{\mathrm{op}}^2 \|\boldsymbol{\Gamma}\|_{\mathrm{op}}.$$

Last, by the definition of $\boldsymbol{W}$ in (25), we have $\|\boldsymbol{W}\|_{\mathrm{op}} \leq n \|\boldsymbol{\Gamma}\|_{\mathrm{op}}$. $\square$

**Lemma C.3.** *Under the same conditions of Theorem 3.6 with $\boldsymbol{\Sigma} = \boldsymbol{I}_p$, we have*

$$\mathbb{E}[\|\boldsymbol{F}^\top \boldsymbol{F}(\widehat{\boldsymbol{A}} - \widehat{\boldsymbol{K}}\boldsymbol{W})^\top\|_{\mathrm{F}} \mid \boldsymbol{\varepsilon}] \leq n^{3/2} C(T, \gamma, \eta_{\max}, c_0)(\delta^2 + \|\boldsymbol{b}^*\|^2),$$

$$\mathbb{E}[\|\boldsymbol{R}^\top \boldsymbol{F}(\widehat{\boldsymbol{A}} - \widehat{\boldsymbol{K}}\boldsymbol{W})^\top\|_{\mathrm{F}} \mid \boldsymbol{\varepsilon}] \leq n^2 (\tfrac{\|\boldsymbol{\varepsilon}\|}{n} + \tfrac{1}{\sqrt{n}}) C(T, \gamma, \eta_{\max}, c_0)(\delta^2 + \|\boldsymbol{b}^*\|^2).$$

*Proof of Lemma C.3.* First, using the definitions of $\boldsymbol{\Theta}_1, \boldsymbol{\Theta}_2, \boldsymbol{\Theta}_4$ in (27), we have

$$n^{-1}\boldsymbol{F}^\top \boldsymbol{F}(\widehat{\boldsymbol{A}} - \widehat{\boldsymbol{K}}\boldsymbol{W})^\top = n^{-1}\boldsymbol{\Theta}_1 \widehat{\boldsymbol{K}}^\top + \boldsymbol{\Theta}_2 + \boldsymbol{\Theta}_4. \tag{34}$$

Hence,

$$\mathbb{E}[\|\boldsymbol{F}^\top \boldsymbol{F}(\widehat{\boldsymbol{A}} - \widehat{\boldsymbol{K}}\boldsymbol{W})^\top\|_{\mathrm{F}} \mid \boldsymbol{\varepsilon}]$$

$$= \mathbb{E}[\|(\boldsymbol{\Theta}_1 \widehat{\boldsymbol{K}}^\top + n\boldsymbol{\Theta}_2 + n\boldsymbol{\Theta}_4)\|_{\mathrm{F}} \mid \boldsymbol{\varepsilon}] \qquad\qquad \text{by (34)}$$

$$\leq \mathbb{E}[\|\boldsymbol{\Theta}_1\|_{\mathrm{F}}^2 \mid \boldsymbol{\varepsilon}]^{1/2} \mathbb{E}[\|\widehat{\boldsymbol{K}}\|_{\mathrm{op}}^2 \mid \boldsymbol{\varepsilon}]^{1/2} + n\mathbb{E}[\|\boldsymbol{\Theta}_2\|_{\mathrm{F}} + \|\boldsymbol{\Theta}_4\|_{\mathrm{F}} \mid \boldsymbol{\varepsilon}] \quad \text{by the Cauchy-Schwarz inequality}$$

$$\leq n^{3/2} C(T, \gamma, \eta_{\max}, c_0)(\delta^2 + \|\boldsymbol{b}^*\|^2).$$

Here the last line uses the upper bounds of $\mathbb{E}[\|\boldsymbol{\Theta}_k\|_{\mathrm{F}} \mid \boldsymbol{\varepsilon}]$ from Lemma B.6, and the bound of $\mathbb{E}[\|\widehat{\boldsymbol{K}}\|_{\mathrm{op}} \mid \boldsymbol{\varepsilon}]$ follows from Lemma C.2 and the bound of $\|\boldsymbol{\Gamma}\|_{\mathrm{op}}$ from Lemma D.4. This proves the first inequality.

For the second inequality, we define

$$\check{\boldsymbol{\Theta}}_1 = \frac{\boldsymbol{R}^\top \boldsymbol{X}\boldsymbol{H} + \check{\boldsymbol{K}}\boldsymbol{H}^\top \boldsymbol{H} - \boldsymbol{R}^\top \boldsymbol{F}\boldsymbol{W}^\top}{\|\boldsymbol{E}\|_{\mathrm{F}}/\sqrt{n} + 1},$$

$$\check{\boldsymbol{\Theta}}_2 = \frac{\boldsymbol{R}^\top \boldsymbol{X}\boldsymbol{X}^\top \widetilde{\boldsymbol{F}} + \check{\boldsymbol{K}}\boldsymbol{H}^\top \boldsymbol{X}^\top \widetilde{\boldsymbol{F}} - p\boldsymbol{R}^\top \widetilde{\boldsymbol{F}} + \boldsymbol{R}^\top \boldsymbol{F}\widehat{\boldsymbol{A}}^\top}{n(\|\boldsymbol{E}\|_{\mathrm{F}}/\sqrt{n} + 1)},$$

$$\check{\boldsymbol{\Theta}}_4 = \frac{p\boldsymbol{R}^\top \widetilde{\boldsymbol{F}} - (\check{\boldsymbol{K}}\boldsymbol{H}^\top + \boldsymbol{R}^\top \boldsymbol{X})(\widehat{\boldsymbol{K}}\boldsymbol{H}^\top + \widetilde{\boldsymbol{F}}^\top \boldsymbol{X})^\top}{n(\|\boldsymbol{E}\|_{\mathrm{F}}/\sqrt{n} + 1)},$$

where $\check{\boldsymbol{K}} = n\boldsymbol{I}_T - \sum_{i=1}^n (\boldsymbol{I}_T \otimes (\boldsymbol{e}_i^\top \boldsymbol{X}))\boldsymbol{\Gamma}(\boldsymbol{I}_T \otimes \boldsymbol{X}^\top)\mathcal{SD}(\boldsymbol{I}_T \otimes (\boldsymbol{X}^\top \boldsymbol{e}_i))$. Using similar argument that proves Lemma B.6, we can obtain the following bound of $\check{\boldsymbol{\Theta}}_1, \check{\boldsymbol{\Theta}}_2, \check{\boldsymbol{\Theta}}_4$.

$$\max_{k \in \{1,2\}} \mathbb{E}[\|\check{\boldsymbol{\Theta}}_k\|_{\mathrm{F}}^2 \mid \boldsymbol{\varepsilon}] \leq n C(T, \gamma, \eta_{\max}, c_0)(\delta^2 + \|\boldsymbol{b}^*\|^2), \tag{35}$$

$$\mathbb{E}[\|\check{\boldsymbol{\Theta}}_4\|_{\mathrm{F}} \mid \boldsymbol{\varepsilon}] \leq n^{1/2} C(T, \gamma, \eta_{\max}, c_0)(\delta^2 + \|\boldsymbol{b}^*\|^2). \tag{36}$$

By the definitions of $\check{\boldsymbol{\Theta}}_1, \check{\boldsymbol{\Theta}}_2, \check{\boldsymbol{\Theta}}_4$, we have

$$(\|\boldsymbol{E}\|_{\mathrm{F}}/\sqrt{n} + 1)[n^{-1}\check{\boldsymbol{\Theta}}_1 \widehat{\boldsymbol{K}}^\top + \check{\boldsymbol{\Theta}}_2 + \check{\boldsymbol{\Theta}}_4] = n^{-1}\boldsymbol{R}^\top \boldsymbol{F}(\widehat{\boldsymbol{A}} - \widehat{\boldsymbol{K}}\boldsymbol{W})^\top.$$

Therefore, conditional on $\boldsymbol{\varepsilon}$, we have

$$\mathbb{E}[\|\boldsymbol{R}^\top \boldsymbol{F}(\widehat{\boldsymbol{A}} - \widehat{\boldsymbol{K}}\boldsymbol{W})^\top\|_{\mathrm{F}} \mid \boldsymbol{\varepsilon}] = (\|\boldsymbol{E}\|_{\mathrm{F}}/\sqrt{n} + 1)\mathbb{E}[\|\check{\boldsymbol{\Theta}}_1 \widehat{\boldsymbol{K}}^\top + n\check{\boldsymbol{\Theta}}_2 + n\check{\boldsymbol{\Theta}}_4\|_{\mathrm{F}} \mid \boldsymbol{\varepsilon}]$$

$$\leq n^{3/2}(\|\boldsymbol{E}\|_{\mathrm{F}}/\sqrt{n} + 1)C(T, \gamma, \eta_{\max}, c_0)(\delta^2 + \|\boldsymbol{b}^*\|^2).$$

This finishes the proof of Lemma C.3. $\qquad\qquad\qquad\qquad\qquad\qquad\qquad\qquad\square$

### C.3 Proof of Theorem 3.7

In the event $\Omega_* = \Omega_1 \cap \Omega_2$, we know from Lemma C.1 that $\widehat{\boldsymbol{K}}$ is invertible and $\|\widehat{\boldsymbol{K}}^{-1}\|_{\mathrm{op}} \leq n^{-1}C$. Define $\widetilde{\boldsymbol{W}} = \widehat{\boldsymbol{K}}^{-1}\widehat{\boldsymbol{A}}$. Using $\boldsymbol{R} + \boldsymbol{F}\widetilde{\boldsymbol{W}}^\top = \boldsymbol{R} + \boldsymbol{F}\boldsymbol{W}^\top + \boldsymbol{F}(\widetilde{\boldsymbol{W}} - \boldsymbol{W})^\top$, we have

$$(\boldsymbol{R} + \boldsymbol{F}\widetilde{\boldsymbol{W}}^\top)^\top(\boldsymbol{R} + \boldsymbol{F}\widetilde{\boldsymbol{W}}^\top) - (\boldsymbol{R} + \boldsymbol{F}\boldsymbol{W}^\top)^\top(\boldsymbol{R} + \boldsymbol{F}\boldsymbol{W}^\top)$$

$$= (\boldsymbol{R} + \boldsymbol{F}\boldsymbol{W}^\top)^\top \boldsymbol{F}(\widetilde{\boldsymbol{W}} - \boldsymbol{W})^\top + (\widetilde{\boldsymbol{W}} - \boldsymbol{W})\boldsymbol{F}^\top(\boldsymbol{R} + \boldsymbol{F}\boldsymbol{W}^\top) + (\widetilde{\boldsymbol{W}} - \boldsymbol{W})\boldsymbol{F}^\top \boldsymbol{F}(\widetilde{\boldsymbol{W}} - \boldsymbol{W})^\top.$$

We have by the triangle inequality

$$\|(\boldsymbol{R} + \boldsymbol{F}\widetilde{\boldsymbol{W}}^\top)^\top(\boldsymbol{R} + \boldsymbol{F}\widetilde{\boldsymbol{W}}^\top) - (\boldsymbol{R} + \boldsymbol{F}\boldsymbol{W}^\top)^\top(\boldsymbol{R} + \boldsymbol{F}\boldsymbol{W}^\top)\|_{\mathrm{F}}$$

$$\leq 2\|(\boldsymbol{R} + \boldsymbol{F}\boldsymbol{W}^\top)^\top \boldsymbol{F}(\widetilde{\boldsymbol{W}} - \boldsymbol{W})^\top\|_{\mathrm{F}} + \|(\widetilde{\boldsymbol{W}} - \boldsymbol{W})\boldsymbol{F}^\top \boldsymbol{F}(\widetilde{\boldsymbol{W}} - \boldsymbol{W})\|_{\mathrm{F}}$$

$$\lesssim \|\boldsymbol{R}^\top \boldsymbol{F}(\widetilde{\boldsymbol{W}} - \boldsymbol{W})^\top\|_{\mathrm{F}} + \|\boldsymbol{W}\boldsymbol{F}^\top \boldsymbol{F}(\widetilde{\boldsymbol{W}} - \boldsymbol{W})^\top\|_{\mathrm{F}} + \|(\widetilde{\boldsymbol{W}} - \boldsymbol{W})\boldsymbol{F}^\top \boldsymbol{F}(\widetilde{\boldsymbol{W}} - \boldsymbol{W})\|_{\mathrm{F}}.$$

Recall that in the event $\Omega_*$, we have $\|\widehat{\boldsymbol{K}}^{-1}\|_{\mathrm{op}} \leq n^{-1}C$. Using $\widehat{\mathbf{A}} - \widehat{\boldsymbol{K}}\boldsymbol{W} = \widehat{\boldsymbol{K}}(\widetilde{\boldsymbol{W}} - \boldsymbol{W})$ and Lemma C.3, we have

$$\mathbb{E}\Big[I(\Omega_*)\|(\boldsymbol{R} + \boldsymbol{F}\widetilde{\boldsymbol{W}}^\top)^\top(\boldsymbol{R} + \boldsymbol{F}\widetilde{\boldsymbol{W}}^\top) - (\boldsymbol{R} + \boldsymbol{F}\boldsymbol{W}^\top)^\top(\boldsymbol{R} + \boldsymbol{F}\boldsymbol{W}^\top)\|_{\mathrm{F}} \mid \boldsymbol{\varepsilon}\Big]$$

$$\lesssim \mathbb{E}\Big[I(\Omega_*)\|\boldsymbol{R}^\top\boldsymbol{F}(\widetilde{\boldsymbol{W}} - \boldsymbol{W})^\top\|_{\mathrm{F}} \mid \boldsymbol{\varepsilon}\Big]$$

$$+ \mathbb{E}\Big[I(\Omega_*)\|\boldsymbol{W}\boldsymbol{F}^\top\boldsymbol{F}(\widetilde{\boldsymbol{W}} - \boldsymbol{W})^\top\|_{\mathrm{F}} \mid \boldsymbol{\varepsilon}\Big]$$

$$+ \mathbb{E}\Big[I(\Omega_*)\|(\widetilde{\boldsymbol{W}} - \boldsymbol{W})\boldsymbol{F}^\top\boldsymbol{F}(\widetilde{\boldsymbol{W}} - \boldsymbol{W})^\top\|_{\mathrm{F}} \mid \boldsymbol{\varepsilon}\Big]$$

$$= \mathbb{E}\Big[I(\Omega_*)\|\boldsymbol{R}^\top\boldsymbol{F}(\widehat{\mathbf{A}} - \widehat{\boldsymbol{K}}\boldsymbol{W})^\top(\widehat{\boldsymbol{K}}^\top)^{-1}\|_{\mathrm{F}} \mid \boldsymbol{\varepsilon}\Big]$$

$$+ \mathbb{E}\Big[I(\Omega_*)\|\boldsymbol{W}\boldsymbol{F}^\top\boldsymbol{F}(\widehat{\mathbf{A}} - \widehat{\boldsymbol{K}}\boldsymbol{W})^\top(\widehat{\boldsymbol{K}}^\top)^{-1}\|_{\mathrm{F}} \mid \boldsymbol{\varepsilon}\Big]$$

$$+ \mathbb{E}\Big[I(\Omega_*)\|\widehat{\boldsymbol{K}}^{-1}(\widehat{\mathbf{A}} - \widehat{\boldsymbol{K}}\boldsymbol{W})\boldsymbol{F}^\top\boldsymbol{F}(\widehat{\mathbf{A}} - \widehat{\boldsymbol{K}}\boldsymbol{W})^\top(\widehat{\boldsymbol{K}}^\top)^{-1}\|_{\mathrm{F}} \mid \boldsymbol{\varepsilon}\Big].$$

According to Lemma C.3 and the bound of $\|\widehat{\boldsymbol{K}}^{-1}\|_{\mathrm{op}}$ in Lemma C.1, the first conditional expectation is bounded from above by

$$n(\tfrac{\|\boldsymbol{\varepsilon}\|}{n} + \tfrac{1}{\sqrt{n}})C(T, \gamma, \eta_{\max}, c_0)(\delta^2 + \|\boldsymbol{b}^*\|^2).$$

Using the bound of $\|\boldsymbol{K}^{-1}\|_{\mathrm{op}}$ in Lemma C.1, the bound of $\|\boldsymbol{W}\|_{\mathrm{op}}$ in Lemma C.2, and the bound of $\mathbb{E}[\|\boldsymbol{F}^\top\boldsymbol{F}(\widehat{\mathbf{A}} - \widehat{\boldsymbol{K}}\boldsymbol{W})^\top\|_{\mathrm{F}} \mid \boldsymbol{\varepsilon}]$ in Lemma C.3, the second conditional expectation is bounded from above by

$$n^{1/2}C(T, \gamma, \eta_{\max}, c_0)(\delta^2 + \|\boldsymbol{b}^*\|^2).$$

Similarly, the third conditional expectation is bounded from above by

$$n^{1/2}C(T, \gamma, \eta_{\max}, c_0)(\delta^2 + \|\boldsymbol{b}^*\|^2).$$

In summary, we have

$$n^{-1}\mathbb{E}[I(\Omega_*)\|(\boldsymbol{R} + \boldsymbol{F}\widetilde{\boldsymbol{W}}^\top)^\top(\boldsymbol{R} + \boldsymbol{F}\widetilde{\boldsymbol{W}}^\top) - (\boldsymbol{R} + \boldsymbol{F}\boldsymbol{W}^\top)^\top(\boldsymbol{R} + \boldsymbol{F}\boldsymbol{W}^\top)\|_{\mathrm{F}} \mid \boldsymbol{\varepsilon}]$$

$$\leq \tfrac{1}{\sqrt{n}}\big(\tfrac{\|\boldsymbol{\varepsilon}\|}{\sqrt{n}} + 1\big)C(T, \gamma, \eta_{\max}, c_0)(\delta^2 + \|\boldsymbol{b}^*\|^2).$$

Since $\tilde{r}_t$ and $\hat{r}_t$ are the $t$-th diagonal entries of $(\boldsymbol{R} + \boldsymbol{F}\widetilde{\boldsymbol{W}}^\top)^\top(\boldsymbol{R} + \boldsymbol{F}\widetilde{\boldsymbol{W}}^\top)/n$ and $(\boldsymbol{R} + \boldsymbol{F}\boldsymbol{W}^\top)^\top(\boldsymbol{R} + \boldsymbol{F}\boldsymbol{W}^\top)/n$, respectively, we have

$$\mathbb{E}[I(\Omega_*)|\tilde{r}_t - \hat{r}_t| \mid \boldsymbol{\varepsilon}]$$

$$\leq \mathbb{E}\Big[I(\Omega_*)\|(\boldsymbol{R} + \boldsymbol{F}\widetilde{\boldsymbol{W}}^\top)^\top(\boldsymbol{R} + \boldsymbol{F}\widetilde{\boldsymbol{W}}^\top) - (\boldsymbol{R} + \boldsymbol{F}\boldsymbol{W}^\top)^\top(\boldsymbol{R} + \boldsymbol{F}\boldsymbol{W}^\top)\|_{\mathrm{F}} \mid \boldsymbol{\varepsilon}\Big]$$

$$\leq \tfrac{1}{\sqrt{n}}\big(\tfrac{\|\boldsymbol{\varepsilon}\|}{\sqrt{n}} + 1\big)C(T, \gamma, \eta_{\max}, c_0)(\delta^2 + \|\boldsymbol{b}^*\|^2).$$

Using the same argument in the proof of Theorem 3.6, we have for any $\epsilon > 0$,

$$\mathbb{P}\Big(I(\Omega_*)|\tilde{r}_t - \hat{r}_t| > \epsilon \mid \boldsymbol{\varepsilon}\Big) \leq \min\Big(1, \tfrac{C(T,\gamma,\eta_{\max},c_0,\delta)}{\epsilon}\big(\tfrac{1}{\sqrt{n}} + \tfrac{\|\boldsymbol{\varepsilon}\|}{n}\big)\Big)$$

$$\leq \max\{1, \tfrac{C(T,\gamma,\eta_{\max},c_0,\delta)}{\epsilon}\}\min(1, \tfrac{1}{\sqrt{n}} + \tfrac{\|\boldsymbol{\varepsilon}\|}{n}).$$

Taking expectation with respect to $\boldsymbol{\varepsilon}$, we have

$$\mathbb{P}\Big(I(\Omega_*)|\tilde{r}_t - \hat{r}_t| > \epsilon\Big) \leq \max\{1, \tfrac{C(T,\gamma,\eta_{\max},c_0,\delta)}{\epsilon}\}\mathbb{E}[\min(1, \tfrac{1}{\sqrt{n}} + \tfrac{\|\boldsymbol{\varepsilon}\|}{n})].$$

Using the union bound and $\mathbb{P}(\Omega_*) \geq 1 - e^{-n/18} - e^{-n/2} \geq 1 - 2e^{-n/18}$, we obtain

$$\mathbb{P}\Big(|\tilde{r}_t - \hat{r}_t| > \epsilon\Big) \leq 2e^{-n/18} + \max\{1, \tfrac{C(T,\gamma,\eta_{\max},c_0,\delta)}{\epsilon}\}\big[\tfrac{1}{\sqrt{n}} + \mathbb{E}[\min(1, \tfrac{\|\boldsymbol{\varepsilon}\|}{n})]\big].$$

Using the triangle inequality and the tail probability of $|\hat{r}_t - r_t|$ in Theorem 3.6, we have

$$\mathbb{P}\Big(|\tilde{r}_t - r_t| > \epsilon\Big) \leq 2e^{-n/18} + \max\{1, \tfrac{C(T,\gamma,\eta_{\max},c_0,\delta)}{\epsilon}\}\big[\tfrac{1}{\sqrt{n}} + \mathbb{E}[\min(1, \tfrac{\|\boldsymbol{\varepsilon}\|}{n})]\big].$$

# D Proof of results in Appendix B.2

## D.1 Proof of Lemma B.1

By assumption, we know $\widehat{b}^1$ is independent of $X$ and $\widehat{b}^{t+1} = \phi_t\big(\widehat{b}^t + \frac{\eta_t}{|I_t|} X^\top S_t \psi(y - X\widehat{b}^t)\big)$ from (4). Recall that $D_t = \frac{\partial \psi(u)}{\partial u}\big|_{u=y-X\widehat{b}^t}$ and $\widetilde{D}_t = \frac{\partial \phi_t(v)}{\partial v}\big|_{v=\widehat{b}^t + \frac{\eta_t}{|I_t|} X^\top S_t \psi(y-X\widehat{b}^t)}$. Let a dot denote the derivative with respect to $x_{ij}$. By product rule and chain rule and using $y - X\widehat{b}^t = \varepsilon - X(\widehat{b}^t - b^*)$, we have

$$\dot{b}^{t+1} = \widetilde{D}_t \Big[ \dot{b}^t + \frac{\eta_t}{|I_t|}\Big( \dot{X}^\top S_t \psi(y - X\widehat{b}^t) - X^\top S_t D_t(\dot{X}(\widehat{b}^t - b^*) + X\dot{b}^t)\Big)\Big]$$

$$= \widetilde{D}_t \Big[ \dot{b}^t + \frac{\eta_t}{|I_t|}\Big( \dot{X}^\top F_t - X^\top S_t D_t(\dot{X}H_t + X\dot{b}^t)\Big)\Big],$$

where the last line uses $F_t = S_t \psi(y - X\widehat{b}^t)$ and $H_t = \widehat{b}^t - b^*$. Arranging terms gives

$$-\widetilde{D}_t(I_p - \tfrac{\eta_t}{|I_t|} X^\top S_t D_t X)\dot{b}^t + \dot{b}^{t+1} = \tfrac{\eta_t}{|I_t|}\widetilde{D}_t(\dot{X}^\top F_t - X^\top S_t D_t \dot{X} H_t).$$

Let $P_t = \widetilde{D}_t(I_p - \tfrac{\eta_t}{|I_t|} X^\top S_t D_t X)$ and $a_t = \tfrac{\eta_t}{|I_t|}\widetilde{D}_t(\dot{X}^\top F_t - X^\top S_t D_t \dot{X} H_t)$, we can rewrite the above recursion of $\dot{b}^t$ as a linear system:

$$\underbrace{\begin{bmatrix} I_p & & & \\ -P_1 & I_p & & \\ & \ddots & \ddots & \\ & & -P_{T-1} & I_p \end{bmatrix}}_{\mathcal{M}} \begin{bmatrix} \dot{b}^1 \\ \dot{b}^2 \\ \vdots \\ \dot{b}^T \end{bmatrix} = \underbrace{\begin{bmatrix} 0 & & & \\ I_p & 0 & & \\ & \ddots & \ddots & \\ & & I_p & 0 \end{bmatrix}}_{L} \underbrace{\begin{bmatrix} a_1 \\ a_2 \\ \vdots \\ a_T \end{bmatrix}}_{a}.$$

Solving the above system, we have $\dot{b}^t = (e_t^\top \otimes I_p)\mathcal{M}^{-1} L a$. Since $\dot{X} = \frac{\partial X}{\partial x_{ij}} = e_i e_j^\top$, $a_t$ can be further simplified as

$$a_t = \tfrac{\eta_t}{|I_t|}\widetilde{D}_t\big(e_j e_i^\top F_t - X^\top S_t D_t e_i e_j^\top H_t\big).$$

Using $\mathcal{D} = \sum_{t=1}^T \big((e_t e_t^\top) \otimes D_t\big)$, $\widetilde{\mathcal{D}} = \sum_{t=1}^T \big((e_t e_t^\top) \otimes \widetilde{D}_t\big)$, $\mathcal{S} = \sum_{t=1}^T \big((e_t e_t^\top) \otimes S_t\big)$, and $\Lambda = \sum_{t=1}^T \frac{\eta_t}{|I_t|} e_t e_t^\top$, we have

$$a = (\Lambda \otimes I_p)\widetilde{\mathcal{D}}\big[\mathrm{vec}(e_j e_i^\top F) - (I_T \otimes X^\top)\mathcal{S}\mathcal{D}\,\mathrm{vec}(e_i e_j^\top H)\big]$$

$$= (\Lambda \otimes I_p)\widetilde{\mathcal{D}}\big[((F^\top e_i) \otimes e_j) - (I_T \otimes X^\top)\mathcal{S}\mathcal{D}((H^\top e_j) \otimes e_i)\big].$$

Plugging this expression for $a$ into $\dot{b}^t = (e_t^\top \otimes I_p)\mathcal{M}^{-1} L a$ gives

$$\frac{\partial \widehat{b}^t}{\partial x_{ij}} = (e_t^\top \otimes I_p)\mathcal{M}^{-1} L(\Lambda \otimes I_p)\widetilde{\mathcal{D}}\big[((F^\top e_i) \otimes e_j) - (I_T \otimes X^\top)\mathcal{S}\mathcal{D}((H^\top e_j) \otimes e_i)\big].$$

This finishes the proof of Lemma B.1.

## D.2 Proof of Lemma B.2

By definition, $F_{lt} = e_l^\top S_t \psi(y - X\widehat{b}^t)$. By the chain rule of differentiation, we have

$$\frac{\partial F_{lt}}{\partial x_{ij}} = e_l^\top \frac{\partial S_t \psi(y - X\widehat{b}^t)}{\partial x_{ij}} = -e_l^\top S_t D_t(e_i e_j^\top H e_t + X \frac{\partial \widehat{b}^t}{\partial x_{ij}}).$$

Notice that $(e_t \otimes (D_t S_t)) = \mathcal{D}\mathcal{S}(e_t \otimes I_n)$. The desired formula then follows by plugging in the expression of $\frac{\partial \widehat{b}^t}{\partial x_{ij}}$ in Lemma B.1.

## D.3 Proof of Lemma B.3

The desired identity follows by

$$\frac{\partial \tilde{F}_{lt}}{\partial x_{ij}} = e_l^\top \frac{\partial \psi(y - X\widehat{b}^t)}{\partial x_{ij}} = -e_l^\top D_t(e_i e_j^\top H e_t + X \frac{\partial \widehat{b}^t}{\partial x_{ij}})$$

and the expression of $\frac{\partial \widehat{b}^t}{\partial x_{ij}}$ in Lemma B.1.

## D.4 Alternative expressions for the matrices defined in Definition B.4

This section derives alternative expressions for the matrices $\mathbf{\Upsilon}_1, ..., \mathbf{\Upsilon}_5$ defined in Definition B.4.

We first study $\mathbf{\Upsilon}_1$ in (18). Using $\boldsymbol{F} = \sum_{t=1}^T \boldsymbol{F} \boldsymbol{e}_t \boldsymbol{e}_t^\top$, we have by Lemma B.2

$$\sum_{j=1}^n \frac{\partial \boldsymbol{e}_i^\top \boldsymbol{F}}{\partial x_{ij}} = \sum_{i=1}^n \sum_{t=1}^T \frac{\partial F_{it}}{\partial x_{ij}} \boldsymbol{e}_t^\top = \sum_{i=1}^n \sum_{t=1}^T D_{ij}^{it} \boldsymbol{e}_t^\top + \sum_{i=1}^n \sum_{t=1}^T \Delta_{ij}^{it} \boldsymbol{e}_t^\top. \tag{37}$$

Now we compute the two terms in the right-hand side of (37). For the first term, using the expression of $D_{ij}^{lt}$ in Lemma B.2,

$$\sum_{i,t} D_{ij}^{it} \boldsymbol{e}_t^\top$$

$$= - \boldsymbol{e}_j^\top \boldsymbol{H} \sum_{t=1}^T \mathrm{Tr}(\boldsymbol{S}_t \boldsymbol{D}_t) \boldsymbol{e}_t \boldsymbol{e}_t^\top + \sum_{i,t} ((\boldsymbol{e}_j^\top \boldsymbol{H}) \otimes \boldsymbol{e}_i^\top) \mathcal{D} \mathcal{S} (\boldsymbol{I}_T \otimes \boldsymbol{X}) \boldsymbol{\Gamma}^\top (\boldsymbol{e}_t \otimes (\boldsymbol{X}^\top \boldsymbol{D}_t \boldsymbol{S}_t \boldsymbol{e}_i)) \boldsymbol{e}_t^\top$$

$$= - \boldsymbol{e}_j^\top \boldsymbol{H} \sum_{t=1}^T \mathrm{Tr}(\boldsymbol{S}_t \boldsymbol{D}_t) \boldsymbol{e}_t \boldsymbol{e}_t^\top + \boldsymbol{e}_j^\top \boldsymbol{H} \sum_{i=1}^n (\boldsymbol{I}_T \otimes \boldsymbol{e}_i^\top) \mathcal{D} \mathcal{S} (\boldsymbol{I}_T \otimes \boldsymbol{X}) \boldsymbol{\Gamma}^\top (\boldsymbol{I}_T \otimes \boldsymbol{X}^\top) \mathcal{D} \mathcal{S} (\boldsymbol{I}_T \otimes \boldsymbol{e}_i)$$

$$= - \boldsymbol{e}_j^\top \boldsymbol{H} \widetilde{\boldsymbol{K}}^\top \qquad\qquad\qquad \text{by (26).}$$

Next, we compute the second term in the right-hand side of (37). Using the expression of $\Delta_{ij}^{lt}$ in Lemma B.2,

$$\sum_{i,t} \Delta_{ij}^{it} \boldsymbol{e}_t^\top = -\sum_{i,t} ((\boldsymbol{e}_i^\top \boldsymbol{F}) \otimes \boldsymbol{e}_j^\top) \boldsymbol{\Gamma}^\top (\boldsymbol{e}_t \otimes (\boldsymbol{X}^\top \boldsymbol{D}_t \boldsymbol{S}_t \boldsymbol{e}_i)) \boldsymbol{e}_t^\top$$

$$= -\boldsymbol{e}_j^\top \underbrace{\sum_i ((\boldsymbol{e}_i^\top \boldsymbol{F}) \otimes \boldsymbol{I}_p) \boldsymbol{\Gamma}^\top (\boldsymbol{I}_T \otimes \boldsymbol{X}^\top) \mathcal{D} \mathcal{S} (\boldsymbol{I}_T \otimes \boldsymbol{e}_i)}_{\mathbf{\Upsilon}_1}.$$

The identity (18) then follows by substituting the above two expressions into (37).

To study $\mathbf{\Upsilon}_2$ in (19), we use a similar procedure. Using the mixed property of Kronecker product and the fact that the transpose of a scalar remains the same, we have

$$\sum_{j=1}^p \frac{\partial \boldsymbol{e}_j^\top \boldsymbol{H}}{\partial x_{ij}} = \sum_{j=1}^p \sum_{t=1}^T \frac{\partial \boldsymbol{e}_j^\top \widehat{\boldsymbol{b}}^t}{\partial x_{ij}} \boldsymbol{e}_t^\top = \sum_{j,t} \boldsymbol{e}_j^\top (\boldsymbol{e}_t^\top \otimes \boldsymbol{I}_p) \boldsymbol{\Gamma} ((\boldsymbol{F}^\top \boldsymbol{e}_i) \otimes \boldsymbol{e}_j) \boldsymbol{e}_t^\top$$

$$\qquad - \sum_{j,t} \boldsymbol{e}_j^\top (\boldsymbol{e}_t^\top \otimes \boldsymbol{I}_p) \boldsymbol{\Gamma} (\boldsymbol{I}_T \otimes \boldsymbol{X}^\top) \mathcal{S} \mathcal{D} ((\boldsymbol{H}^\top \boldsymbol{e}_j) \otimes \boldsymbol{e}_i) \boldsymbol{e}_t^\top \quad \text{by (15)}$$

$$= \boldsymbol{e}_i^\top \boldsymbol{F} \sum_j (\boldsymbol{I}_T \otimes \boldsymbol{e}_j^\top) \boldsymbol{\Gamma}^\top (\boldsymbol{I}_T \otimes \boldsymbol{e}_j)$$

$$\qquad - \sum_{j,t} ((\boldsymbol{e}_j^\top \boldsymbol{H}) \otimes \boldsymbol{e}_i^\top) \mathcal{D} \mathcal{S} (\boldsymbol{I}_T \otimes \boldsymbol{X}) \boldsymbol{\Gamma}^\top (\boldsymbol{e}_t \otimes \boldsymbol{e}_j) \boldsymbol{e}_t^\top$$

$$= \boldsymbol{e}_i^\top \boldsymbol{F} \sum_j (\boldsymbol{I}_T \otimes \boldsymbol{e}_j^\top) \boldsymbol{\Gamma}^\top (\boldsymbol{I}_T \otimes \boldsymbol{e}_j)$$

$$\qquad - \boldsymbol{e}_i^\top \sum_j ((\boldsymbol{e}_j^\top \boldsymbol{H}) \otimes \boldsymbol{I}_n) \mathcal{D} \mathcal{S} (\boldsymbol{I}_T \otimes \boldsymbol{X}) \boldsymbol{\Gamma}^\top (\boldsymbol{I}_T \otimes \boldsymbol{e}_j)$$

$$= \boldsymbol{e}_i^\top \boldsymbol{F} \boldsymbol{W}^\top - \boldsymbol{e}_i^\top \mathbf{\Upsilon}_2,$$

where $\boldsymbol{W} = \sum_j (\boldsymbol{I}_T \otimes \boldsymbol{e}_j^\top) \boldsymbol{\Gamma} (\boldsymbol{I}_T \otimes \boldsymbol{e}_j)$ and $\mathbf{\Upsilon}_2 = \sum_j ((\boldsymbol{e}_j^\top \boldsymbol{H}) \otimes \boldsymbol{I}_n) \mathcal{D} \mathcal{S} (\boldsymbol{I}_T \otimes \boldsymbol{X}) \boldsymbol{\Gamma}^\top (\boldsymbol{I}_T \otimes \boldsymbol{e}_j)$.

To study $\boldsymbol{\Upsilon}_3$ in (20), we use the product rule of differentiation and (18)-(19):

$$\sum_{i=1}^{n}\sum_{j=1}^{p}\frac{\partial \boldsymbol{F}^{\top}\boldsymbol{e}_i\boldsymbol{e}_j^{\top}\boldsymbol{H}}{\partial x_{ij}} = \sum_{i=1}^{n}\sum_{j=1}^{p}\frac{\partial \boldsymbol{F}^{\top}}{\partial x_{ij}}\boldsymbol{e}_i\boldsymbol{e}_j^{\top}\boldsymbol{H} + \boldsymbol{F}^{\top}\sum_{i=1}^{n}\sum_{j=1}^{p}\boldsymbol{e}_i\frac{\partial \boldsymbol{e}_j^{\top}\boldsymbol{H}}{\partial x_{ij}}$$

$$= -\sum_{j}(\widetilde{\boldsymbol{K}}\boldsymbol{H}^{\top}\boldsymbol{e}_j + \boldsymbol{\Upsilon}_1^{\top}\boldsymbol{e}_j)\boldsymbol{e}_j^{\top}\boldsymbol{H} + \boldsymbol{F}^{\top}\sum_{i}\boldsymbol{e}_i(\boldsymbol{e}_i^{\top}\boldsymbol{F}\boldsymbol{W}^{\top} - \boldsymbol{e}_i^{\top}\boldsymbol{\Upsilon}_2)$$

$$= -\widetilde{\boldsymbol{K}}\boldsymbol{H}^{\top}\boldsymbol{H} + \boldsymbol{F}^{\top}\boldsymbol{F}\boldsymbol{W}^{\top} - \underbrace{(\boldsymbol{\Upsilon}_1^{\top}\boldsymbol{H} + \boldsymbol{F}^{\top}\boldsymbol{\Upsilon}_2)}_{\boldsymbol{\Upsilon}_3}.$$

For $\boldsymbol{\Upsilon}_4$ in (21), by the product rule,

$$\sum_{i=1}^{n}\sum_{j=1}^{p}\frac{\partial \boldsymbol{H}^{\top}\boldsymbol{X}^{\top}\boldsymbol{e}_i\boldsymbol{e}_j^{\top}\boldsymbol{H}}{\partial x_{ij}}$$

$$= \sum_{i,j}\frac{\partial \boldsymbol{H}^{\top}}{\partial x_{ij}}\boldsymbol{X}^{\top}\boldsymbol{e}_i\boldsymbol{e}_j^{\top}\boldsymbol{H} + \boldsymbol{H}^{\top}\sum_{i,j}\boldsymbol{e}_j\boldsymbol{e}_i^{\top}\boldsymbol{e}_i\boldsymbol{e}_j^{\top}\boldsymbol{H} + \boldsymbol{H}^{\top}\boldsymbol{X}^{\top}\sum_{i,j}\boldsymbol{e}_i\boldsymbol{e}_j^{\top}\frac{\partial \boldsymbol{H}}{\partial x_{ij}}$$

$$= \sum_{i,j}\frac{\partial \boldsymbol{H}^{\top}}{\partial x_{ij}}\boldsymbol{X}^{\top}\boldsymbol{e}_i\boldsymbol{e}_j^{\top}\boldsymbol{H} + n\boldsymbol{H}^{\top}\boldsymbol{H} + \boldsymbol{H}^{\top}\boldsymbol{X}^{\top}(\boldsymbol{F}\boldsymbol{W}^{\top} - \boldsymbol{\Upsilon}_2) \qquad \text{by (19).}$$

We then compute the first term of the last line as follows

$$\sum_{i,j}\frac{\partial \boldsymbol{H}^{\top}}{\partial x_{ij}}\boldsymbol{X}^{\top}\boldsymbol{e}_i\boldsymbol{e}_j^{\top}\boldsymbol{H}$$

$$= \sum_{i,j,t}\boldsymbol{e}_t\boldsymbol{e}_t^{\top}\frac{\partial \boldsymbol{H}^{\top}}{\partial x_{ij}}\boldsymbol{X}^{\top}\boldsymbol{e}_i\boldsymbol{e}_j^{\top}\boldsymbol{H}$$

$$= \sum_{i,j,t}\boldsymbol{e}_t\Big(\frac{\partial \widehat{\boldsymbol{b}}^{t}}{\partial x_{ij}}\Big)^{\top}\boldsymbol{X}^{\top}\boldsymbol{e}_i\boldsymbol{e}_j^{\top}\boldsymbol{H}$$

$$= \sum_{i,j,t}\boldsymbol{e}_t\boldsymbol{e}_i^{\top}\boldsymbol{X}\frac{\partial \widehat{\boldsymbol{b}}^{t}}{\partial x_{ij}}\boldsymbol{e}_j^{\top}\boldsymbol{H}$$

$$= -\sum_{i,j,t}\boldsymbol{e}_t\boldsymbol{e}_i^{\top}\boldsymbol{X}(\boldsymbol{e}_t^{\top}\otimes \boldsymbol{I}_p)\boldsymbol{\Gamma}(\boldsymbol{I}_T\otimes \boldsymbol{X}^{\top})\mathcal{SD}((\boldsymbol{H}^{\top}\boldsymbol{e}_j)\otimes \boldsymbol{e}_i)\boldsymbol{e}_j^{\top}\boldsymbol{H} + \widetilde{\boldsymbol{\Upsilon}}_1 \qquad \text{by (15)}$$

$$= -\sum_{i}(\boldsymbol{I}_T\otimes(\boldsymbol{e}_i^{\top}\boldsymbol{X}))\boldsymbol{\Gamma}(\boldsymbol{I}_T\otimes \boldsymbol{X}^{\top})\mathcal{SD}(\boldsymbol{I}_T\otimes \boldsymbol{e}_i)\boldsymbol{H}^{\top}\boldsymbol{H} + \widetilde{\boldsymbol{\Upsilon}}_1$$

$$= -\widetilde{\boldsymbol{A}}\boldsymbol{H}^{\top}\boldsymbol{H} + \widetilde{\boldsymbol{\Upsilon}}_1 \qquad \text{by (23),}$$

where

$$\widetilde{\boldsymbol{\Upsilon}}_1 = \sum_{i,j,t}\boldsymbol{e}_t\boldsymbol{e}_i^{\top}\boldsymbol{X}(\boldsymbol{e}_t^{\top}\otimes \boldsymbol{I}_p)\boldsymbol{\Gamma}((\boldsymbol{F}^{\top}\boldsymbol{e}_i)\otimes \boldsymbol{e}_j)\boldsymbol{e}_j^{\top}\boldsymbol{H} = \sum_{i}(\boldsymbol{I}_T\otimes \boldsymbol{e}_i^{\top}\boldsymbol{X})\boldsymbol{\Gamma}((\boldsymbol{F}^{\top}\boldsymbol{e}_i)\otimes \boldsymbol{I}_p)\boldsymbol{H}.$$

Combining the above pieces, we have

$$\sum_{i=1}^{n}\sum_{j=1}^{p}\frac{\partial \boldsymbol{H}^{\top}\boldsymbol{X}^{\top}\boldsymbol{e}_i\boldsymbol{e}_j^{\top}\boldsymbol{H}}{\partial x_{ij}} = -\widetilde{\boldsymbol{A}}\boldsymbol{H}^{\top}\boldsymbol{H} + \widetilde{\boldsymbol{\Upsilon}}_1 + n\boldsymbol{H}^{\top}\boldsymbol{H} + \boldsymbol{H}^{\top}\boldsymbol{X}^{\top}(\boldsymbol{F}\boldsymbol{W}^{\top} - \boldsymbol{\Upsilon}_2)$$

$$= (n\boldsymbol{I}_T - \widetilde{\boldsymbol{A}})\boldsymbol{H}^{\top}\boldsymbol{H} + \boldsymbol{H}^{\top}\boldsymbol{X}^{\top}\boldsymbol{F}\boldsymbol{W}^{\top} - \underbrace{(\boldsymbol{H}^{\top}\boldsymbol{X}^{\top}\boldsymbol{\Upsilon}_2 - \widetilde{\boldsymbol{\Upsilon}}_1)}_{\boldsymbol{\Upsilon}_4}.$$

This provides an alternative expression for $\boldsymbol{\Upsilon}_4$ in (21).

Last, we study $\boldsymbol{\Upsilon}_5$ in (22). We have

$$\sum_{i=1}^n \sum_{j=1}^p \frac{\partial \boldsymbol{F}^\top \boldsymbol{e}_i \boldsymbol{e}_j^\top \boldsymbol{X}^\top \widetilde{\boldsymbol{F}}}{\partial x_{ij}} = \sum_{i,j} \Big[ \frac{\partial \boldsymbol{F}^\top \boldsymbol{e}_i}{\partial x_{ij}} \boldsymbol{e}_j^\top \boldsymbol{X}^\top \widetilde{\boldsymbol{F}} + \boldsymbol{F}^\top \boldsymbol{e}_i \boldsymbol{e}_j^\top \boldsymbol{e}_j \boldsymbol{e}_i^\top \widetilde{\boldsymbol{F}} + \boldsymbol{F}^\top \boldsymbol{e}_i \boldsymbol{e}_j^\top \boldsymbol{X}^\top \frac{\partial \widetilde{\boldsymbol{F}}}{\partial x_{ij}} \Big]$$

$$= -(\widetilde{\boldsymbol{K}} \boldsymbol{H}^\top + \boldsymbol{\Upsilon}_1^\top) \boldsymbol{X}^\top \widetilde{\boldsymbol{F}} + p \boldsymbol{F}^\top \widetilde{\boldsymbol{F}} + \sum_{i,j} \boldsymbol{F}^\top \boldsymbol{e}_i \boldsymbol{e}_j^\top \boldsymbol{X}^\top \frac{\partial \widetilde{\boldsymbol{F}}}{\partial x_{ij}}.$$

It remains to compute the third term in the last display. Using the fact that $\widetilde{\boldsymbol{F}} = \sum_{l=1}^n \sum_{t=1}^T \boldsymbol{e}_l \boldsymbol{e}_l^\top \widetilde{\boldsymbol{F}} \boldsymbol{e}_t \boldsymbol{e}_t^\top$, we have by Lemma B.3 that

$$\frac{\partial \widetilde{\boldsymbol{F}}}{\partial x_{ij}} = \sum_{l,t} \boldsymbol{e}_l \frac{\partial \boldsymbol{e}_l^\top \widetilde{\boldsymbol{F}} \boldsymbol{e}_t}{\partial x_{ij}} \boldsymbol{e}_t^\top = \sum_{l,t} \boldsymbol{e}_l (\tilde{D}_{ij}^{lt} + \tilde{\Delta}_{ij}^{lt}) \boldsymbol{e}_t^\top.$$

Using

$$\tilde{D}_{ij}^{lt} = -\boldsymbol{e}_l^\top \boldsymbol{D}_t \boldsymbol{e}_i \boldsymbol{e}_j^\top \boldsymbol{H} \boldsymbol{e}_t + ((\boldsymbol{e}_j^\top \boldsymbol{H}) \otimes \boldsymbol{e}_i^\top) \mathcal{D} \mathcal{S} (\boldsymbol{I}_T \otimes \boldsymbol{X}) \boldsymbol{\Gamma}^\top (\boldsymbol{I}_T \otimes \boldsymbol{X}^\top) \mathcal{D} (\boldsymbol{e}_t \otimes \boldsymbol{e}_l),$$
$$\tilde{\Delta}_{ij}^{lt} = -((\boldsymbol{e}_i^\top \boldsymbol{F}) \otimes \boldsymbol{e}_j^\top) \boldsymbol{\Gamma}^\top (\boldsymbol{I}_T \otimes \boldsymbol{X}^\top) \mathcal{D} (\boldsymbol{e}_t \otimes \boldsymbol{e}_l),$$

we find

$$\sum_{i,j} \boldsymbol{F}^\top \boldsymbol{e}_i \boldsymbol{e}_j^\top \boldsymbol{X}^\top \frac{\partial \widetilde{\boldsymbol{F}}}{\partial x_{ij}} = \sum_{i,j,l,t} \boldsymbol{F}^\top \boldsymbol{e}_i \boldsymbol{e}_j^\top \boldsymbol{X}^\top \boldsymbol{e}_l \frac{\partial \boldsymbol{e}_l^\top \widetilde{\boldsymbol{F}} \boldsymbol{e}_t}{\partial x_{ij}} \boldsymbol{e}_t^\top$$

$$= \sum_{i,j,l,t} \boldsymbol{F}^\top \boldsymbol{e}_i \boldsymbol{e}_j^\top \boldsymbol{X}^\top \boldsymbol{e}_l \tilde{\Delta}_{ij}^{lt} \boldsymbol{e}_t^\top + \sum_{i,j,l,t} \boldsymbol{F}^\top \boldsymbol{e}_i \boldsymbol{e}_j^\top \boldsymbol{X}^\top \boldsymbol{e}_l \tilde{D}_{ij}^{lt} \boldsymbol{e}_t^\top.$$

We now compute the two terms in the above display. For the first term, we have

$$\sum_{i,j,l,t} \boldsymbol{F}^\top \boldsymbol{e}_i \boldsymbol{e}_j^\top \boldsymbol{X}^\top \boldsymbol{e}_l \tilde{\Delta}_{ij}^{lt} \boldsymbol{e}_t^\top$$

$$= \sum_{i,j,l,t} \boldsymbol{F}^\top \boldsymbol{e}_i \boldsymbol{e}_j^\top \boldsymbol{X}^\top \boldsymbol{e}_l \big[ -((\boldsymbol{e}_i^\top \boldsymbol{F}) \otimes \boldsymbol{e}_j^\top) \boldsymbol{\Gamma}^\top (\boldsymbol{I}_T \otimes \boldsymbol{X}^\top) \mathcal{D} (\boldsymbol{e}_t \otimes \boldsymbol{e}_l) \big] \boldsymbol{e}_t^\top$$

$$= -\boldsymbol{F}^\top \boldsymbol{F} \sum_{l=1}^n (\boldsymbol{I}_T \otimes (\boldsymbol{e}_l^\top \boldsymbol{X})) \boldsymbol{\Gamma}^\top (\boldsymbol{I}_T \otimes \boldsymbol{X}^\top) \mathcal{D} (\boldsymbol{I}_T \otimes \boldsymbol{e}_l)$$

$$= -\boldsymbol{F}^\top \boldsymbol{F} \widehat{\boldsymbol{A}}^\top \qquad\qquad \text{by (24).}$$

For the second term, we have

$$\sum_{i,j,l,t} \boldsymbol{F}^\top \boldsymbol{e}_i \boldsymbol{e}_j^\top \boldsymbol{X}^\top \boldsymbol{e}_l \tilde{D}_{ij}^{lt} \boldsymbol{e}_t^\top$$

$$= \sum_{i,j,l,t} \boldsymbol{F}^\top \boldsymbol{e}_i \boldsymbol{e}_j^\top \boldsymbol{X}^\top \boldsymbol{e}_l [-\boldsymbol{e}_l^\top \boldsymbol{D}_t \boldsymbol{e}_i \boldsymbol{e}_j^\top \boldsymbol{H} \boldsymbol{e}_t + ((\boldsymbol{e}_j^\top \boldsymbol{H}) \otimes \boldsymbol{e}_i^\top) \mathcal{D} \mathcal{S} (\boldsymbol{I}_T \otimes \boldsymbol{X}) \boldsymbol{\Gamma}^\top (\boldsymbol{e}_t \otimes (\boldsymbol{X}^\top \boldsymbol{D}_t \boldsymbol{e}_l))] \boldsymbol{e}_t^\top$$

$$= -\underbrace{\sum_{t=1}^T \boldsymbol{F}^\top \boldsymbol{D}_t \boldsymbol{X} \boldsymbol{H} \boldsymbol{e}_t + \sum_{l=1}^n (\boldsymbol{e}_l^\top \boldsymbol{X} \boldsymbol{H} \otimes \boldsymbol{F}^\top) \mathcal{D} \mathcal{S} (\boldsymbol{I}_T \otimes \boldsymbol{X}) \boldsymbol{\Gamma}^\top (\boldsymbol{I}_T \otimes \boldsymbol{X}^\top) \mathcal{D} (\boldsymbol{I}_T \otimes \boldsymbol{e}_l)}_{\widetilde{\boldsymbol{\Upsilon}}_2}.$$

Thus, we have established

$$\sum_{i=1}^n \sum_{j=1}^p \frac{\partial \boldsymbol{F}^\top \boldsymbol{e}_i \boldsymbol{e}_j^\top \boldsymbol{X}^\top \widetilde{\boldsymbol{F}}}{\partial x_{ij}} = -\widetilde{\boldsymbol{K}} \boldsymbol{H}^\top \boldsymbol{X}^\top \widetilde{\boldsymbol{F}} + p \boldsymbol{F}^\top \widetilde{\boldsymbol{F}} - \boldsymbol{F}^\top \boldsymbol{F} \widehat{\boldsymbol{A}}^\top - \underbrace{(\boldsymbol{\Upsilon}_1^\top \boldsymbol{X}^\top \widetilde{\boldsymbol{F}} - \widetilde{\boldsymbol{\Upsilon}}_2)}_{\boldsymbol{\Upsilon}_5}.$$

This provides an alternative expression for $\boldsymbol{\Upsilon}_5$ in (22).

## D.5 Preparation results for proving Lemmas B.5 and B.6

**Lemma D.1** (Moment bounds for $\boldsymbol{H}, \boldsymbol{F}, \widetilde{\boldsymbol{F}}$). *Under Assumptions 3.1, 3.3 and 3.5 with $\boldsymbol{\Sigma} = \boldsymbol{I}_p$. Let $\boldsymbol{H}, \boldsymbol{F}$ be defined in* (13)*, we have for any finite integer $k$,*

$$\mathbb{E}[\|\boldsymbol{X}/\sqrt{n}\|_{\mathrm{op}}^{2k}] \leq C(\gamma, k),$$
$$\mathbb{E}[\|\boldsymbol{H}\|_{\mathrm{F}}^{2k} \mid \boldsymbol{\varepsilon}] \leq C(T, \gamma, c_0, \eta_{\max}, k)(\delta^2 + \|\boldsymbol{b}^*\|)^{2k},$$
$$\mathbb{E}[\|\boldsymbol{F}/\sqrt{n}\|_{\mathrm{F}}^{2k} \mid \boldsymbol{\varepsilon}] \leq \mathbb{E}[\|\widetilde{\boldsymbol{F}}/\sqrt{n}\|_{\mathrm{F}}^{2k} \mid \boldsymbol{\varepsilon}] \leq C(T, k)\delta^{2k}.$$

*Proof of Lemma D.1.* For the first inequality, according to Assumption 3.1 and $\boldsymbol{\Sigma} = \boldsymbol{I}_p$, $\boldsymbol{X}$ has i.i.d. standard normal entries. By [11, Theorem II.13], there exists a random variable $z \sim N(0, 1)$ such that $\|\boldsymbol{X}\|_{\mathrm{op}} \leq \sqrt{n} + \sqrt{p} + z$ almost surely. Under Assumption 3.5 that $p/n \leq \gamma$, we have $\mathbb{E}[\|\boldsymbol{X}/\sqrt{n}\|_{\mathrm{op}}^k] \leq C(\gamma, k)$ for any finite integer $k$.

For the second inequality, since $\|\boldsymbol{H}\|_{\mathrm{F}}^2 = \sum_{t=1}^{T} \|\widehat{\boldsymbol{b}}^t - \boldsymbol{b}^*\|^2$, it suffices to bound $\|\widehat{\boldsymbol{b}}^t - \boldsymbol{b}^*\|^2$ for each $t \in [T]$. Define the sequence of scalars $a_t \overset{\mathrm{def}}{=} \max\{\|\widehat{\boldsymbol{b}}^t\|, \delta\}$. Since $\widehat{\boldsymbol{b}}^t = \phi_{t-1}(\widehat{\boldsymbol{b}}^{t-1} + \frac{\eta_t}{n_t}\boldsymbol{X}^\top \boldsymbol{S}_t \boldsymbol{\psi}(\boldsymbol{y} - \boldsymbol{X}\widehat{\boldsymbol{b}}^{t-1}))$ where $n_t := |I_t|$. Note that $\|\boldsymbol{\psi}(\boldsymbol{y}_{I_t} - \boldsymbol{X}_{I_t}\boldsymbol{b})\| \leq \sqrt{|I_t|}\delta$ by Assumption 3.3, we have

$$\begin{aligned}
\|\widehat{\boldsymbol{b}}^t - \boldsymbol{0}\| = \|\widehat{\boldsymbol{b}}^t - \phi_{t-1}(\boldsymbol{0})\| && \text{since } \phi_{t-1} = \boldsymbol{0} \\
&\leq \|\widehat{\boldsymbol{b}}^{t-1} + \tfrac{\eta_t}{|I_t|}\boldsymbol{X}^\top \boldsymbol{S}_t \boldsymbol{\psi}(\boldsymbol{y} - \boldsymbol{X}\widehat{\boldsymbol{b}}^{t-1})\| && \text{since } \phi_t \text{ is 1-Lipschitz} \\
&= \|\widehat{\boldsymbol{b}}^{t-1} + \tfrac{\eta_t}{|I_t|}\boldsymbol{X}_{I_t}^\top \boldsymbol{\psi}(\boldsymbol{y}_{I_t} - \boldsymbol{X}_{I_t}\widehat{\boldsymbol{b}}^{t-1})\| \\
&\leq \|\widehat{\boldsymbol{b}}^{t-1}\| + \tfrac{\eta_t}{\sqrt{|I_t|}}\|\boldsymbol{X}_{I_t}\|_{\mathrm{op}}\delta && \text{by the triangle inequality} \\
&\leq \|\widehat{\boldsymbol{b}}^{t-1}\| + \tfrac{\eta_{\max}}{\sqrt{c_0 n}}\|\boldsymbol{X}\|_{\mathrm{op}}\delta && \text{by } |I_t| \geq c_0 n \\
&\leq a_{t-1} + \tfrac{\eta_{\max}}{\sqrt{c_0}}\|\boldsymbol{X}/\sqrt{n}\|_{\mathrm{op}}a_{t-1} \\
&= (1 + \tfrac{\eta_{\max}}{\sqrt{c_0}}\|\boldsymbol{X}/\sqrt{n}\|_{\mathrm{op}})a_{t-1}.
\end{aligned}$$

Since $a_t = \max\{\|\widehat{\boldsymbol{b}}^t\|, \delta\}$ and $\delta \leq a_{t-1}$, we have

$$a_t \leq (1 + \tfrac{\eta_{\max}}{\sqrt{c_0}}\|\boldsymbol{X}/\sqrt{n}\|_{\mathrm{op}})a_{t-1}.$$

Notice $a_1 = \delta$ since $\widehat{\boldsymbol{b}}^1 = \boldsymbol{0}_p$, we obtain

$$a_t \leq (1 + \tfrac{\eta_{\max}}{\sqrt{c_0}}\|\boldsymbol{X}/\sqrt{n}\|_{\mathrm{op}})^{t-1}\delta.$$

Hence, using the inequality $\|\widehat{\boldsymbol{b}}^t - \boldsymbol{b}^*\|^2 \leq 2\|\widehat{\boldsymbol{b}}^t\|^2 + 2\|\boldsymbol{b}^*\|^2$, we have

$$\begin{aligned}
\|\boldsymbol{H}\|_{\mathrm{F}}^2 &\lesssim \sum_{t=1}^{T}(\|\widehat{\boldsymbol{b}}^t\|^2 + \|\boldsymbol{b}^*\|^2) \\
&\leq \sum_{t=1}^{T}\left[(1 + \tfrac{\eta_{\max}}{\sqrt{c_0}}\|\boldsymbol{X}/\sqrt{n}\|_{\mathrm{op}})^{2t-2}\delta^2 + \|\boldsymbol{b}^*\|^2\right] \\
&\leq T(\delta^2 + \|\boldsymbol{b}^*\|^2)(1 + \tfrac{\eta_{\max}}{\sqrt{c_0}}\|\boldsymbol{X}/\sqrt{n}\|_{\mathrm{op}})^{2T}. && (38)
\end{aligned}$$

Taking conditional expectation on both sides given $\boldsymbol{\varepsilon}$, the desired moment bound for $\boldsymbol{H}$ follows from the moment bound for $\|\boldsymbol{X}/\sqrt{n}\|_{\mathrm{op}}$.

For the third inequality, since $|\psi(x)| \leq \delta$ from Assumption 3.3, we have $\|\boldsymbol{\psi}(\boldsymbol{u})\| \leq n\delta^2$ for any $\boldsymbol{u} \in \mathbb{R}^n$. By the definitions of $\boldsymbol{F}$ and $\widetilde{\boldsymbol{F}}$, we have

$$\|\boldsymbol{F}\|_{\mathrm{F}}^2 \leq \|\widetilde{\boldsymbol{F}}\|_{\mathrm{F}}^2 = \sum_{t=1}^{T}\|\boldsymbol{\psi}(\boldsymbol{y} - \boldsymbol{X}\widehat{\boldsymbol{b}}^t)\|^2 \leq Tn\delta^2.$$

Since the above display holds for any $\boldsymbol{\varepsilon}$, it implies the desired conditional moment bounds for $\boldsymbol{F}, \widetilde{\boldsymbol{F}}$. $\qquad\square$

**Lemma D.2** (Frobenius norm bound for $\boldsymbol{XH}$). *Under Assumptions 3.1, 3.3 and 3.5, we have*

$$\|\boldsymbol{XH}\|_{\mathrm{F}}^2 \leq C(T, \gamma, \eta_{\max}, c_0) n (\delta^2 + \|\boldsymbol{b}^*\|^2)$$

*with probability at least* $1 - \exp(-n/2)$.

*Proof of Lemma D.2.* By [11, Theorem II.13], under Assumption 3.1 with $\boldsymbol{\Sigma} = \boldsymbol{I}_p$, we have

$$\mathbb{P}(\|\boldsymbol{X}/\sqrt{n}\|_{\mathrm{op}} \leq 2 + \sqrt{\gamma}) \geq 1 - \exp(-n/2).$$

Using $\|\boldsymbol{XH}\|_{\mathrm{F}}^2 \leq \|\boldsymbol{X}\|_{\mathrm{op}}^2 \|\boldsymbol{H}\|_{\mathrm{F}}^2$ and the bound (38), we have

$$\|\boldsymbol{XH}\|_{\mathrm{F}}^2 \leq n C(T, \gamma, \eta_{\max}, c_0)(\delta^2 + \|\boldsymbol{b}^*\|^2)$$

holds with probability at least $1 - \exp(-n/2)$. $\qquad\square$

**Lemma D.3** (Operator norm bound for $\mathcal{M}$). *Under Assumptions 3.1, 3.3 and 3.5, we have*

$$\|\mathcal{M}^{-1}\|_{\mathrm{op}} \leq C(T)(1 + \xi)^T,$$

*where* $\xi = \frac{\eta_{\max}}{c_0 n} \|\boldsymbol{X}\|_{\mathrm{op}}^2$.

*Proof of Lemma D.3.* By the definition of $\mathcal{M}$ in Lemma B.1, we have

$$\mathcal{M} = \begin{bmatrix} \boldsymbol{I}_p & & & \\ -\boldsymbol{P}_1 & \boldsymbol{I}_p & & \\ & \ddots & \ddots & \\ & & -\boldsymbol{P}_{T-1} & \boldsymbol{I}_p \end{bmatrix} \quad \text{where } \boldsymbol{P}_t = \widetilde{\boldsymbol{D}}_t(\boldsymbol{I}_p - \tfrac{\eta_t}{|I_t|}\boldsymbol{X}^\top \boldsymbol{S}_t \boldsymbol{D}_t \boldsymbol{X}).$$

Hence, we can write $\mathcal{M} = \boldsymbol{I}_{pT} - \boldsymbol{A}$, where $\boldsymbol{A}$ is the lower triangular matrix with off-diagonal blocks $\boldsymbol{P}_1, ..., \boldsymbol{P}_{T-1}$. Using the matrix identity $(\boldsymbol{I} - \boldsymbol{A})^{-1} = \sum_{k=0}^{\infty} \boldsymbol{A}^k$ and noticing $\boldsymbol{A}^k = \boldsymbol{0}$ for $k \geq T$, we have

$$\mathcal{M}^{-1} = \sum_{k=0}^{T-1} \begin{bmatrix} \boldsymbol{0} & & & \\ \boldsymbol{P}_1 & \boldsymbol{0} & & \\ & \ddots & \ddots & \\ & & \boldsymbol{P}_{T-1} & \boldsymbol{0} \end{bmatrix}^k.$$

Taking operator norm on both sides, we obtain

$$\|\mathcal{M}^{-1}\|_{\mathrm{op}} \leq \sum_{k=0}^{T-1} \left(\sum_{t=1}^{T-1} \|\boldsymbol{P}_t\|_{\mathrm{op}}\right)^k. \tag{39}$$

Since $\boldsymbol{D}_t = \frac{\partial \boldsymbol{\psi}(\boldsymbol{u})}{\partial \boldsymbol{u}}\big|_{\boldsymbol{u} = \boldsymbol{y} - \boldsymbol{X}\widehat{\boldsymbol{b}}^t}$ and $\psi$ is 1-Lipschitz, we have $\|\boldsymbol{D}_t\|_{\mathrm{op}} \leq 1$. By the definition that $\boldsymbol{S}_t = \sum_{i \in I_t} \boldsymbol{e}_i \boldsymbol{e}_i^\top$, we know $\|\boldsymbol{S}_t\|_{\mathrm{op}} \leq 1$. Since $|I_t| \geq c_0 n$ and $\eta_t \leq \eta_{\max}$ for any $t \in [T]$, we have

$$\|\boldsymbol{P}_t\|_{\mathrm{op}} \leq 1 + \frac{\eta_t}{|I_t|} \|\boldsymbol{X}_{I_t}\|_{\mathrm{op}}^2 \leq 1 + \frac{\eta_{\max}}{c_0 n} \|\boldsymbol{X}\|_{\mathrm{op}}^2 \overset{\text{def}}{=} 1 + \xi.$$

Plugging this inequality into (39) gives

$$\|\mathcal{M}^{-1}\|_{\mathrm{op}} \leq \sum_{k=0}^{T-1} (T(1 + \xi))^k \leq C(T)(1 + \xi)^T.$$

$\qquad\square$

**Lemma D.4.** *Under the same conditions as Lemma D.3, we have*

$$\|\boldsymbol{\Gamma}\|_{\mathrm{op}} \leq n^{-1} C(T, \eta_{\max}, c_0)(1 + \xi)^T,$$

*where* $\xi = \frac{\eta_{\max}}{c_0 n} \|\boldsymbol{X}\|_{\mathrm{op}}^2$.

*Proof of Lemma D.4.* By the definition of $\boldsymbol{\Gamma}$ in Lemma B.1, we have $\boldsymbol{\Gamma} = \mathcal{M}^{-1}\boldsymbol{L}(\boldsymbol{\Lambda}\otimes\boldsymbol{I}_p)\widetilde{\mathcal{D}}$. Notice that $\boldsymbol{\Lambda} = \sum_{t=1}^{T}\frac{\eta_t}{|I_t|}\boldsymbol{e}_t\boldsymbol{e}_t^\top$, we have $\|\boldsymbol{\Lambda}\|_{\mathrm{op}} = \max_{t\in[T]}\frac{\eta_t}{|I_t|} \leq n^{-1}\frac{\eta_{\max}}{c_0}$ using $|I_t| \geq c_0 n$ and $\eta_t \leq \eta_{\max}$. Since $\phi$ is 1-Lipschitz, we have $\|\widetilde{\mathcal{D}}\|_{\mathrm{op}} \leq 1$. By definition of $\boldsymbol{L}$ in Lemma B.1, we have $\|\boldsymbol{L}\|_{\mathrm{op}} = 1$. Using these upper bounds of $\|\boldsymbol{L}\|_{\mathrm{op}}, \|\boldsymbol{\Lambda}\|_{\mathrm{op}}, \|\widetilde{\mathcal{D}}\|_{\mathrm{op}}$ and the upper bound of $\|\mathcal{M}^{-1}\|_{\mathrm{op}}$ in Lemma D.3, we obtain

$$\|\boldsymbol{\Gamma}\|_{\mathrm{op}} \leq \|\mathcal{M}^{-1}\|_{\mathrm{op}}\|\boldsymbol{L}\|_{\mathrm{op}}\|\boldsymbol{\Lambda}\otimes\boldsymbol{I}_p\|_{\mathrm{op}}\|\widetilde{\mathcal{D}}\|_{\mathrm{op}}$$
$$\leq n^{-1}C(T, \eta_{\max}, c_0)(1+\xi)^T.$$

$\square$

**Lemma D.5** (Moment bounds for derivative of $\boldsymbol{H}, \boldsymbol{F}, \widetilde{\boldsymbol{F}}$). *Under Assumptions 3.1, 3.3 and 3.5 and $\boldsymbol{\Sigma} = \boldsymbol{I}_p$, we have for any finite integer $k$,*

$$\mathbb{E}\Big[\Big(\sum_{i=1}^{n}\sum_{j=1}^{p}\Big\|\frac{\partial\boldsymbol{H}}{\partial x_{ij}}\Big\|_{\mathrm{F}}^2\Big)^k \;\Big|\; \boldsymbol{\varepsilon}\Big] \leq C(T, \gamma, \eta_{\max}, c_0)(\delta^2 + \|\boldsymbol{b}^*\|^2)^{2k},$$

$$\mathbb{E}\Big[\Big(\sum_{i=1}^{n}\sum_{j=1}^{p}\Big\|\frac{\partial\boldsymbol{F}/\sqrt{n}}{\partial x_{ij}}\Big\|_{\mathrm{F}}^2\Big)^k \;\Big|\; \boldsymbol{\varepsilon}\Big] \leq \mathbb{E}\Big[\Big(\sum_{i=1}^{n}\sum_{j=1}^{p}\Big\|\frac{\partial\widetilde{\boldsymbol{F}}/\sqrt{n}}{\partial x_{ij}}\Big\|_{\mathrm{F}}^2\Big)^k \;\Big|\; \boldsymbol{\varepsilon}\Big] \leq C(T, \gamma, \eta_{\max}, c_0)(\delta^2 + \|\boldsymbol{b}^*\|^2)^{2k}.$$

*Proof of Lemma D.5.* We first prove the first bound. By Lemma B.1, we have

$$\frac{\partial\widehat{\boldsymbol{b}}^t}{\partial x_{ij}} = (\boldsymbol{e}_t^\top \otimes \boldsymbol{I}_p)\boldsymbol{\Gamma}[((\boldsymbol{F}^\top\boldsymbol{e}_i)\otimes\boldsymbol{e}_j) - (\boldsymbol{I}_T\otimes\boldsymbol{X}^\top)\mathcal{S}\mathcal{D}((\boldsymbol{H}^\top\boldsymbol{e}_j)\otimes\boldsymbol{e}_i)].$$

Hence, using $\frac{\partial\boldsymbol{e}_k^\top\boldsymbol{H}\boldsymbol{e}_t}{\partial x_{ij}} = \frac{\partial\boldsymbol{e}_k^\top\widehat{\boldsymbol{b}}^t}{\partial x_{ij}}$, we have

$$\frac{\partial\boldsymbol{e}_k^\top\boldsymbol{H}\boldsymbol{e}_t}{\partial x_{ij}} = (\boldsymbol{e}_t^\top \otimes \boldsymbol{e}_k^\top)\boldsymbol{\Gamma}[((\boldsymbol{F}^\top\boldsymbol{e}_i)\otimes\boldsymbol{e}_j) - (\boldsymbol{I}_T\otimes\boldsymbol{X}^\top)\mathcal{S}\mathcal{D}((\boldsymbol{H}^\top\boldsymbol{e}_j)\otimes\boldsymbol{e}_i)]$$
$$= (\boldsymbol{e}_t^\top \otimes \boldsymbol{e}_k^\top)\boldsymbol{\Gamma}[(\boldsymbol{F}^\top\otimes\boldsymbol{I}_p)(\boldsymbol{e}_i\otimes\boldsymbol{e}_j) - (\boldsymbol{I}_T\otimes\boldsymbol{X}^\top)\mathcal{S}\mathcal{D}(\boldsymbol{H}^\top\otimes\boldsymbol{I}_n)(\boldsymbol{e}_j\otimes\boldsymbol{e}_i)].$$

Using the above equality, $\sum_{i,j,t,k}[(\boldsymbol{e}_t^\top\otimes\boldsymbol{e}_k^\top)\boldsymbol{A}(\boldsymbol{e}_j\otimes\boldsymbol{e}_i)]^2 = \|\boldsymbol{A}\|_{\mathrm{F}}^2$ for $\boldsymbol{A}\in\mathbb{R}^{pT\times np}$, and the triangle inequality, we have

$$\sum_{i=1}^{n}\sum_{j=1}^{p}\Big\|\frac{\partial\boldsymbol{H}}{\partial x_{ij}}\Big\|_{\mathrm{F}}^2 = \sum_{i=1}^{n}\sum_{j=1}^{p}\sum_{k=1}^{p}\sum_{t=1}^{T}\Big(\frac{\partial\boldsymbol{e}_k^\top\boldsymbol{H}\boldsymbol{e}_t}{\partial x_{ij}}\Big)^2$$
$$\lesssim \|\boldsymbol{\Gamma}(\boldsymbol{F}^\top\otimes\boldsymbol{I}_p)\|_{\mathrm{F}}^2 + \|\boldsymbol{\Gamma}(\boldsymbol{I}_T\otimes\boldsymbol{X}^\top)\mathcal{S}\mathcal{D}(\boldsymbol{H}^\top\otimes\boldsymbol{I}_n)\|_{\mathrm{F}}^2$$
$$\leq p\|\boldsymbol{\Gamma}\|_{\mathrm{op}}^2\|\boldsymbol{F}\|_{\mathrm{F}}^2 + n\|\boldsymbol{\Gamma}\|_{\mathrm{op}}^2\|\boldsymbol{X}\|_{\mathrm{op}}^2\|\mathcal{S}\|_{\mathrm{op}}^2\|\mathcal{D}\|_{\mathrm{op}}^2\|\boldsymbol{H}\|_{\mathrm{F}}^2$$
$$\leq p\|\boldsymbol{\Gamma}\|_{\mathrm{op}}^2\|\boldsymbol{F}\|_{\mathrm{F}}^2 + n\|\boldsymbol{\Gamma}\|_{\mathrm{op}}^2\|\boldsymbol{X}\|_{\mathrm{op}}^2\|\boldsymbol{H}\|_{\mathrm{F}}^2$$
$$\leq C(T, \gamma, \eta_{\max}, c_0)(1+\xi)^{2T}\|\boldsymbol{F}\|_{\mathrm{F}}^2/n + C(T, \gamma, \eta_{\max}, c_0)(1+\xi)^{2T}\|\boldsymbol{X}\|_{\mathrm{op}}^2/n\|\boldsymbol{H}\|_{\mathrm{F}}^2,$$

where the last inequality uses Lemma D.4. Taking the conditional expectation on both sides given $\boldsymbol{\varepsilon}$, the desired moment bound follows from the moment bounds of $\boldsymbol{X}, \boldsymbol{H}, \boldsymbol{F}$ in Lemma D.1.

Now we prove the second bound. By definition, the $t$-th column of $\boldsymbol{F}$ is $F_t = \boldsymbol{S}_t\boldsymbol{\psi}(\boldsymbol{y} - \boldsymbol{X}\widehat{\boldsymbol{b}}^t)$, it can be written using the $t$-th column of $\widetilde{\boldsymbol{F}}$ as $F_t = \boldsymbol{S}_t\tilde{F}_t$. Since $\|\boldsymbol{S}_t\|_{\mathrm{op}} \leq 1$, we have

$$\|\frac{\partial\boldsymbol{F}}{\partial x_{ij}}\|_{\mathrm{F}}^2 = \sum_t \|\frac{\partial F_t}{\partial x_{ij}}\|^2 \leq \sum_t \|\frac{\partial\tilde{F}_t}{\partial x_{ij}}\|^2 = \|\frac{\partial\widetilde{\boldsymbol{F}}}{\partial x_{ij}}\|_{\mathrm{F}}^2.$$

By Lemma B.3, we have $\frac{\partial\boldsymbol{e}_l^\top\widetilde{\boldsymbol{F}}\boldsymbol{e}_t}{\partial x_{ij}} = \tilde{D}_{ij}^{lt} + \tilde{\Delta}_{ij}^{lt}$ where

$$\tilde{D}_{ij}^{lt} = -\boldsymbol{e}_l^\top\boldsymbol{D}_t\boldsymbol{e}_i\boldsymbol{e}_j^\top\boldsymbol{H}\boldsymbol{e}_t + ((\boldsymbol{e}_j^\top\boldsymbol{H})\otimes\boldsymbol{e}_i^\top)\mathcal{D}\mathcal{S}(\boldsymbol{I}_T\otimes\boldsymbol{X})\boldsymbol{\Gamma}^\top(\boldsymbol{I}_T\otimes\boldsymbol{X}^\top)\mathcal{D}(\boldsymbol{e}_t\otimes\boldsymbol{e}_l),$$
$$= ((\boldsymbol{e}_j^\top\boldsymbol{H})\otimes\boldsymbol{e}_i^\top)[-\boldsymbol{I}_{pT} + \mathcal{D}\mathcal{S}(\boldsymbol{I}_T\otimes\boldsymbol{X})\boldsymbol{\Gamma}^\top(\boldsymbol{I}_T\otimes\boldsymbol{X}^\top)]\mathcal{D}(\boldsymbol{e}_t\otimes\boldsymbol{e}_l),$$
$$\tilde{\Delta}_{ij}^{lt} = -((\boldsymbol{e}_i^\top\boldsymbol{F})\otimes\boldsymbol{e}_j^\top)\boldsymbol{\Gamma}^\top(\boldsymbol{I}_T\otimes\boldsymbol{X}^\top)\mathcal{D}(\boldsymbol{e}_t\otimes\boldsymbol{e}_l).$$

Using $\sum_{i,j,t,k}[(e_j^\top \otimes e_i^\top)A(e_t \otimes e_l)]^2 = \|A\|_{\mathrm{F}}^2$ for $A \in \mathbb{R}^{np \times nT}$ and the triangle inequality, we have

$$\sum_{i=1}^n \sum_{j=1}^p \|\frac{\partial \widetilde{F}}{\partial x_{ij}}\|_{\mathrm{F}}^2$$

$$= \sum_{i=1}^n \sum_{j=1}^p \sum_{l=1}^n \sum_{t=1}^T (\tilde{D}_{ij}^{lt} + \tilde{\Delta}_{ij}^{lt})^2$$

$$\lesssim \|(H \otimes I_n)[-I_{pT} + \mathcal{D}\mathcal{S}(I_T \otimes X)\Gamma^\top(I_T \otimes X^\top)]\mathcal{D}\|_{\mathrm{F}}^2 + \|(F \otimes I_p)\Gamma^\top(I_T \otimes X^\top)\mathcal{D}\|_{\mathrm{F}}^2$$

$$\lesssim n\|H\|_{\mathrm{F}}^2(1 + \|X\|_{\mathrm{op}}^4\|\Gamma\|_{\mathrm{op}}^2) + \|F\|_{\mathrm{F}}^2\|\Gamma\|_{\mathrm{op}}^2\|X\|_{\mathrm{op}}^2,$$

where the last inequality uses $\|\mathcal{D}\|_{\mathrm{op}} \leq 1$ and $\|\mathcal{S}\|_{\mathrm{op}} \leq 1$. Taking the conditional expectation on both sides given $\varepsilon$, the desired moment bound follows from the bound of $\Gamma$ in Lemma D.4 and the moment bounds of $X, H, F$ in Lemma D.1. $\qquad\square$

## D.6 Proof of Lemma B.5

**Bound of $\Upsilon_1$.** By the expression for the matrix $\Upsilon_1 \in \mathbb{R}^{p \times T}$ obtained in Appendix D.4,

$$\Upsilon_1 = \sum_{i=1}^n ((e_i^\top F) \otimes I_p)\Gamma^\top(I_T \otimes X^\top)\mathcal{D}\mathcal{S}(I_T \otimes e_i)$$

$$= \sum_{i=1}^n \sum_{t=1}^T ((e_i^\top F e_t e_t^\top) \otimes I_p)\Gamma^\top(I_T \otimes X^\top)\mathcal{D}\mathcal{S}(I_T \otimes e_i)$$

$$= \sum_{i=1}^n \sum_{t=1}^T (e_t^\top \otimes I_p)\Gamma^\top(I_T \otimes X^\top)\mathcal{D}\mathcal{S}(I_T \otimes e_i)e_i^\top F e_t$$

$$= \sum_{t=1}^T (e_t^\top \otimes I_p)\Gamma^\top(I_T \otimes X^\top)\mathcal{D}\mathcal{S}(I_T \otimes (F e_t)).$$

By the triangle inequality and $\|\mathcal{D}\|_{\mathrm{op}} \vee \|\mathcal{S}\|_{\mathrm{op}} \leq 1$, $\|\Upsilon_1\|_{\mathrm{op}} \leq T\|\Gamma\|_{\mathrm{op}}\|X\|_{\mathrm{op}}\|F\|_{\mathrm{F}}$. Using the bound of $\|\Gamma\|_{\mathrm{op}}$ in Lemma D.4, the moment bound of $\|X\|_{\mathrm{op}}, \|F\|_{\mathrm{F}}$ in Lemma D.1 gives

$$\mathbb{E}[\|\Upsilon_1\|_{\mathrm{op}}^2 \mid \varepsilon] \leq C(T, \gamma, \eta_{\max}, c_0)(\delta^2 + \|b^*\|^2).$$

**Bound of $\Upsilon_2$.** By the expression for the matrix $\Upsilon_2 \in \mathbb{R}^{n \times T}$ obtained in Appendix D.4, we have

$$\Upsilon_2 = \sum_{j=1}^p ((e_j^\top H) \otimes I_n)\mathcal{D}\mathcal{S}(I_T \otimes X)\Gamma^\top(I_T \otimes e_j)$$

$$= \sum_{j=1}^p \sum_{t=1}^T ((e_j^\top H e_t e_t^\top) \otimes I_n)\mathcal{D}\mathcal{S}(I_T \otimes X)\Gamma^\top(I_T \otimes e_j)$$

$$= \sum_{j=1}^p \sum_{t=1}^T (e_t^\top \otimes I_n)\mathcal{D}\mathcal{S}(I_T \otimes X)\Gamma^\top(I_T \otimes e_j)e_j^\top H e_t$$

$$= \sum_{t=1}^T (e_t^\top \otimes I_n)\mathcal{D}\mathcal{S}(I_T \otimes X)\Gamma^\top(I_T \otimes (H e_t)).$$

By the triangle inequality and $\|\mathcal{D}\|_{\mathrm{op}} \vee \|\mathcal{S}\|_{\mathrm{op}} \leq 1$, $\|\Upsilon_2\|_{\mathrm{op}} \leq T\|\Gamma\|_{\mathrm{op}}\|X\|_{\mathrm{op}}\|H\|_{\mathrm{F}}$. Similar to the moment bound of $\|\Upsilon_1\|_{\mathrm{op}}$, we obtain

$$\mathbb{E}[\|\Upsilon_2\|_{\mathrm{op}}^2 \mid \varepsilon] \leq C(T, \gamma, \eta_{\max}, c_0)n^{-1}(\delta^2 + \|b^*\|^2).$$

**Bound of $\boldsymbol{\Upsilon}_3$.** By the expression for the matrix $\boldsymbol{\Upsilon}_3 \in \mathbb{R}^{T \times T}$ obtained in Appendix D.4, we have $\boldsymbol{\Upsilon}_3 = (\boldsymbol{\Upsilon}_1^\top \boldsymbol{H} + \boldsymbol{F}^\top \boldsymbol{\Upsilon}_2)$. It directly follows that

$$\|\boldsymbol{\Upsilon}_3\|_{\mathrm{op}} \leq \|\boldsymbol{\Upsilon}_1\|_{\mathrm{op}} \|\boldsymbol{H}\|_{\mathrm{F}} + \|\boldsymbol{F}\|_{\mathrm{F}} \|\boldsymbol{\Upsilon}_2\|_{\mathrm{op}}.$$

Using the triangle inequality and the moment bounds of $\|\boldsymbol{H}\|_{\mathrm{F}}, \|\boldsymbol{F}\|_{\mathrm{F}}$ in Lemma D.1 and the moment bounds of $\|\boldsymbol{\Upsilon}_1\|_{\mathrm{op}}, \|\boldsymbol{\Upsilon}_2\|_{\mathrm{op}}$ we just obtained, we have

$$\mathbb{E}[\|\boldsymbol{\Upsilon}_3\|_{\mathrm{op}}^2 \mid \boldsymbol{\varepsilon}] \leq C(T, \gamma, \eta_{\max}, c_0)(\delta^2 + \|\boldsymbol{b}^*\|^2).$$

**Bound of $\boldsymbol{\Upsilon}_4$.** By the expression for the matrix $\boldsymbol{\Upsilon}_4 \in \mathbb{R}^{T \times T}$ obtained in Appendix D.4, we have

$$\boldsymbol{\Upsilon}_4 = \boldsymbol{H}^\top \boldsymbol{X}^\top \boldsymbol{\Upsilon}_2 - \widetilde{\boldsymbol{\Upsilon}}_1,$$

where $\widetilde{\boldsymbol{\Upsilon}}_1 = \sum_i (\boldsymbol{I}_T \otimes \boldsymbol{e}_i^\top \boldsymbol{X}) \boldsymbol{\Gamma}((\boldsymbol{F}^\top \boldsymbol{e}_i) \otimes \boldsymbol{I}_p) \boldsymbol{H}$. We can rewrite $\widetilde{\boldsymbol{\Upsilon}}_1$ as

$$\begin{aligned}
\widetilde{\boldsymbol{\Upsilon}}_1 &= \sum_{i=1}^n (\boldsymbol{I}_T \otimes \boldsymbol{e}_i^\top \boldsymbol{X}) \boldsymbol{\Gamma}((\boldsymbol{F}^\top \boldsymbol{e}_i) \otimes \boldsymbol{I}_p) \boldsymbol{H} \\
&= \sum_{i=1}^n \sum_{t=1}^T (\boldsymbol{I}_T \otimes \boldsymbol{e}_i^\top \boldsymbol{X}) \boldsymbol{\Gamma}((\boldsymbol{e}_t \boldsymbol{e}_t^\top \boldsymbol{F}^\top \boldsymbol{e}_i) \otimes \boldsymbol{I}_p) \boldsymbol{H} \\
&= \sum_{i=1}^n \sum_{t=1}^T (\boldsymbol{I}_T \otimes (\boldsymbol{e}_t^\top \boldsymbol{F}^\top \boldsymbol{e}_i \boldsymbol{e}_i^\top \boldsymbol{X})) \boldsymbol{\Gamma}(\boldsymbol{e}_t \otimes \boldsymbol{I}_p) \boldsymbol{H} \\
&= \sum_{t=1}^T (\boldsymbol{I}_T \otimes (\boldsymbol{e}_t^\top \boldsymbol{F}^\top \boldsymbol{X})) \boldsymbol{\Gamma}(\boldsymbol{e}_t \otimes \boldsymbol{I}_p) \boldsymbol{H}.
\end{aligned}$$

We have by the triangle inequality,

$$\|\widetilde{\boldsymbol{\Upsilon}}_1\|_{\mathrm{op}} \leq T \|\boldsymbol{\Gamma}\|_{\mathrm{op}} \|\boldsymbol{X}\|_{\mathrm{op}} \|\boldsymbol{F}\|_{\mathrm{F}} \|\boldsymbol{H}\|_{\mathrm{F}}.$$

By the triangle inequality and the upper bound of $\|\boldsymbol{\Upsilon}_2\|_{\mathrm{op}}$, we have

$$\|\boldsymbol{\Upsilon}_4\|_{\mathrm{op}} \leq \|\boldsymbol{\Upsilon}_2\|_{\mathrm{op}} \|\boldsymbol{X}\|_{\mathrm{op}} \|\boldsymbol{H}\|_{\mathrm{F}} + \|\widetilde{\boldsymbol{\Upsilon}}_1\|_{\mathrm{op}} \leq T \|\boldsymbol{\Gamma}\|_{\mathrm{op}} \|\boldsymbol{X}\|_{\mathrm{op}}^2 \|\boldsymbol{H}\|_{\mathrm{F}}^2 + T \|\boldsymbol{\Gamma}\|_{\mathrm{op}} \|\boldsymbol{X}\|_{\mathrm{op}} \|\boldsymbol{F}\|_{\mathrm{F}} \|\boldsymbol{H}\|_{\mathrm{F}}.$$

Squaring both sides and taking conditional expectation, we have

$$\mathbb{E}[\|\boldsymbol{\Upsilon}_4\|_{\mathrm{op}}^2 \mid \boldsymbol{\varepsilon}] \leq C(T, \gamma, \eta_{\max}, c_0)(\delta^2 + \|\boldsymbol{b}^*\|^2),$$

thanks to the upper bound of $\|\boldsymbol{\Gamma}\|_{\mathrm{op}}$ in Lemma D.4 and the moment bounds of $\|\boldsymbol{X}\|_{\mathrm{op}}, \|\boldsymbol{F}\|_{\mathrm{F}}, \|\boldsymbol{H}\|_{\mathrm{F}}$ in Lemma D.1.

**Bound of $\boldsymbol{\Upsilon}_5$.** By the expression for the matrix $\boldsymbol{\Upsilon}_5 \in \mathbb{R}^{T \times T}$ obtained in Appendix D.4, we have $\boldsymbol{\Upsilon}_5 = \boldsymbol{\Upsilon}_1^\top \boldsymbol{X}^\top \widetilde{\boldsymbol{F}} - \widetilde{\boldsymbol{\Upsilon}}_2$, where

$$\widetilde{\boldsymbol{\Upsilon}}_2 = -\sum_{t=1}^T \boldsymbol{F}^\top \boldsymbol{D}_t \boldsymbol{X} \boldsymbol{H} \boldsymbol{e}_t + \sum_{l=1}^n (\boldsymbol{e}_l^\top \boldsymbol{X} \boldsymbol{H} \otimes \boldsymbol{F}^\top) \mathcal{D} \mathcal{S}(\boldsymbol{I}_T \otimes \boldsymbol{X}) \boldsymbol{\Gamma}^\top (\boldsymbol{I}_T \otimes \boldsymbol{X}^\top) \mathcal{D}(\boldsymbol{I}_T \otimes \boldsymbol{e}_l).$$

By the triangle inequality,

$$\|\widetilde{\boldsymbol{\Upsilon}}_2\|_{\mathrm{op}} \leq T \|\boldsymbol{X}\|_{\mathrm{op}} \|\boldsymbol{F}\|_{\mathrm{F}} \|\boldsymbol{H}\|_{\mathrm{F}} + \|\boldsymbol{X}\|_{\mathrm{op}}^3 \|\boldsymbol{\Gamma}\|_{\mathrm{op}} \|\boldsymbol{F}\|_{\mathrm{F}} \|\boldsymbol{H}\|_{\mathrm{F}}.$$

Using the moment bounds of $\|\boldsymbol{\Upsilon}_1\|_{\mathrm{op}}, \|\boldsymbol{X}\|_{\mathrm{op}}, \|\boldsymbol{F}\|_{\mathrm{F}}, \|\widetilde{\boldsymbol{F}}\|_{\mathrm{F}}, \|\boldsymbol{H}\|_{\mathrm{F}}$, we have

$$\mathbb{E}[\|\boldsymbol{\Upsilon}_5\|_{\mathrm{op}}^2 \mid \boldsymbol{\varepsilon}] \leq n^2 C(T, \gamma, \eta_{\max}, c_0)(\delta^2 + \|\boldsymbol{b}^*\|^2).$$

## D.7 Proof of Lemma B.6

We first state three useful lemmas.

**Lemma D.6** (Adopted from Lemma E.10 of [26])**.** *Let $\boldsymbol{U}, \boldsymbol{V} : \mathbb{R}^{n \times p} \to \mathbb{R}^{n \times T}$ be two locally Lipschitz functions of $\boldsymbol{Z}$ with i.i.d. $\mathcal{N}(0, 1)$ entries, then*

$$\mathbb{E}\Big[\Big\|\boldsymbol{U}^\top \boldsymbol{Z} \boldsymbol{V} - \sum_{j=1}^p \sum_{i=1}^n \frac{\partial}{\partial z_{ij}}\Big(\boldsymbol{U}^\top \boldsymbol{e}_i \boldsymbol{e}_j^\top \boldsymbol{V}\Big)\Big\|_{\mathrm{F}}^2\Big]$$

$$\leq \mathbb{E}\|\boldsymbol{U}\|_{\mathrm{F}}^2\|\boldsymbol{V}\|_{\mathrm{F}}^2 + \mathbb{E}\sum_{ij}\Big[2\|\boldsymbol{V}\|_{\mathrm{F}}^2\Big\|\frac{\partial \boldsymbol{U}}{\partial z_{ij}}\Big\|_{\mathrm{F}}^2 + 2\|\boldsymbol{U}\|_{\mathrm{F}}^2\Big\|\frac{\partial \boldsymbol{V}}{\partial z_{ij}}\Big\|_{\mathrm{F}}^2\Big].$$

**Lemma D.7** (Adopted from Lemma F.5 of [5])**.** *Let $\boldsymbol{U}, \boldsymbol{V} : \mathbb{R}^{n \times p} \to \mathbb{R}^{n \times T}$ be two locally Lipschitz functions of $\boldsymbol{Z}$ with i.i.d. $\mathsf{N}(0, 1)$ entries. Provided the following expectations are finite, we have*

$$\mathbb{E}\Big[\Big\|p\boldsymbol{U}^\top \boldsymbol{V} - \sum_{j=1}^p \Big(\sum_{i=1}^n \partial_{ij}\boldsymbol{U}^\top \boldsymbol{e}_i - \boldsymbol{U}^\top \boldsymbol{Z} \boldsymbol{e}_j\Big)\Big(\sum_{i=1}^n \partial_{ij}\boldsymbol{e}_i^\top \boldsymbol{V} - \boldsymbol{e}_j^\top \boldsymbol{Z}^\top \boldsymbol{V}\Big)\Big\|_{\mathrm{F}}\Big]$$

$$\leq (1 + 2\sqrt{p})\big(\mathbb{E}[\|\boldsymbol{U}\|_{\mathrm{F}}^4]^{1/2} + \mathbb{E}[\|\boldsymbol{V}\|_{\mathrm{F}}^4]^{1/2} + \mathbb{E}[\|\boldsymbol{U}\|_{\partial}^4]^{1/2} + \mathbb{E}[\|\boldsymbol{V}\|_{\partial}^4]^{1/2}\big),$$

*where $\partial_{ij}\boldsymbol{U} = \partial \boldsymbol{U}/\partial z_{ij}$ and $\|\boldsymbol{U}\|_{\partial} = (\sum_{i=1}^n \sum_{j=1}^p \|\partial_{ij}\boldsymbol{U}\|_{\mathrm{F}}^2)^{1/2}$.*

We will use the above two lemmas, conditionally on $\boldsymbol{\varepsilon}$, to bound the conditional moments of $\boldsymbol{\Theta}_1, \boldsymbol{\Theta}_2, \boldsymbol{\Theta}_3, \boldsymbol{\Theta}_4$ and $\boldsymbol{\Theta}_5$ given $\boldsymbol{\varepsilon}$.

**Bound of $\boldsymbol{\Theta}_1$.** By the definition of $\boldsymbol{\Theta}_1$, we have

$$\boldsymbol{F}^\top \boldsymbol{X} \boldsymbol{H} - \sum_{i,j} \frac{\partial \boldsymbol{F}^\top \boldsymbol{e}_i \boldsymbol{e}_j^\top \boldsymbol{H}}{\partial x_{ij}} = \boldsymbol{F}^\top \boldsymbol{X} \boldsymbol{H} + \widetilde{\boldsymbol{K}} \boldsymbol{H}^\top \boldsymbol{H} - \boldsymbol{F}^\top \boldsymbol{F} \boldsymbol{W}^\top + \boldsymbol{\Upsilon}_3 \qquad \text{by (20)}$$

$$= \boldsymbol{\Theta}_1 + \boldsymbol{\Upsilon}_3.$$

Applying Lemma D.6 conditionally on $\boldsymbol{\varepsilon}$ to $(\boldsymbol{Z}, \boldsymbol{U}, \boldsymbol{V}) = (\boldsymbol{X}, \boldsymbol{F}, \boldsymbol{H})$ gives

$$\mathbb{E}[\|\boldsymbol{\Theta}_1\|_{\mathrm{F}}^2 \mid \boldsymbol{\varepsilon}] \lesssim \mathbb{E}\Big[\Big\|\boldsymbol{F}^\top \boldsymbol{X} \boldsymbol{H} - \sum_{i,j} \frac{\partial \boldsymbol{F}^\top \boldsymbol{e}_i \boldsymbol{e}_j^\top \boldsymbol{H}}{\partial x_{ij}}\Big\|^2 \mid \boldsymbol{\varepsilon}\Big] + \mathbb{E}[\|\boldsymbol{\Upsilon}_3\|_{\mathrm{F}}^2 \mid \boldsymbol{\varepsilon}]$$

$$\lesssim \mathbb{E}[\|\boldsymbol{F}\|_{\mathrm{F}}^2\|\boldsymbol{H}\|_{\mathrm{F}}^2 \mid \boldsymbol{\varepsilon}] + \mathbb{E}\sum_{ij}\Big[\|\boldsymbol{H}\|_{\mathrm{F}}^2\|\frac{\partial \boldsymbol{F}}{\partial x_{ij}}\|_{\mathrm{F}}^2 + \|\boldsymbol{F}\|_{\mathrm{F}}^2\|\frac{\partial \boldsymbol{H}}{\partial x_{ij}}\|_{\mathrm{F}}^2 \mid \boldsymbol{\varepsilon}\Big] + \mathbb{E}[\|\boldsymbol{\Upsilon}_3\|_{\mathrm{F}}^2 \mid \boldsymbol{\varepsilon}]$$

$$\leq nC(T, \gamma, \eta_{\max}, c_0)(\delta^2 + \|\boldsymbol{b}^*\|^2),$$

where the last line uses the moment bounds of $\|\boldsymbol{F}\|_{\mathrm{F}}, \|\boldsymbol{H}\|_{\mathrm{F}}$ in Lemma D.1, the moment bounds of $\|\frac{\partial \boldsymbol{H}}{\partial x_{ij}}\|_{\mathrm{F}}, \|\frac{\partial \boldsymbol{F}}{\partial x_{ij}}\|_{\mathrm{F}}$ in Lemma D.5, the moment bound of $\|\boldsymbol{\Upsilon}_3\|_{\mathrm{op}}$ in Lemma B.5.

**Bound of $\boldsymbol{\Theta}_2$.** By the definition of $\boldsymbol{\Theta}_2$, we have by (21)

$$\boldsymbol{F}^\top \boldsymbol{X} \boldsymbol{X}^\top \widetilde{\boldsymbol{F}} - \sum_{i,j} \frac{\partial \boldsymbol{F}^\top \boldsymbol{e}_i \boldsymbol{e}_j^\top \boldsymbol{X}^\top \widetilde{\boldsymbol{F}}}{\partial x_{ij}} = \boldsymbol{F}^\top \boldsymbol{X} \boldsymbol{X}^\top \widetilde{\boldsymbol{F}} + \widetilde{\boldsymbol{K}} \boldsymbol{H}^\top \boldsymbol{X}^\top \widetilde{\boldsymbol{F}} - p\boldsymbol{F}^\top \widetilde{\boldsymbol{F}} + \boldsymbol{F}^\top \boldsymbol{F} \widehat{\boldsymbol{A}}^\top + \boldsymbol{\Upsilon}_5$$

$$= n\boldsymbol{\Theta}_2 + \boldsymbol{\Upsilon}_5.$$

Applying Lemma D.6 conditionally on $\boldsymbol{\varepsilon}$ to $(\boldsymbol{Z}, \boldsymbol{U}, \boldsymbol{V}) = (\boldsymbol{X}, \boldsymbol{F}, \boldsymbol{X}^\top \widetilde{\boldsymbol{F}})$ gives

$$n^2\mathbb{E}[\|\boldsymbol{\Theta}_2\|_{\mathrm{F}}^2 \mid \boldsymbol{\varepsilon}] \lesssim \mathbb{E}\Big[\Big\|\boldsymbol{F}^\top \boldsymbol{X} \boldsymbol{X}^\top \widetilde{\boldsymbol{F}} - \sum_{i,j} \frac{\partial \boldsymbol{F}^\top \boldsymbol{e}_i \boldsymbol{e}_j^\top \boldsymbol{X}^\top \widetilde{\boldsymbol{F}}}{\partial x_{ij}}\Big\|^2 \mid \boldsymbol{\varepsilon}\Big] + \mathbb{E}[\|\boldsymbol{\Upsilon}_5\|_{\mathrm{F}}^2 \mid \boldsymbol{\varepsilon}]$$

$$\lesssim \mathbb{E}[\|\boldsymbol{F}\|_{\mathrm{F}}^2\|\boldsymbol{X}^\top \widetilde{\boldsymbol{F}}\|_{\mathrm{F}}^2 \mid \boldsymbol{\varepsilon}] + \mathbb{E}\sum_{ij}\Big[\|\boldsymbol{X}^\top \widetilde{\boldsymbol{F}}\|_{\mathrm{F}}^2\|\frac{\partial \boldsymbol{F}}{\partial x_{ij}}\|_{\mathrm{F}}^2 + \|\boldsymbol{F}\|_{\mathrm{F}}^2\|\frac{\partial \boldsymbol{X}^\top \widetilde{\boldsymbol{F}}}{\partial x_{ij}}\|_{\mathrm{F}}^2 \mid \boldsymbol{\varepsilon}\Big] + \mathbb{E}[\|\boldsymbol{\Upsilon}_5\|_{\mathrm{F}}^2 \mid \boldsymbol{\varepsilon}]$$

$$\lesssim \mathbb{E}[\|\widetilde{\boldsymbol{F}}\|_{\mathrm{F}}^4\|\boldsymbol{X}\|_{\mathrm{op}}^2 \mid \boldsymbol{\varepsilon}] + \mathbb{E}\Big[(1 + \|\boldsymbol{X}\|_{\mathrm{op}}^2)\|\widetilde{\boldsymbol{F}}\|_{\mathrm{F}}^2 \sum_{ij}\|\frac{\partial \widetilde{\boldsymbol{F}}}{\partial x_{ij}}\|_{\mathrm{F}}^2 \mid \boldsymbol{\varepsilon}\Big] + T\mathbb{E}[\|\boldsymbol{\Upsilon}_5\|_{\mathrm{op}}^2 \mid \boldsymbol{\varepsilon}]$$

$$\leq n^3 C(T, \gamma, \eta_{\max}, c_0)(\delta^2 + \|\boldsymbol{b}^*\|^2).$$

Here, the penultimate line uses $\|\boldsymbol{F}\|_{\mathrm{F}} \leq \|\widetilde{\boldsymbol{F}}\|_{\mathrm{F}}$, $\|\frac{\partial \boldsymbol{F}}{\partial x_{ij}}\|_{\mathrm{F}} \leq \|\frac{\partial \widetilde{\boldsymbol{F}}}{\partial x_{ij}}\|_{\mathrm{F}}$, and $\|\frac{\partial \boldsymbol{X}^\top \widetilde{\boldsymbol{F}}}{\partial x_{ij}}\|_{\mathrm{F}} = \|\boldsymbol{e}_j^\top \boldsymbol{e}_i^\top \frac{\partial \widetilde{\boldsymbol{F}}}{\partial x_{ij}} + \boldsymbol{X}^\top \frac{\partial \widetilde{\boldsymbol{F}}}{\partial x_{ij}}\|_{\mathrm{F}} \leq \|\frac{\partial \widetilde{\boldsymbol{F}}}{\partial x_{ij}}\|_{\mathrm{F}} + \|\boldsymbol{X}\|_{\mathrm{op}}\|\frac{\partial \widetilde{\boldsymbol{F}}}{\partial x_{ij}}\|_{\mathrm{F}}$. The last line uses the moment bounds of $\|\boldsymbol{X}\|_{\mathrm{op}}, \|\widetilde{\boldsymbol{F}}\|_{\mathrm{F}}$ in Lemma D.1, the moment bound of $\|\frac{\partial \widetilde{\boldsymbol{F}}}{\partial x_{ij}}\|_{\mathrm{F}}$ in Lemma D.5, the moment bound of $\|\boldsymbol{\Upsilon}_5\|_{\mathrm{op}}$ in Lemma B.5.

**Bound of $\boldsymbol{\Theta}_3$.** By the definition of $\boldsymbol{\Theta}_3$, we have

$$\boldsymbol{H}^\top \boldsymbol{X}^\top \boldsymbol{X} \boldsymbol{H} - \sum_{i,j} \frac{\partial \boldsymbol{H}^\top \boldsymbol{X}^\top \boldsymbol{e}_i \boldsymbol{e}_j^\top \boldsymbol{H}}{\partial x_{ij}}$$
$$= \boldsymbol{H}^\top \boldsymbol{X}^\top \boldsymbol{X} \boldsymbol{H} - (n \boldsymbol{I}_T - \widetilde{\mathbf{A}}) \boldsymbol{H}^\top \boldsymbol{H} - \boldsymbol{H}^\top \boldsymbol{X}^\top \boldsymbol{F} \boldsymbol{W}^\top + \boldsymbol{\Upsilon}_4 \qquad \text{by (21)}$$
$$= \boldsymbol{\Theta}_3 + \boldsymbol{\Upsilon}_4.$$

Applying Lemma D.6 conditionally on $\varepsilon$ to $(\boldsymbol{Z}, \boldsymbol{U}, \boldsymbol{V}) = (\boldsymbol{X}, \boldsymbol{X} \boldsymbol{H}, \boldsymbol{H})$ gives

$$\mathbb{E}[\|\boldsymbol{\Theta}_3\|_{\mathrm{F}}^2 \mid \varepsilon] \lesssim \mathbb{E}\Big[\Big\| \boldsymbol{H}^\top \boldsymbol{X}^\top \boldsymbol{X} \boldsymbol{H} - \sum_{i,j} \frac{\partial \boldsymbol{H}^\top \boldsymbol{X}^\top \boldsymbol{e}_i \boldsymbol{e}_j^\top \boldsymbol{H}}{\partial x_{ij}} \Big\|^2 \mid \varepsilon \Big] + \mathbb{E}[\|\boldsymbol{\Upsilon}_4\|_{\mathrm{F}}^2 \mid \varepsilon]$$

$$\lesssim \mathbb{E}[\|\boldsymbol{X} \boldsymbol{H}\|_{\mathrm{F}}^2 \|\boldsymbol{H}\|_{\mathrm{F}}^2 \mid \varepsilon] + \mathbb{E} \sum_{ij} \Big[ \|\boldsymbol{X} \boldsymbol{H}\|_{\mathrm{F}}^2 \|\frac{\partial \boldsymbol{H}}{\partial x_{ij}}\|_{\mathrm{F}}^2 + \|\boldsymbol{H}\|_{\mathrm{F}}^2 \|\frac{\partial \boldsymbol{X} \boldsymbol{H}}{\partial x_{ij}}\|_{\mathrm{F}}^2 \mid \varepsilon \Big] + \mathbb{E}[\|\boldsymbol{\Upsilon}_4\|_{\mathrm{F}}^2 \mid \varepsilon]$$

$$\lesssim \mathbb{E}[\|\boldsymbol{X}\|_{\mathrm{op}}^2 \|\boldsymbol{H}\|_{\mathrm{F}}^4 \mid \varepsilon] + \mathbb{E}\Big[ (1 + \|\boldsymbol{X}\|_{\mathrm{op}}^2) \|\boldsymbol{H}\|_{\mathrm{F}}^2 \sum_{ij} \|\frac{\partial \boldsymbol{H}}{\partial x_{ij}}\|_{\mathrm{F}}^2 \mid \varepsilon \Big] + T \mathbb{E}[\|\boldsymbol{\Upsilon}_4\|_{\mathrm{op}}^2 \mid \varepsilon]$$

$$\leq n C(T, \gamma, \eta_{\max}, c_0)(\delta^2 + \|\boldsymbol{b}^*\|^2).$$

Here, the penultimate line uses $\|\frac{\partial \boldsymbol{X} \boldsymbol{H}}{\partial x_{ij}}\|_{\mathrm{F}} = \|\boldsymbol{e}_i \boldsymbol{e}_j^\top \frac{\partial \boldsymbol{H}}{\partial x_{ij}} + \boldsymbol{X} \frac{\partial \boldsymbol{H}}{\partial x_{ij}}\|_{\mathrm{F}} \leq (1 + \|\boldsymbol{X}\|_{\mathrm{op}}) \|\frac{\partial \boldsymbol{H}}{\partial x_{ij}}\|_{\mathrm{F}}$, and the last line uses the moment bounds of $\|\boldsymbol{X}\|_{\mathrm{op}}, \|\boldsymbol{H}\|_{\mathrm{F}}$ in Lemma D.1, the moment bounds of $\|\frac{\partial \boldsymbol{H}}{\partial x_{ij}}\|_{\mathrm{F}}$ in Lemma D.5, the moment bound of $\|\boldsymbol{\Upsilon}_4\|_{\mathrm{op}}$ in Lemma B.5.

**Bound of $\boldsymbol{\Theta}_4$.** By definition, we have

$$\sum_{i=1}^n \frac{\partial \boldsymbol{F}^\top \boldsymbol{e}_i}{\partial x_{ij}} = -(\widetilde{\boldsymbol{K}} \boldsymbol{H}^\top + \boldsymbol{\Upsilon}_1^\top) \boldsymbol{e}_j \text{ and } \sum_{i=1}^n \frac{\partial \widetilde{\boldsymbol{F}}^\top \boldsymbol{e}_i}{\partial x_{ij}} = -(\widehat{\boldsymbol{K}} \boldsymbol{H}^\top + \widetilde{\boldsymbol{\Upsilon}}_1^\top) \boldsymbol{e}_j.$$

Using $\sum_{j=1}^p \boldsymbol{e}_j \boldsymbol{e}_j^\top = \boldsymbol{I}_p$, we find

$$\sum_{j=1}^p \Big(\sum_{i=1}^n \frac{\partial \boldsymbol{F}^\top \boldsymbol{e}_i}{\partial x_{ij}} - \boldsymbol{F}^\top \boldsymbol{X} \boldsymbol{e}_j\Big)\Big(\sum_{i=1}^n \frac{\partial \widetilde{\boldsymbol{F}}^\top \boldsymbol{e}_i}{\partial x_{ij}} - \widetilde{\boldsymbol{F}}^\top \boldsymbol{X} \boldsymbol{e}_j\Big)^\top$$
$$= (\widetilde{\boldsymbol{K}} \boldsymbol{H}^\top + \boldsymbol{F}^\top \boldsymbol{X} + \boldsymbol{\Upsilon}_1^\top)(\widehat{\boldsymbol{K}} \boldsymbol{H}^\top + \widetilde{\boldsymbol{F}}^\top \boldsymbol{X} + \widetilde{\boldsymbol{\Upsilon}}_1^\top).$$

This further implies

$$p \boldsymbol{F}^\top \widetilde{\boldsymbol{F}} - \sum_{j=1}^p \Big(\sum_{i=1}^n \frac{\partial \boldsymbol{F}^\top \boldsymbol{e}_i}{\partial x_{ij}} - \boldsymbol{F}^\top \boldsymbol{X} \boldsymbol{e}_j\Big)\Big(\sum_{i=1}^n \frac{\partial \widetilde{\boldsymbol{F}}^\top \boldsymbol{e}_i}{\partial x_{ij}} - \widetilde{\boldsymbol{F}}^\top \boldsymbol{X} \boldsymbol{e}_j\Big)^\top$$
$$= n \boldsymbol{\Theta}_4 + \underbrace{\boldsymbol{\Upsilon}_1^\top(\widehat{\boldsymbol{K}} \boldsymbol{H}^\top + \widetilde{\boldsymbol{F}}^\top \boldsymbol{X}) + (\widetilde{\boldsymbol{K}} \boldsymbol{H}^\top + \boldsymbol{F}^\top \boldsymbol{X})\widetilde{\boldsymbol{\Upsilon}}_1^\top + \boldsymbol{\Upsilon}_1^\top \widetilde{\boldsymbol{\Upsilon}}_1^\top}_{\widetilde{\boldsymbol{\Upsilon}}_4}.$$

Applying Lemma D.7 conditionally on $\varepsilon$ to $(\boldsymbol{Z}, \boldsymbol{U}, \boldsymbol{V}) = (\boldsymbol{X}, \boldsymbol{F}, \widetilde{\boldsymbol{F}})$,

$$n \mathbb{E}[\|\boldsymbol{\Theta}_4\|_{\mathrm{F}} \mid \varepsilon] \lesssim \mathbb{E}\Big[\Big\| p \boldsymbol{F}^\top \widetilde{\boldsymbol{F}} - \sum_{j=1}^p \Big(\sum_{i=1}^n \frac{\partial \boldsymbol{F}^\top \boldsymbol{e}_i}{\partial x_{ij}} - \boldsymbol{F}^\top \boldsymbol{X} \boldsymbol{e}_j\Big)\Big(\sum_{i=1}^n \frac{\partial \widetilde{\boldsymbol{F}}^\top \boldsymbol{e}_i}{\partial x_{ij}} - \widetilde{\boldsymbol{F}}^\top \boldsymbol{X} \boldsymbol{e}_j\Big)^\top \Big\|_{\mathrm{F}} \mid \varepsilon \Big] + \mathbb{E}[\|\widetilde{\boldsymbol{\Upsilon}}_4\|_{\mathrm{F}} \mid \varepsilon]$$

$$\lesssim (1 + 2\sqrt{p})\big(\mathbb{E}[\|\boldsymbol{F}\|_{\mathrm{F}}^4 \mid \varepsilon]^{1/2} + \mathbb{E}[\|\widetilde{\boldsymbol{F}}\|_{\mathrm{F}}^4 \mid \varepsilon]^{1/2} + \mathbb{E}[\|\boldsymbol{F}\|_{\partial}^4 \mid \varepsilon]^{1/2} + \mathbb{E}[\|\widetilde{\boldsymbol{F}}\|_{\partial}^4 \mid \varepsilon]^{1/2}\big) + T \mathbb{E}[\|\widetilde{\boldsymbol{\Upsilon}}_4\|_{\mathrm{op}}^2 \mid \varepsilon]$$
$$\leq n^{3/2} C(T, \gamma, \eta_{\max}, c_0)(\delta^2 + \|\boldsymbol{b}^*\|^2).$$

Here, the last inequality uses the moment bounds of $\|\boldsymbol{F}\|_{\mathrm{F}}, \|\widetilde{\boldsymbol{F}}\|_{\mathrm{F}}$ in Lemma D.1, and the moment bounds of $\|\boldsymbol{F}\|_{\partial}^2, \|\widetilde{\boldsymbol{F}}\|_{\partial}^2$ in Lemma D.5, and the moment bound of $\|\widetilde{\boldsymbol{\Upsilon}}_4\|_{\mathrm{op}}$ in Lemma B.5.

**Bound of $\boldsymbol{\Theta}_5$.** By the identity (19), we have

$$\sum_{i=1}^{n}\Big(\sum_{j=1}^{p}\frac{\partial \boldsymbol{H}^\top \boldsymbol{e}_j}{\partial x_{ij}} - \boldsymbol{H}^\top \boldsymbol{X}^\top \boldsymbol{e}_i\Big)\Big(\sum_{j=1}^{p}\frac{\partial \boldsymbol{H}^\top \boldsymbol{e}_j}{\partial x_{ij}} - \boldsymbol{H}^\top \boldsymbol{X}^\top \boldsymbol{e}_i\Big)^\top$$

$$= (\boldsymbol{W}\boldsymbol{F}^\top - \boldsymbol{H}^\top \boldsymbol{X}^\top - \boldsymbol{\Upsilon}_2^\top)(\boldsymbol{W}\boldsymbol{F}^\top - \boldsymbol{H}^\top \boldsymbol{X}^\top - \boldsymbol{\Upsilon}_2^\top)^\top.$$

By the definition of $\boldsymbol{\Theta}_5$, we have

$$n\boldsymbol{H}^\top \boldsymbol{H} - \sum_{i=1}^{n}\Big(\sum_{j=1}^{p}\frac{\partial \boldsymbol{H}^\top \boldsymbol{e}_j}{\partial x_{ij}} - \boldsymbol{H}^\top \boldsymbol{X}^\top \boldsymbol{e}_i\Big)\Big(\sum_{j=1}^{p}\frac{\partial \boldsymbol{H}^\top \boldsymbol{e}_j}{\partial x_{ij}} - \boldsymbol{H}^\top \boldsymbol{X}^\top \boldsymbol{e}_i\Big)^\top$$

$$= \boldsymbol{\Theta}_5 + \underbrace{\boldsymbol{\Upsilon}_2^\top(\boldsymbol{W}\boldsymbol{F}^\top - \boldsymbol{H}^\top \boldsymbol{X}^\top)^\top + (\boldsymbol{W}\boldsymbol{F}^\top - \boldsymbol{H}^\top \boldsymbol{X}^\top)\boldsymbol{\Upsilon}_2 - \boldsymbol{\Upsilon}_2^\top \boldsymbol{\Upsilon}_2}_{\widetilde{\boldsymbol{\Upsilon}}_5}.$$

Here $\boldsymbol{\Upsilon}_2 = \sum_j((\boldsymbol{e}_j^\top \boldsymbol{H}) \otimes \boldsymbol{I}_n)\mathcal{DS}(\boldsymbol{I}_T \otimes \boldsymbol{X})\boldsymbol{\Gamma}^\top(\boldsymbol{I}_T \otimes \boldsymbol{e}_j)$.

Applying Lemma D.7 conditionally on $\boldsymbol{\varepsilon}$ to $(\boldsymbol{Z}, \boldsymbol{U}, \boldsymbol{V}) = (\boldsymbol{X}^\top, \boldsymbol{H}, \boldsymbol{H})$ (i.e., consider the mapping from $\mathbb{R}^{p\times n}$ to $\mathbb{R}^{p\times T}$: $\boldsymbol{X}^\top \mapsto \boldsymbol{H}$) gives

$$\mathbb{E}[\|\boldsymbol{\Theta}_5\|_{\mathrm{F}} \mid \boldsymbol{\varepsilon}]$$

$$\leq \mathbb{E}\Big[\Big\|n\boldsymbol{H}^\top \boldsymbol{H} - \sum_{i=1}^{n}\Big(\sum_{j=1}^{p}\frac{\partial \boldsymbol{H}^\top \boldsymbol{e}_j}{\partial x_{ij}} - \boldsymbol{H}^\top \boldsymbol{X}^\top \boldsymbol{e}_i\Big)\Big(\sum_{j=1}^{p}\frac{\partial \boldsymbol{H}^\top \boldsymbol{e}_j}{\partial x_{ij}} - \boldsymbol{H}^\top \boldsymbol{X}^\top \boldsymbol{e}_i\Big)^\top\Big\|_{\mathrm{F}} \mid \boldsymbol{\varepsilon}\Big] + \mathbb{E}[\|\widetilde{\boldsymbol{\Upsilon}}_5\|_{\mathrm{F}} \mid \boldsymbol{\varepsilon}]$$

$$\lesssim (1 + 2\sqrt{p})\big(\mathbb{E}[\|\boldsymbol{H}\|_{\mathrm{F}}^4 \mid \boldsymbol{\varepsilon}]^{1/2} + \mathbb{E}[\|\boldsymbol{H}\|_{\partial}^4 \mid \boldsymbol{\varepsilon}]^{1/2}\big) + T\mathbb{E}[\|\widetilde{\boldsymbol{\Upsilon}}_5\|_{\mathrm{op}}^2 \mid \boldsymbol{\varepsilon}]$$

$$\leq n^{1/2}C(T, \gamma, \eta_{\max}, c_0)(\delta^2 + \|\boldsymbol{b}^*\|^2).$$

Here, the last line use the moment bounds of $\|\boldsymbol{H}\|_{\mathrm{F}}$ in Lemma D.1, the moment bound of $\|\boldsymbol{H}\|_{\partial}^2$ in Lemma D.5, and the moment bound of $\|\widetilde{\boldsymbol{\Upsilon}}_5\|_{\mathrm{op}}$ in Lemma B.5.

**Bound of $\boldsymbol{\Theta}_6$.** Using (19), we have

$$\sum_{i=1}^{n}\sum_{j=1}^{p}\frac{\partial \boldsymbol{E}^\top \boldsymbol{e}_i \boldsymbol{e}_j^\top \boldsymbol{H}}{\partial x_{ij}} = \boldsymbol{E}^\top(\boldsymbol{F}\boldsymbol{W}^\top - \boldsymbol{\Upsilon}_2).$$

It follows that

$$\boldsymbol{E}^\top \boldsymbol{X}\boldsymbol{H} - \sum_{i=1}^{n}\sum_{j=1}^{p}\frac{\partial \boldsymbol{E}^\top \boldsymbol{e}_i \boldsymbol{e}_j^\top \boldsymbol{H}}{\partial x_{ij}} = \boldsymbol{E}^\top \boldsymbol{X}\boldsymbol{H} - \boldsymbol{E}^\top(\boldsymbol{F}\boldsymbol{W}^\top - \boldsymbol{\Upsilon}_2)$$

$$= \|\boldsymbol{E}\|_{\mathrm{F}}\boldsymbol{\Theta}_6 + \boldsymbol{E}^\top \boldsymbol{\Upsilon}_2.$$

Thus, using $\widetilde{\boldsymbol{E}} = \boldsymbol{E}/\|\boldsymbol{E}\|_{\mathrm{F}}$, we have

$$\boldsymbol{\Theta}_6 = \widetilde{\boldsymbol{E}}^\top \boldsymbol{X}\boldsymbol{H} - \sum_{i=1}^{n}\sum_{j=1}^{p}\frac{\partial \widetilde{\boldsymbol{E}}^\top \boldsymbol{e}_i \boldsymbol{e}_j^\top \boldsymbol{H}}{\partial x_{ij}} - \widetilde{\boldsymbol{E}}^\top \boldsymbol{\Upsilon}_2.$$

Applying Lemma D.6 conditionally on $\boldsymbol{\varepsilon}$ to $(\boldsymbol{Z}, \boldsymbol{U}, \boldsymbol{V}) = (\boldsymbol{X}, \widetilde{\boldsymbol{E}}, \boldsymbol{H})$ gives

$$\mathbb{E}[\|\boldsymbol{\Theta}_6\|_{\mathrm{F}}^2 \mid \boldsymbol{\varepsilon}] \lesssim \mathbb{E}\Big[\Big\|\widetilde{\boldsymbol{E}}^\top \boldsymbol{X}\boldsymbol{H} - \sum_{i=1}^{n}\sum_{j=1}^{p}\frac{\partial \widetilde{\boldsymbol{E}}^\top \boldsymbol{e}_i \boldsymbol{e}_j^\top \boldsymbol{H}}{\partial x_{ij}}\Big\|^2 \mid \boldsymbol{\varepsilon}\Big] + \mathbb{E}[\|\widetilde{\boldsymbol{E}}^\top \boldsymbol{\Upsilon}_2\|_{\mathrm{F}}^2 \mid \boldsymbol{\varepsilon}]$$

$$\lesssim \mathbb{E}[\|\widetilde{\boldsymbol{E}}\|_{\mathrm{F}}^2\|\boldsymbol{H}\|_{\mathrm{F}}^2 \mid \boldsymbol{\varepsilon}] + \mathbb{E}\Big[\|\widetilde{\boldsymbol{E}}\|_{\mathrm{F}}^2\|\frac{\partial \boldsymbol{H}}{\partial x_{ij}}\|_{\mathrm{F}}^2 \mid \boldsymbol{\varepsilon}\Big] + \mathbb{E}[\|\widetilde{\boldsymbol{E}}^\top \boldsymbol{\Upsilon}_2\|_{\mathrm{F}}^2 \mid \boldsymbol{\varepsilon}]$$

$$\leq \mathbb{E}[\|\boldsymbol{H}\|_{\mathrm{F}}^2 \mid \boldsymbol{\varepsilon}] + \mathbb{E}\Big[\|\frac{\partial \boldsymbol{H}}{\partial x_{ij}}\|_{\mathrm{F}}^2 \mid \boldsymbol{\varepsilon}\Big] + \mathbb{E}[\|\boldsymbol{\Upsilon}_2\|_{\mathrm{op}}^2 \mid \boldsymbol{\varepsilon}]$$

$$\leq C(T, \gamma, \eta_{\max}, c_0)(\delta^2 + \|\boldsymbol{b}^*\|^2).$$

Here, the last line uses the moment bound of $\|\boldsymbol{H}\|_{\mathrm{F}}$ in Lemma D.1, the moment bounds of $\|\frac{\partial \boldsymbol{H}}{\partial x_{ij}}\|_{\mathrm{F}}$ in Lemma D.5, and the moment bound of $\|\boldsymbol{\Upsilon}_2\|_{\mathrm{op}}$ in Lemma B.5. This finishes the proof of Lemma B.6.

