# OpenReview forum: "Estimating Generalization Performance Along the Trajectory of Proximal SGD in Robust Regression"
_NeurIPS.cc/2024/Conference — NeurIPS 2024 poster_

### Official Review · Reviewer_boVy · 2024-07-06

**Soundness:** 3
**Presentation:** 3
**Contribution:** 3
**Rating:** 7
**Confidence:** 4

**Summary:**

The paper explores linear regression with Gaussian design corrupted by noise that has bounded first moments, focusing on the 'high-dimensional' regime where the feature dimensionality $p$ scales proportionally with the number of training data $n$. The authors propose two estimators that, aside from an additive noise term, approximate the out-of-sample error (measured with the square loss) achieved by a general iterative scheme. This scheme includes commonly used solvers like GD, SGD, proximal GD, and proximal SGD, which are used to compute the regularized empirical risk minimizer with a convex, differentiable, and smooth loss function (with Lipschitz-continuous gradients) and a potentially non-smooth regularizer. This framework encompasses robust linear regression with heavy-tailed noise using the Huber loss function. The authors provide non-asymptotic guarantees in probability for finite $n$ (and $p$), demonstrating the asymptotic consistency of their estimators (Theorems 3.6 and 3.7). They also present examples and numerical experiments. Their primary tool is a generalized probabilistic approximation of the generalization error previously discussed in specific settings in the references [3] and [5].

**Strengths:**

The authors have developed a procedure that reliably determines the best stopping time for solvers in contexts where no consistent estimators were previously available, particularly in robust regression with heavy-tailed noise, Huber-like loss functions, and non-smooth penalty terms. This methodology offers an alternative to cross-validation techniques, which are generally inconsistent (e.g., V-fold) or computationally expensive (e.g., leave-one-out). The paper is well-written and includes clear proofs.

**Weaknesses:**

The main results of the paper can be seen as an extension of the findings in [5], applied to a broader context that encompasses the stochastic case and extends beyond the square loss. While the generality of the weaker conditions is appreciated and introduces a different structure, the work is somewhat marginal in its contribution. Although the new methodology can track the generalization performance over training time for various iterative solver schemes of interest, it remains uncertain how this methodology can be used to (provably) "optimally" tune parameters other than the stopping time. This limitation seems notable, as the increased flexibility of the algorithmic schemes—allowing for additional tuning parameters such as mini-batch size and loss-specific parameters—is not accompanied by an enhanced analysis of these tuning parameters.

**Questions:**

1) In the case of the square loss and full-batch gradient descent, do the authors recover exactly the same theoretical results in [5]? If not, what are the differences in this case?

2) The authors claim that their procedure can be used to choose the "tuning parameters that achieves the smallest out-of-sample error" (Page 2). While this is clear for stopping time, can the authors give examples of how their methodology can be use to tune other tuning parameters?

3) While the authors clearly highlight the relevance of their contributions to robust statistics and their goal of tracking solver performance over iteration time, a specific comparison of their methodology and consistency results to other classical methods in some fundamental models, such as ridge regression and lasso regression, would be beneficial. Although Section 1.1 provides a discussion on this topic, it lacks quantitative details, making it unclear how the proposed methods compare in terms of non-asymptotic statistical rates (related to consistency results) and computational efficiency. For example, the computational complexity of the data-dependent estimator the authors consider ($\tilde{r}_t$ on page 6) is not clearly compared to other estimators in basic settings. Can the authors address this?

4) The paper does not seem to make any assumptions on the penalty term, i.e. the function g in (2). This seems surprising. Can the authors elaborate on this?

5) Could the authors comment on the challenges of extending their analysis beyond the proportional regime they consider, where the data dimension is of the same order as the sample size?

****
I have increased my score to 'Accept' post rebuttal.

**Limitations:**

The paper discusses limitation of the current approach as a way to motivate follow up research in the conclusions.

---

> ### Author Rebuttal · Authors · 2024-08-07
>
> Thank you for the comments, we provide our response below.
>
> > **Q1:**
> In the case of the square loss and full-batch gradient descent, do the authors recover exactly the same theoretical results in [5]? If not, what are the differences in this case?
>
> **A1:**
> If the squared loss and full-batch gradient descent is used, our proposed $\tilde{r}_t$ has the same formula as $\hat{r}_t$ in [5]. However, our work allows the noise vector can be heavy-tailed (e.g., infinite variance), which is significantly different from the Gaussian noise condition in [5].
> To handle heavy-tailed noise, we consider a data-fitting loss whose derivative is Lipschitz continuous and bounded (Assumption 3.3). Hence, the squared loss is excluded by Assumption 3.3 of our current paper, so the result of [5] is not formally implied by the theorems of the submission.
>
> > **Q2:**
> ...can the authors give examples of how their methodology can be use to tune other tuning parameters?
>
> **A2:**
> Let's take the L1 regularization parameter $\lambda$ as an example. In proximal SGD, $\lambda$ appears in the soft-thresholding parameter of Example 2.4. The algorithm's iterates, $\hat{b}^t$, depend on the choice of $\lambda$, so we can write it as $\hat{b}^t(\lambda)$. For each candidate $\lambda \in \{\lambda_1, \ldots, \lambda_K\}$ (a finite grid of tuning parameters), we compute the estimated risk $\tilde{r}_t(\lambda)$. This estimate serves as a criterion to choose the $(t^*, k^*)$ that minimizes $\tilde{r}_t(\lambda)$. As long as the grid of $\lambda$ is finite, the risk estimate for $\hat{b}^t(\lambda)$ is consistent across all parameters in the grid. The selected $(t^*, k^*)$ leads to the smallest generalization error over the grid, up to a vanishing error term.
>
> > **Q3:**
> ... a specific comparison of their methodology and consistency results to other classical methods ... would be beneficial.
>
> **A3:**
> We are not 100% sure to understand the question, but try to provide some useful pointers below. If the question is not answered by the pointers below, we would be happy to provide more.
>
> 1. Comparison of Methodology.
> Methods for estimating the generalization performance of ridge and lasso regression focus on the minimizer (denoted as $\hat{b}$ in eq. (2)) of the corresponding optimization problem. Such estimates, say for the Lasso, are studied in [1, 3, 9, 22] (citations from the submission PDF). These risk estimates are only valid for the final minimizer $\hat b$ in eq. (2).
> In particular, such estimators are not applicable to estimate the risk of intermediate iterates of algorithms.
>
> For Lasso and Ridge, the risk estimates take the form of $\frac{||y - X \hat{b}||^2/n}{(1-df/n)^2}$, which can be viewed as an adjusted training error, where the degrees of freedom $df$ is, for the Lasso, the number of nonzero coefficients of $\hat b$.
> In contrast, our proposed risk estimator provides a consistent estimate of the risk of $\hat{b}^t$ at each $t$.
> The proposed risk estimate takes a very different form (weighted average of residual and previous gradients in eq. (11)) than the simpler ones available for Lasso/Ridge that can be simply described as an adjusted training error.
>
> 2. Computational comparison.
> As we mentioned in our response to Question 1 from Reviewer 3EXM, the computational complexity of calculating our estimate $\tilde{r}_t$ for all $t\in[T]$ is $O(npT^6)$. The computational complexity of the Lasso estimate is $O(np^2 + p^3)$ (see, for example, [1]). Thus, for a fixed $T$, our risk estimate has a lower complexity than that of Lasso.
>
> > (...) in terms of non-asymptotic statistical rates (related to consistency results)
>
> The rate of convergence in Theorem 3.6 depends on the tails of the entries of the noise vector.
> If the variance of the noise is finite then the right-hand side of (12) in Theorem 3.6 reduces to $C/(\varepsilon\sqrt n)$ (where $C$ has the same dependence as the constant in the numerator of (12)). Thus, if the variance is finite, we recover an error term of order $O_P(1/\sqrt n)$, same as the best known rate for the difference between the risk and its estimate ([3] from the submission PDF).
>
> Now if the noise has infinite variance and is heavy-tailed, the rate of convergence of $\min\{1,||\varepsilon||/n\}$ in the right-hand side in (12) will depend on which moment of the noise exists (with a finite moment of order one, (12) converges to 0. If the moment of order $1+\delta$ is finite, explicit rates can be obtained).
>
> > **Q4:**
> The paper does not seem to make any assumptions on the penalty term, i.e. the function g in (2). Can the authors elaborate on this?
>
> **A4:**
> Yes, we did not make explicit assumptions about the penalty term. Instead, we focus on the algorithm iterates that have the form of Eq. (4). The only requirement on the functions $\phi_t,\psi$ are given in Assumption 3.3. As long as the regression problem can be solved by algorithms similar to Eq. (4), our proposed risk estimate can be used to track the generalization error of these algorithms.
>
> For instance, for any convex penalty function $g$, consider the corresponding proximal SGD algorithm in Example 2.4, with the soft-thresholding
> replaced with the proximal of the penalty $g$. Since the proximal of $g$ is Lipschitz for any convex $g$, Assumption 3.3 is satisfied.
> No further assumption is needed on the penalty $g$ except convexity.
>
> > **Q5:**
> Could the authors comment on the challenges of extending their analysis beyond the proportional regime they consider, where the data dimension is of the same order as the sample size?
>
> **A5:**
> The consistency of the proposed estimate $\hat r_t$ requires the regime where the ratio $p/n \le \gamma$, because in the proof (e.g., the proof of Lemma D.5), we need to bound the finite moments of $ ||X||\_{op} /\sqrt{n}$  by a constant. This is not possible if $p\ggg n$
> as $||X||_{op}$ is known to be of order $\sqrt n + \sqrt p$  [11].
>
> ---
> [1] Efron, Bradley, et al. "Least angle regression." (2004): 407-499.

---

> > ### Comment · Reviewer_boVy · 2024-08-10
> >
> > I appreciate the authors' response and maintain a positive outlook about this work. As a result of the authors' reply, I am raising my evaluation to Accept.

---

> > > ### Author Response · Authors · 2024-08-14
> > >
> > > Thanks for your final comments and your work throughout the process.

---

### Official Review · Reviewer_yB9W · 2024-07-11

**Soundness:** 3
**Presentation:** 3
**Contribution:** 3
**Rating:** 6
**Confidence:** 4

**Summary:**

This paper focuses on deriving consistent estimators of the generalization error for robust regression with Gaussian design and heavy tailed noise along the SGD trajectory. They propose two estimators, $\hat r$ (which requires knowledge of the covariance) and $\tilde r$ (which does not) and prove consistency of both estimators when the number of samples is at least linear in the dimension. They support this with experiments which demonstrate that both $\hat r$ and $\tilde r$ accurately capture the generalization error in various settings.

**Strengths:**

The paper is overall very well written, including the overall motivation, the related work, and the technical ideas behind Theorems 3.6 and 3.7.

**Weaknesses:**

- The numerators in Theorems 3.6 and 3.7 include unspecified constants $C(T,\ldots)$. From the appendix, it appears that $C(T,\ldots)$ scales like $T^T$, so that you can only run for $T \approx \log(n)/\log\log(n)$ steps before the bound is vacuous, which is severely restricting. I'm concerned it may not be possible for SGD to significantly decrease the loss in this number of steps, rendering the generalization estimate vacuous.

- While some intuition for $\boldsymbol{W}$ (eq 8) is given in Section 3.1, none is given for $\widehat{\mathbf{A}}$ (eq 9) or $\widehat{\mathbf{K}}$ except that they are used to construct the weights for Theorem 3.7. This makes the notation in sections 3.2 and 3.3 somewhat difficult to follow.

Minor Points:
- line 29: starts -> start
- line 118: maybe provide some intuition here for $\phi_t, \psi$ rather than deferring it to sections 2.1/2.2 (e.g. just $\phi_t$ is a proximal operator)
- line 170: all -> both
- line 207: it is strange to define $\hat r$ in terms of $\mathbf{W}$ and $\tilde r$ in terms of $\widehat{\mathbf{W}}$. Perhaps use $\widetilde{\mathbf{W}}$ instead of $\widehat{\mathbf{W}}$?
- line 269: the three ... reveal

**Questions:**

- What are the dependencies on $T,\eta_{max}$ in the numerators of Theorems 2.6 and 2.7?
- Is the reason that $W$ must be computed recursively simply to invert $\mathcal{M}$ block by block? I don't see how to connect the unrolling in section 3.1 to the Kronecker expressions in 3.2.

**Limitations:**

The authors have adequately addressed the limitations of their work.

---

> ### Author Rebuttal · Authors · 2024-08-06
>
> > **Q1:**
> (...) dependencies on $T, \eta_{max}$ in numerators of Theorems 2.6,2.7?
>
> **A1:**
> The dependence on $T$ is $T^T$ and the dependence on $\eta_{\text{max}}$ is $\eta_{\text{max}}^{T}$, as can be seen in Lemma D.4. We do not expect this bound to be tight. Simulation results confirm that the proposed risk estimate is still accurate for all iterations, even when $T > n$, at least in the simulation settings that we tried.
>
> > **Q2:**
> Is the reason that $\hat W$ must be computed recursively simply to invert $\mathcal{M}$ ? (...) how to connect the unrolling in section 3.1 to the Kronecker expressions in 3.2.
>
> **A2:**
> Yes, the recursive computation of $\hat W$ is related to the inversion of the triangular matrix $\mathcal{M}$.
> To expand on this, as intuited in Section 3.1, unrolling the derivatives by the chain rule brings a matrix product of previous
> Jacobian matrices from the current iteration to the first. The connection to $\mathcal{M}^{-1}$ is easier to see
> from the identity
>
> $
> \begin{pmatrix}
>     1 & 0 & 0 & \cdots & 0 & 0 \\\\
>     -p_1 & 1 & 0 & \cdots & 0 & 0 \\\\
>     0 & -p_2 & 1 & \cdots & 0 & 0 \\\\
>     \vdots & \vdots & \vdots & \ddots & \vdots & \vdots \\\\
>     0 & 0 & 0 & \cdots & 1 & 0 \\\\
>     0 & 0 & 0 & \cdots & -p_{T-1} & 1
> \end{pmatrix}^{-1}
> $
> $=
> \begin{pmatrix}
>     1 & 0 & 0 & \cdots & 0 & 0 \\\\
>     p_1 & 1 & 0 & \cdots & 0 & 0 \\\\
>     p_1 p_2 & p_2 & 1 & \cdots & 0 & 0 \\\\
>     p_1 p_2 p_3 & p_2 p_3 & p_3 & \ddots & \vdots & 0 \\\\
>     \vdots & \vdots & \vdots & \ddots & 1 & 0 \\\\
>     p_1 p_2 \cdots p_{T-1} & p_2 p_3 \cdots p_{T-1} & p_3 p_4 \cdots p_{T-1} & \cdots & p_{T-1} & 1
> \end{pmatrix}.
> $
>
> On the left, the matrix has ones on the diagonal and $-p_1,...,-p_{t-1}$ just below the diagonal, like our $\mathcal M$
> but with blocks of size 1. On the right, the matrix is similar to the product of Jacobian matrices brought by unrolling the derivatives
> by the chain rule. With this in mind, the matrix $W$ in equation (8) is obtained by
> continuing to unroll the derivatives by the chain rule for $t=3,4,...$ as for $t=2,3$ in Section 3.1 (see also equation (19)).
>
> Using $\mathcal M^{-1}$ is useful to manipulate a compact notation with no product over $T$ matrices.
> However, for the implementation the matrices are computed recursively by leveraging the above structure
> (for instance, if we have computed the $t-1$-th row of the inverse, most of the next row is easily obtained by multiplying the $t-1$-th row by $p_{T-1}$).
>
> > **Weakness 1:**
> Numerators in Theorems 3.6-3.7 include unspecified $C(T,...)$. From the appendix, it appears that $C(T,...)$ scales like $T^T$, so that you can only run for
> $T \approx \log(n)/\log\log(n)$ steps before the bound is vacuous (...). I'm concerned it may not be possible for SGD to significantly decrease the loss in this number of steps, rendering the generalization estimate vacuous.
>
> **Response:**
> Yes, our current analysis provides a bound on the iteration number as $T^T$. We expect this dependence in $T$ to be an artefact of the proof. As illustrated in Figures 1-2 and further supported by additional simulations with $T > n$, the simulations suggest that the generalization error estimate is accurate for all iterations, even as $T$ diverges.
>
> Note that even though the theory of the submission incurs these constants of order $T^T$, this still allows us to pinpoint
> what the risk estimate should be for all $T$.
> While the theory only holds for $T \le \log(n)/\log\log(n)$ for which the risk estimate is consistent,
> once the definition of the estimator is known,
> revealing the form of the estimator is still useful:
> once we have identified the definition of the estimate:
>
> - simulation studies such as those provided in the submission allow
> us to check the empirical validity of the estimate beyond $T \le \log(n)/\log\log(n)$, and
> - it will be easier for future work to improve this dependence in $T$, now that identifying what the risk estimate should be has been done.
> It is common for new estimators to be proposed first, before further works refine the bounds on these estimators.
>
> Improving the dependence on $T$ appears difficult and possibly out of reach of current tools,
> even in the well-studied Approximate Message Passing (AMP) algorithms (which is also an algorithm typically studied
> in the proportional regime of interest here). The papers [1, 2] feature for instance the same dependence
> $T \le \log(n)/\log\log(n)$ for approximating the risk of AMP.
> The 2024 preprint [3] offers the latest advances on the dependence on $T$ in the bounds satisfied by AMP.
> It allows $T\asymp \text{poly}(n)$ to control certain AMP related quantities, although for the risk [3, equations (16)-(17)]
> the dependence $t=O(\log n)$ is required which is still logarithmic. This suggests that advances on this front are possible, at least
> for isotropic design for separable loss and regularizer such as those studied in [3]: Lasso or Robust M-estimation with no regularizer.
>
> Since these latest advances in [3] are obtained for specific estimates (Lasso or Robust M-estimation with no regularizer),
> it may be possible to follow a similar strategy and improve our bounds for specific examples of iterative algorithms closer to AMP
> or featuring only separable losses and penalty. But we believe following such a strategy for specific examples would be
> out of scope for the current submission, which tackles a general framework allowing iterations of the form
> (4) with little restriction on the nonlinear functions
> $\phi_t,\psi$ except being Lipschitz (and bounded by $\psi$).
>
> **Minor points**: We agree and will implement the suggestion regarding line 207 in the camera-ready version. Thanks!
>
> * * *
>
> [1]: "Finite Sample Analysis of Approximate Message Passing Algorithms".
> Rush and Venkataramanan, 2016.
>
> [2]: "An Asymptotic Rate for the LASSO Loss". Rush, 2020.
>
> [3]: "A non-asymptotic distributional theory of approximate message passing for sparse and robust regression".
> Li and Wei, 2024.

---

> > ### Comment · Reviewer_yB9W · 2024-08-10
> >
> > I thank the authors for addressing my questions and concerns, but I prefer to keep my current score. I am skeptical of the “revealing the form of the estimator” claim as an estimator that works for small T at might require additional correction terms to work for large T. I understanding fixing this issue may be challenging and can be reasonably left to future work, but it would be helpful to include a discussion of C(T) and the disconnect between the theory and the experiments somewhere below Theorem 3.6.

---

> > > ### Author Response · Authors · 2024-08-14
> > >
> > > Thanks for reading the rebuttal and the additional comment. We agree to include, in the camera-ready version, a discussion of C(T), the disconnect between the theory and simulations somewhere below Theorem 3.6, as well as the pointers to the literature on recent advances to lower dependence on T (including Li and Wei, 2024).

---

### Official Review · Reviewer_1ZGt · 2024-07-12

**Soundness:** 4
**Presentation:** 4
**Contribution:** 2
**Rating:** 4
**Confidence:** 4

**Summary:**

This manuscript aims at finding computational efficient measure of generalization performance for high-dimensional robust regression with regularization. In this scenario, when loss function is not quadratic, estimating the out-of-sample error $\| \Sigma^{1/2}(b_t - b^* ) \|^2$ (where $\Sigma$ is the covariance matrix, $b_t$ is the $t$-iteration variable and $b^*$ is the ground truth) for iterative methods like GD, SGD or proximal GD can be very difficult and is not well-explored in previous literature. To overcome this issue, the authors propose a new estimator for the out-of-sample error for a general class of gradient methods and show that the difference converges to zero in probability when sample size goes to infinity.

**Strengths:**

This paper is generally quite well-written. The proof and result appears to be reasonable. Empirical evidences confirm that the proposed estimator is indeed effective and accurate.

**Weaknesses:**

Despite being more general, this paper seems to constitute *limited* progress towards the community. When comparing it with the work by Bellec and Tan, it is not hard to notice the estimator $\hat{r}_t$ in Theorem 3.6 is a trivial extension of Bellec and Tan's estimator from square loss to convex loss. The exact form of the proposed estimator and its guarantee in Theorem 3.6 are very similar to the result of the mentioned paper. Also, the proof looks quite straighforward and extending from square loss to convex loss does not rely on technical innovations. In this regard, I believe this paper falls short in getting accepted to NeurIPS, unless the authors could provide more convincing arguments how this result is *fundamentally* different from previous works.

**Questions:**

There are no further questions.

**Limitations:**

This work is purely theoretical and has no negative influences.

---

> ### Author Rebuttal · Authors · 2024-08-07
>
> Thanks for the comments and the opportunity to clarify our contributions.
>
> > **Weakness:**
> Despite being more general, this paper seems to constitute limited progress towards the community. When comparing it with the work by Bellec and Tan [5], it is not hard to notice the estimator in Theorem 3.6 is a trivial extension of Bellec and Tan's estimator from square loss to convex loss. The exact form of the proposed estimator and its guarantee in Theorem 3.6 are very similar to the result of the mentioned paper. (...)
>
> ## **Response:**
>
> While some notation and techniques (in particular probabilistic lemmas based on Gaussian integration  by parts) are reused compared the previous result of [5] for the square loss (we did not try to hide this with new notation, and we did not reinvent the wheel unless necessary), let us point out some significant technical differences.
>
> - In terms of applicability, estimators applicable to SGD (or proximal SGD) and robust errors are a significant step forward compared to the full-batch and square loss setting of [5].
> - For the square loss, residuals and gradients are the same. For robust loss functions or the iterative algorithms studied here,
> residuals and gradients are different; this defines a different structure which can be seen in the estimators (involving both the residuals
> and the gradients), as well as in the weight matrices $\hat A, \hat K, W$. In the square loss only two weight matrices appear, and it was a surprise to us to find the necessity to introduce three matrices to analysis the problem for proximal SGD or robust losses.
> - [5] studies Gaussian noise. We allow heavy-tailed noise with infinite variance. This requires different tools to control the noise,
> and the resulting rate is also different, with the rate explicitly depending on the noise through the quantity $\mathbb{E}[\min\\\{1, ||\varepsilon||/n\\\}]$
> appearing in Theorem 3.6.
>
> **More importantly, directly generalizing the approach in [5] fails for SGD.**
> This failure for SGD resides in the difference between the matrices $\hat K$ and $\tilde K$
> in Equations (26) and (27). Generalizing the approach of [5] leads our research to the
> matrix $\tilde K$ in (26). In order to build an estimate of the weight matrix $W$ of interest
> (as discussed in the submission, $W$ cannot be used directly as $\Sigma$ is typically unknown),
> one wishes to invert $\tilde K$ in the approximation $\tilde A \approx \tilde K W$.
> This inversion fails for SGD for small (but still very realistic) batch sizes of order $0.1n$.
>
> The matrix $\tilde K$ is lower triangular, and the reason for the lack of invertibility
> of $\tilde K$ can be seen in the diagonal terms equal to $\text{Tr}[S_tD_t]$ in (26),
> where $S_t \in \\\{0,1\\\}^{n \times n}$ is the diagonal matrix with 1 in position $ii$
> if and only if the $i$-th observation is used in the $t$-th batch.
> This diagonal element of $\tilde K$ can easily be small (or even 0) for small batches,
> if the batch only contains observations such that $(D_t)_{ii}$ is 0 or small.
>
> For SGD and proximal SGD, we solved this failure of the invertibility of $\tilde K$ by using out-of-batch
> samples in the construction of $\hat K$ and $\hat A$, in order to avoid $S_t$ in the diagonal elements
> of $\hat K$ in equation (27). This is the key to making these estimators work for SGD and proximal SGD,
> and this use of out-of-batch samples could be anticipated by reading or
> generalizing [5] (which only tackles the square loss with full-batch
> gradients).
>
> This phenomenon is seen in generic SGD simulations, for instance with the Huber loss and
> $n, p, T = 4000, 1000, 20$ and batch_size equal to $n/10$. From iteration 10 to 19, the diagonal elements of $\tilde K$ are close to 0, while using out-of-batch samples in $\hat K$ provides diagonal values bounded away from 0, and thus numerically stable invertibility of the triangular matrix $\hat K$:
>
> | $t$  | $\tilde{K}_{tt}/n$ |   $\hat{K}_{tt}/n$ |
> |---:|----:|----:|
> | 10 | 0.04 | 0.4  |
> | 11 | 0.05 | 0.42 |
> | 12 | 0.04 |  0.43 |
> | 13 | 0.04 | 0.44 |
> | 14 | 0.04 | 0.46 |
> | 15 | 0.05 | 0.47 |
> | 16 | 0.05 | 0.49 |
> | 17 | 0.05 | 0.5  |
> | 18 | 0.05 | 0.51 |
> | 19 | 0.06 | 0.52 |
>
> While using $\tilde K,\tilde A$ (without using out-of-batch samples) is successful for estimating the risk for large batch sizes (above $0.3n$), it quickly deteriorates as the batch size decreases: in the same setting as above, with 100 repetitions, the true and risk estimates using $\tilde K$ and $\hat K$ give
>
> | $t$ | True risk | Estimate using $W$ | Using $\hat A \hat K^{-1}$ | Using $\tilde A \tilde K^{-1}$  |
> |:---|---:|----:|----:|----:|
> | 10 | 4.67 | 4.66 | 4.66 | 4.4  |
> | 11 | 4.29 | 4.29 | 4.29 | 3.98 |
> | 12 | 3.94 | 3.94 | 3.94 | 3.64 |
> | 13 | 3.63 | 3.63 | 3.63 | 3.33 |
> | 14 | 3.35 | 3.35 | 3.35 | 3.04 |
> | 15 | 3.09 | 3.1  | 3.1  | 2.77 |
> | 16 | 2.86 | 2.87 | 2.87 | 2.54 |
> | 17 | 2.65 | 2.66 | 2.66 | 2.34 |
> | 18 | 2.46 | 2.47 | 2.47 | 2.17 |
> | 19 | 2.29 | 2.3  | 2.3  | 2    |
> | 20 | 2.13 | 2.14 | 2.14 | 1.83 |
>
> The direct generalization from [5], that does not leverage out-of-batch samples (right column), is inconsistent while our proposed estimate leveraging out-of-batch samples is consistent.
>
> Note that beyond simulations, it wasn't clear at first that using out-of-batch samples would provably work. After significant trial and error we eventually found the serendipitous combination (cf. line 440) of the probabilistic identities that grants the approximation $\hat A\approx\hat K W$ in the row space of $F$ in eq (35).
>
> **Thanks for raising this point.** Highlighting and explaining this significant departure from [5] on the use of out-of-batch samples is something that was admittedly overlooked in the main text of the initial submission, and we will use the extra page available for the camera ready version to clearly explain this.
>  **We kindly suggest to reconsider the review rating in light of this.** We would be happy to provide further clarifications if needed.

---

> > ### Comment · Reviewer_1ZGt · 2024-08-14
> >
> > I thank the reviewers for addressing my concerns. The detailed explanations regarding the technical difficulties and the empirical evidence provided are very helpful. I would like to slightly increase my score, however, I am still very dubious about the contribution and novelty of this work. I took some time to re-read the proof and response of the authors. While I acknowledge that, directly generalizing [5] to SGD and non-square loss is not possible, I still question if this technique is truly innovative. The invertibility of $\tilde K$ seems to be solved by easy tricks and the rest proof still looks very similar to [5]. As a result, I change my score to 4.

---

> ### Author Response · Authors · 2024-08-14
>
> Thanks for reading the rebuttal and taking the time to re-read the proof. One challenge that arose when attempting to leverage out-of-batch samples, is that we are now manipulating quantities that are not involved in the iterative algorithm or its derivatives. This leads to some difficulty in obtaining bounds such as (35) around line 444 where the approximation between $\hat A$ and $\hat KW$ (involving out-of-bag samples for invertibility of $\hat K$) holds in the row space of $F$ (the stochastic gradient matrix, that does not involve out of bag samples). What we would like to point out as a final remark with this, is that it is subjective to argue about the challenges/difficulty of a proof once the final product is finished, since the final product does not showcase the challenges and difficulty necessary to produce it.
>
> In any case, many thanks for your comments and work on this manuscript. We believe that case of (proximal) SGD with robust losses is important for the NeurIPS readership and community. We will emphasize the role and necessity to use out-of-bag samples, and how this departs from [5], in the camera-ready version (should the paper be accepted).

---

### Official Review · Reviewer_3EXM · 2024-07-13

**Soundness:** 3
**Presentation:** 3
**Contribution:** 3
**Rating:** 7
**Confidence:** 3

**Summary:**

#### Summary
This paper examines the generalization performance of iterates produced by Gradient Descent (GD), Stochastic Gradient Descent (SGD), and their proximal variants in high-dimensional robust regression problems. The paper introduces estimators that accurately track the generalization error of the iterates along the trajectory of the iterative algorithms. These estimators are proven to be consistent under certain conditions and are illustrated through several examples, including Huber regression, pseudo-Huber regression, and their penalized variants with non-smooth regularizers.

**Strengths:**

#### Strengths
1. **Innovative Methodology**: The introduction of estimators that track the generalization error of iterates is novel and provides a significant contribution to the field of robust regression.
2. **Theoretical Rigor**: The paper provides a thorough theoretical foundation for the proposed estimators, including consistency proofs and detailed mathematical derivations.
3. **Practical Relevance**: The approach is applicable to a variety of robust regression problems, including those with heavy-tailed errors and non-smooth regularizers.
4. **Empirical Validation**: Extensive simulations demonstrate the effectiveness of the proposed estimators in tracking the generalization error and determining the optimal stopping iteration.

**Weaknesses:**

#### Weaknesses
1. **Computational Complexity**: The proposed estimators involve complex calculations that may be computationally intensive, especially for large datasets. More discussion on computational efficiency and scalability is needed.
2. **Generality**: The paper focuses on specific types of robust regression problems. Extending the methodology to a broader range of regression problems would increase its impact.
3. **Comparison with Existing Methods**: The paper provides limited empirical comparisons with existing state-of-the-art methods for estimating generalization error. More comparative analysis would strengthen the validity of the proposed approach.
4. **Practical Guidelines**: While the theoretical results are robust, practical guidelines for implementing the estimators in real-world scenarios are not sufficiently detailed.

**Questions:**

#### Questions
1. **Computational Complexity**:
    - Could you provide more details on the computational complexity of the proposed estimators? How do they scale with increasing dataset size and dimensionality?

2. **Generality**:
    - The paper focuses on specific types of robust regression problems. Are there any challenges in extending the methodology to other types of regression problems, such as those with different types of regularizers or loss functions?

3. **Comparison with Existing Methods**:
    - How do the proposed estimators compare empirically with existing methods for estimating generalization error? Are there scenarios where your approach significantly outperforms others?

4. **Practical Implementation**:
    - Can you provide practical guidelines or heuristics for implementing the proposed estimators in real-world scenarios? What are the key considerations practitioners should keep in mind?

---

> ### Author Rebuttal · Authors · 2024-08-06
>
> Thank you for the insightful and encouraging feedback. Here, we respond to your comments point by point.
>
> > **Q1**: Could you provide more details on the computational complexity of the proposed estimators? How do they scale with increasing dataset size and dimensionality?
>
> **A1:**
> We first note that our proposed risk estimates, $ \hat{r}_t $ and $ \tilde{r}_t $, only require computing the iterates $ \hat{b}^t $ and the weight matrices $ W $ and $ \hat{W} $. Since $ W $ is a $ T \times T $ lower triangular matrix, and the computational complexity of each entry of $ W $ is $ O(npT) $ using the Hutchinson trace estimator, the total computational complexity of $ W $ is $ O(npT^3) $.
>
> Similarly, for the computation of $ \hat{W} = \hat{K}^{-1} \hat{A} $, the computation cost of both $ \hat{A} \in \mathbb{R}^{T \times T} $ and $ \hat{K} \in \mathbb{R}^{T \times T} $ is $ O(npT^3) $. Thus, the overall computational complexity of $ \hat{W} $ is $ O(npT^6) $, with $T^3$ coming from inverting the triangular matrix $\hat{K}$. (Note that in practice, for numerical stability we do not compute the inverse directly, but instead solve the corresponding linear system).
>
> Overall, the implementation (provided in the supplementary material) for separable loss/penalty to compute $ \hat{b}^t $ and the weight matrix avoids any operation that would be $O(\min(n,p)^3)$ such as multiplying two matrices of sizes larger $\min(n,p)\times \min(n,p)$, for instance computing the full Gram matrix $X^TX$. The implementation only performs matrix-vector products with matrices of size smaller than $\max(n,p) \times \max(n,p)$, or multiplication of a dense matrix by a diagonal matrix both of sizes smaller than $\max(n,p) \times \max(n,p)$. It never incurs an operation with complexity larger than $\max(n,p)^2$ (ignoring here the dependence on $T$). This makes it possible to run the implementation even on a laptop for $n,p$ both 10,000.
>
>
> > **Q2**: The paper focuses on specific types of robust regression problems. Are there any challenges in extending the methodology to other types of regression problems, such as those with different types of regularizers or loss functions?
>
> **A2:**
> Our analysis focuses on the performance of the iterations that can be solved using SGD or proximal SGD methods.
> This includes any gradient-Lipschitz loss function (for instance Huber) with a non-smooth penalty as illustrated in the simulations.
>
> If the regression problem has both a non-smooth data-fitting loss and a non-smooth penalty, for instance
>
> $$
> \hat{b} = \arg\min_{b \in \mathbb{R}^p} ||y - Xb||_1 + \lambda ||b||_1,
> $$
>
> we expect that one cannot use the analysis and algorithms studied here (proximal GD or proximal SGD) due to the non-differentiability of the data-fitting loss. For such optimization problems, other primal-dual algorithms are needed, such as the alternating direction method of multipliers (ADMM) and the Chambolle-Pock algorithm. However, these algorithms have a different structure, and we expect for these algorithms a significantly different result than our Lemma B.1 for instance. Because of the different structure, we expect these algorithms to require different analysis and risk estimates.
> We leave this as an interesting direction for future work.
>
> > **Q3**:
> How do the proposed estimators compare empirically with existing methods for estimating generalization error? Are there scenarios where your approach significantly outperforms others?
>
> **A3:**
> We are not aware of any existing methods that can estimate the generalization error of the iterates of proximal SGD algorithms in our settings of the proportional regime ($ n\asymp p $). In this regime, cross-validation with a finite number of folds is known to be inconsistent for instance [2, Figure 1], and diverging number of folds would be impractical computationally. Thus we did not provide a direct comparison (though we would be happy to provide empirical comparisons if reviewers suggest existing competing risk estimates that we missed). One related work is by Luo et al. (2023), which proposed an estimate for the cross-validation error of the iterates of SGD algorithms. However, their method has many restrictions on loss function and does not work for high-dimensional regression settings with $p > n$.
>
>
> > **Q4:**
> Can you provide practical guidelines or heuristics for implementing the proposed estimators in real-world scenarios? What are the key considerations practitioners should keep in mind?
>
> **A4:**
> Thank you for this question.
>
> We provide the following practical guidelines for implementing the proposed estimators:
>
> 1. If practitioners are solving a regression problem with a smooth loss function and are using SGD or proximal SGD algorithms, they can use the proposed estimator $\hat{r}_t$ to estimate the generalization error of the iterates if the covariance matrix of the features is known (either because lots of additional unlabeled data are available, making estimating $\Sigma$ possible, or because the practitioner samples the design $X$ themselves). Otherwise, if the covariance is unknown, use $\tilde{r}_t$ which uses out-of-bag samples to maintain good performance (cf. answer to Reviewer 2 for a discussion on the use of out-of-bag samples for SGD).
>
> 2. Once $\tilde{r}_t$ is computed, plot the estimated generalization error of the iterates as a function of the number of iterations. Use this plot to choose the stopping time that achieves the smallest out-of-sample error, or as an additional tool to diagnose to study the convergence of the algorithm at the population level.
>
> ---
> [1] Luo, Yuetian, Zhimei Ren, and Rina Barber. "Iterative approximate cross-validation." International Conference on Machine Learning. PMLR, 2023.
>
> [2] Rad, Kamiar Rahnama, and Arian Maleki. "A scalable estimate of the out-of-sample prediction error via approximate leave-one-out cross-validation." Journal of the Royal Statistical Society Series B: Statistical Methodology 82.4 (2020): 965-996.

---

### Decision · Program_Chairs · 2024-09-25

**Decision:**

Accept (poster)

**Comment:**

The paper introduces estimators that accurately track the generation error of iterates produced by gradient descent and SGD (and their proximal variants) on high-dimensional (proportional regime) robust regression problems.  The proposed estimators are very similar to the previous work of Bellec and Tan ’24 -- a distinction between this work and that of Bellec and Tan should be clarified.